# Incorporating EarthCARE observations into a multi-lidar cloud climate record: the ATLID Cloud Climate Product

Artem G. Feofilov[1], Hélène Chepfer[1], Vincent Noël[2], and Frederic Szczap[3]

[1] LMD/IPSL, Sorbonne Université, UPMC Univ Paris 06, CNRS, École Polytechnique, Paris, France
[2] Laboratoire d'Aérologie (LAERO), CNRS/UPS, Observatoire Midi-Pyrénées, Toulouse, France
[3] Laboratoire de Météorologie Physique, UMR6016, CNRS, Aubière, France

*Correspondence to*: Artem G. Feofilov (artem.feofilov@lmd.polytechnique.fr)

**Abstract**

Despite significant advances in atmospheric measurements and modeling, clouds' response to human-induced climate warming
remains the largest source of uncertainty in model predictions of climate. The launch of Cloud-Aerosol Lidar and Infrared Pathfinder Satellite Observation (CALIPSO) in 2006 started the era of long-term space-borne optical active sounding of the Earth's atmosphere, which continued with the CATS (Cloud-Aerosol Transport System) lidar on-board ISS in 2015 and the Atmospheric Laser Doppler INstrument (ALADIN) lidar on-board Aeolus in 2018. The next important step is the ATmospheric LIDar (ATLID) instrument from the EarthCARE mission, expected to launch in 2024.
In this article, we define the ATLID Climate Product, Short-Term (CLIMP-ST) and ATLID Climate Product, Long-Term (CLIMP-LT). The purpose of CLIMP-ST is to help evaluate the description of cloud processes in climate models, beyond what is already done with existing space lidar observations, thanks to ATLID new capabilities. The CLIMP-LT will merge the ATLID cloud observations with previous space lidar observations to build a long-term cloud lidar record useful to evaluate the cloud climate variability predicted by climate models.
We start with comparing the cloud detection capabilities of ATLID and CALIOP (Cloud-Aerosol Lidar with Orthogonal Polarization) in day- and nighttime, on a profile-to-profile basis in analyzing virtual ATLID (355nm) and CALIOP (532nm) measurements over synthetic cirrus and stratocumulus cloud scenes. We show that solar background noise affects the cloud detectability in daytime conditions differently for ATLID and CALIPSO.
We found that the simulated daytime ATLID measurements have lower noise than the simulated daytime CALIOP
measurements. This allows lowering the cloud detection thresholds for ATLID compared to CALIOP and enables ATLID to detect optically thinner clouds than CALIOP in daytime at high horizontal resolution without false cloud detection. These lower threshold values will be used to build the CLIMP-ST (short-term, related only to ATLID observational period). This product should provide an advance to evaluate optically thin clouds like cirrus in climate models compared to the current existing capability.
We also found that ATLID and CALIPSO may detect similar clouds if we convert ATLID 355nm profiles to 532nm profiles and apply the same cloud detection thresholds as the ones used in GOCCP (GCM Oriented Calipso Cloud Product). Therefore,

this approach will be used to build the CLIMP-LT. The CLIMP-LT data will be merged with the GOCCP data to get a long-term (2006-2030's) cloud climate record. Finally, we investigate the detectability of cloud changes induced by human-caused climate warming within a virtual long-term cloud monthly gridded lidar dataset over the 2008-2034 period that we obtained

from two ocean-atmosphere-coupled climate models coupled with a lidar simulator. We found that a long-term trend of opaque cloud cover should emerge from short-term natural climate variability after 4 years (possible lifetime) to 7 years (best case scenario) of ATLID merged with CALIPSO measurements according to predictions from the considered climate models. We conclude that a long-term lidar cloud record built from the merge of the actual ATLID-LT data with CALIPSO-GOCCP data will be a useful tool to monitor cloud changes and to evaluate the realism of the cloud changes predicted by climate models.

**1 Introduction**

Clouds play an important role in the radiative energy budget of Earth. The radiative effect of clouds is twofold: on the one hand, clouds reflect some of the Sun's radiance during the day, thus preventing surface warming. On the other hand, high thin clouds trap some of the outgoing infrared radiation emitted by the surface and re-emit it back to the ground, thus contributing to its heating. Overall, at global scale, clouds contribute to cool the Earth radiatively, but quantifying precisely this global

effect as well as the influence of clouds on the Earth radiative budget everywhere requires knowing the coverage of clouds, as well as their geographical and vertical distributions, temperature, and optical properties. Cloud properties are expected to change under the influence of climate warming, leading to changes in the amplitude of the overall cloud radiative cooling. But how cloud properties change as climate warms is uncertain (e.g., Zelinka et al., 2012, 2016; Chepfer et al., 2014; Vaillant de Guélis et al., 2018; Perpina et al., 2021). Cloud feedback uncertainties are an important contributor to climate sensitivity

uncertainty and therefore limit our ability to predict the future evolution of climate for a given $CO_2$ emission scenario (e.g. Winker, 2017; Zelinka et al., 2020).

Global-scale round-the-clock satellite observations of Earth's atmosphere provide invaluable information that improves our knowledge of current clouds' properties and helps to evaluate the cloud description in climate models in current climate simulations. Among the remote sensing techniques, active sounding plays a special role, because of its high vertical and

horizontal resolution and high sensitivity. The launch in 2006 of Cloud-Aerosol Lidar and Infrared Pathfinder Satellite Observation (CALIPSO, Winker et al., 2010) started the era of operational space-borne optical active sounding of the Earth's atmosphere for clouds and aerosols. It was followed with the CATS (Cloud-Aerosol Transport System) lidar on-board ISS in 2015 (McGill et al., 2015) and the Atmospheric Laser Doppler INstrument (ALADIN) lidar on-board Aeolus in 2018 (Reitebuch et al., 2020; Straume et al., 2020). The next important step is the ATLID instrument (do Carmo et al., 2021), from

the EarthCARE mission (e.g. Héliere et al., 2012; Illingworth et al., 2015), expected to launch in 2024. With this lidar, the scientific community will continue receiving invaluable vertically resolved information of atmospheric optical properties needed for the estimation of cloud occurrence frequency, thickness, and height. Cloud profiles deduced from CALIOP observations have been widely used to evaluate the cloud description in climate models (e.g., Nam et al., 2012; Cesana et al.,

2019), and have provided leads to improve this description (e.g., Konsta et al., 2012). To avoid any discrepancy in cloud definition between model and observation, and to allow consistent comparisons between clouds simulated by climate models and observed by satellite, the Cloud Feedback Model Intercomparison Project (CFMIP) has developed the CFMIP Observation Simulator Package (COSP1, Bodas-Salcedo et al., 2011), followed by COSP2 (Swales et al., 2018). These packages include a lidar simulator (Chepfer et al. 2008; Reverdy et al. 2015; Guzman et al. 2017; Cesana et al. 2019) that mimics the measurements that would be obtained by spaceborne lidars if they were overflying the atmosphere simulated by a climate model. In parallel to the COSP lidar simulator, a Level 2 and 3 cloud product named CALIPSO-GOCCP (Chepfer et al. 2008, 2010, 2013; Guzman et al. 2017; Cesana et al., 2019) was designed to ensure scale-aware and definition-aware comparison between simulated and observed clouds.

Despite the similarity of the measuring principle of ATLID and CALIOP lidars – the emitter sends a brief pulse of laser radiation to the atmosphere, and the receiver registers a time-resolved backscatter signal collected through its telescope – the sensitivity of both lidars to the same clouds is different. This is explained by differences in observational geometry, in wavelength, pulse energy and repetition frequency, in telescope diameter and detector type, in the capability of detecting molecular backscatter separately from the particulate one, in vertical and horizontal resolution and averaging, and so on. Since the CALIPSO-GOCCP algorithm cannot be applied directly to ATLID data, a specific algorithm had to be developed, which generates the ATLID cloud product CLIMP (CLIMate Product).

The present paper describes the design of the CLIMP product and its associated algorithm, developed with the following two goals in mind:

(1) On short time scales, such as the period of ATLID operation, CLIMP should help improve the current evaluation of cloud description in climate models beyond CALIPSO. From this point of view, CLIMP should take advantage of ATLID capabilities compared to CALIPSO from the point of view of evaluation of clouds in climate models, while maintaining compliance with the COSP/lidar framework.

(2) On long time scales, CLIMP should enable building a merged CALIPSO+ATLID long-term lidar cloud product, in which the same clouds are detected despite the instrumental and orbital differences between ATLID and CALIOP. From this point of view, CLIMP should maximize consistency with GOCCP. The GOCCP+CLIMP long-term dataset should describe more than 20 years of cloud profiles at global scale, which will enable the study and evaluation in climate models of inter-annual variability in cloud profiles due to multiannual climate variations (e.g., El Niño, NAO, Madden-Julian oscillation). Its analysis will moreover make possible the detection of cloud changes because of human-induced climate warming, and their evaluation in climate model simulations.

Therefore, CLIMP will be composed of two datasets named CLIMP-ST (short-term) and CLIMP-LT (long-term). Both will mainly differ by their cloud detection threshold, as we will see later in the text. This threshold is parameterized in COSP/lidar and can be easily changed when comparing simulated data to CLIMP-ST and CLIMP-LT.

The CLIMP product and algorithm inherit from the approach developed for CALIPSO-GOCCP. This algorithm processes L1 data in exactly the same way as the COSP lidar simulator does. GOCCP is part of the CFMIP-OBS database included in

Obs4Mips (Waliser et al., 2020) for model evaluation. Differences between GOCCP, NASA, and JAXA CALIOP cloud products were documented in Chepfer et al. (2013) and Cesana et al. (2019).

The three key elements of the GOCCP algorithm, which need to be kept when developing CLIMP, are:

(i) lidar profiles, which are not averaged horizontally before cloud detection to (1) keep consistency with the subgridding module SCOPS (Klein and Jacob, 1999) included in COSP that is required to respect the Eulerian framework of climate model simulations, and (2) avoid overestimation of the cloud fraction in shallow clouds (e.g., Chepfer 2008, 2013; Feofilov et al. 2022).

(ii) lidar measurements are averaged vertically every 480 m, to improve the signal-to-noise ratio (SNR) while maintaining consistency with CloudSat data used to compare with COSP/radar outputs (Marchand et al., 2009; Haynes et al., 2007). This value of 480 m can be different in CLIMP as it can be changed in COSP/lidar, but averaging the lidar signal vertically before cloud detection should remain the way to increase ATLID SNR when needed for climate mode evaluation.

(iii) cloud detection thresholds are chosen for consistency with COSP/lidar and to prevent false cloud detections in CALIOP

L1 daytime data at full horizontal resolution and 480 m-averaged vertical resolution. The cloud detection threshold can be modified in CLIMP, but then should also be modified in COSP/lidar. This threshold needs to be constant over a full dataset and cannot be scene-dependent.

We would like to stress that the main two purposes of this article are (a) to compare two space-borne lidars in terms of cloud detection and signal-to-noise ratio for given observational conditions and (b) to develop a method for merging the data from

several space borne lidars into a continuous cloud record to detect long-term changes and get a seamless cloud climatology. We assume that the calibration of the instruments is performed dynamically onboard the satellites and that the calibration coefficients and cross-talk parameters are known with high accuracy. In this case, we can study the theoretically achievable cloud detection for a given experimental setup, which is defined by a number of parameters like telescope diameter, transmission of the system, solar noise filtering, detector type, and so on. For the sake of simplicity, we do not discuss the

depolarized component of the radiation backscattered by particles, assuming that it is backscattered the same way at these wavelengths and that one can always consider a sum of parallel and perpendicular backscatter for cloud detection.

The structure of the article is as follows. In Section 2, we briefly describe the differences and similarities between ATLID and CALIOP, the formalism necessary to understand the analysis presented in the next sections, and the cloud variables used in this study. Section 3 describes the physical elements that matter for the development of CLIMP-ST. Using synthetic cloud

scenes (3.1) and a numerical chain which simulates lidar profiles observed by CALIPSO and ATLID over the cloud scenes at full spatial resolution and instantaneous time scales (3.2). In this section, we also pay specific attention to the estimates of lidar signal noise. Then we define the cloud detection scheme of CLIMP-ST (3.3) and we try to answer whether ATLID might observe optically thinner clouds in daytime than CALIOP at full horizontal resolution, a useful capability to evaluate the description of cirrus in climate models. Section 4 describes the physical elements that matter for the development of CLIMP-

LT. Section 4.1 presents the cloud detection scheme used in CLIMP-LT to detect the same cloud as CALIPSO-GOCCP, despite the instrumental differences between ATLID and CALIOP. Then we analyze a long-term (multi-decadal, monthly averaged),

global-scale space lidar virtual dataset built from climate models + COSP/lidar simulation (Sect. 4.2) to illustrate how a merged dataset "CLIMP-LT + CALIPSO-GOCCP" could help evaluate climate models' predictions of multi-decadal cloud changes (Sect. 4.3). We conclude in Section 5.

## 2. Two spaceborne lidars, lidar equation, and cloud detection

### 2.1. Differences between CALIOP/CALIPSO and ATLID/EarthCARE space borne lidars

CALIOP, a two-wavelength polarization-sensitive near-nadir viewing lidar, provides high-resolution vertical profiles of aerosols and clouds (Winker et al., 2010). Its initial orbital altitude was 705 km (now 688 km to match that of CloudSat), and its orbit is inclined at 98.05°. The lidar overpasses the equator at 1h30 and 13h30 LST (local solar time). It uses three receiver channels: one measuring the 1064 nm backscatter intensity and two channels measuring orthogonally polarized components of the 532 nm backscattered signal. Cloud and aerosol layers are detected by comparing the measured 532 nm signal return with the return expected from a molecular atmosphere (see the definitions later). The other instrumental parameters of this lidar are described in Table 1 (see also Fig. 1 of (Hunt et al., 2009) for block diagram of CALIOP).

The goals of the EarthCARE mission are "to retrieve vertical profiles of clouds and aerosols, and the characteristics of their radiative and microphysical properties to determine flux gradients within the atmosphere and fluxes at the Earth's surface, as well as to measure directly the fluxes at the top of the atmosphere and also to clarify the processes involved in aerosol-cloud and cloud-precipitation-convection interactions" (Héliere et al., 2012; Illingworth et al., 2015). The ATLID instrument onboard the EarthCARE satellite will measure the attenuated atmospheric backscatter with a vertical resolution of ~100 m and ~500 m in the altitude ranges of 0−20 km and 20−40 km, respectively. ATLID is a polarization-sensitive, high-spectral resolution lidar (HSRL), which can separate the thermally broadened molecular backscatter (Rayleigh) from the unbroadened backscatter from atmospheric particles (Mie) (Durand et al., 2007; see also Fig. 2 of (do Carmo et al., 2021)). This helps ATLID retrieve extinction and backscatter vertical profiles without assuming the extinction-to-backscatter ratio (as in CALIOP retrievals), which is poorly known, especially for aerosols (e.g. Rogers et al., 2014).

| Parameter | Symbol | CALIOP | ATLID |
|---|---|---|---|
| Altitude, [km] | $Z$ | 688[#] | 393 |
| Orbital inclination, [deg] | $I$ | 98.05 | 97.050 |
| Wavelength, [nm] | $\lambda$ | 532/1064 | 355 |
| Pulse Repetition Frequency, [Hz] | PRF | 20 | 51 (25.5)[*] |
| Horizontal distance between profiles, [m] | $\Delta x$ | 333 | 285 |
| Finest Vertical resolution (troposphere), [m] | $\Delta z$ | 30 | 100 |
| Telescope diameter, [m] | $d_{tel}$ | 1.0 | 0.6 |
| Telescope Field of view, [µrad] | $\varphi$ | 130 | 64 |
| Energy/pulse, [mJ] | $E_{pulse}$ | 110 | 35 (70)[*] |
| Footprint, [m] | $d_{fp}$ | 90 | 29 |
| Laser beam divergence [µrad] | $\theta$ | 100 | 45 |
| Solar filter bandwidth, [nm] | $\Delta\lambda$ | 0.04/0.475 | 0.71 (0.35)[**] |
| Solar filter transmission | $\xi_{filter}$ | 0.85 | 0.87 |
| Total optical system transmission | $\xi_{rec}$ | 0.67/0.68 | 0.62 |
| Detector type | | PMT/APD | CCD |
| Detector efficiency | $\gamma$ | 0.11/0.4 | 0.79mol/0.75part |
| Excess noise factor | ENF | 1.46 | 1.44 |
| Single-shot noise scale factor [photoelectons$^{1/2}$] | NSF | 5.14 | 1.0 |
| Dark current, [phot/s] | $N_{dark}$ | 1331/1.85e7 | 153 |
| Readout noise [photoelectrons] | RON | 3−5 | < 3 |

Table 1: specifications of the CALIOP and ATLID spaceborne lidars considered in this article. We gathered specifications from Hunt et al. (2009) for CALIOP and from do Carmo et al. (2021) for ATLID. (#) the nominal orbit altitude at launch was 705 km, but was lowered to 688 km in September 2018 to maintain formation flying with CloudSat; (*) the original pulse repetition frequency of ATLID laser is 51Hz at the energy of 35 mJ per pulse, but the measurements will be doubled onboard the satellite (do Carmo et al., 2021), so one can consider the effective frequency and energy per pulse to be equal to 25.5 and

70 mJ, respectively. (**) the solar filter bandwidth of ATLID is 0.71 nm, but the transmission function of the Mie channel is approximately half of that, so one should calculate the solar noise in this channel with narrower effective filter width.

When considering signal quality and performance, some parameters are in favor of CALIOP (telescope diameter, energy per pulse, solar filter bandwidth) whereas others favor ATLID (altitude, noise level). In the next section, we show how these differences affect the detectability of clouds. We excluded the multiple scattering coefficient from the table since it is an

important and complex parameter of lidar instrument, which depends on its several parameters. Instead, we discuss it in a dedicated paragraph below.

### 2.2. Lidar equation

The formalism used in this work was described in (Feofilov et al., 2022). In this section, we repeat only the basic definitions needed for understanding the material presented below. Since we will discuss both conventional (non-HRSL) and HSRL lidars,

we will introduce necessary quantities in parallel and label them correspondingly: the molecular, particulate, and total components will get the indices "mol", "part", and "tot", respectively.

An atmospheric lidar sends a brief pulse of laser radiation towards the atmosphere. The lidar optics collect the backscattered photons and drive them to a detector. The detected signal is time-resolved: supposing each photon travelled straight forward and back, each time bin corresponds to a fixed distance from the lidar to the atmospheric layer where backscattering occurred.

The propagation of laser light through the atmosphere and backwards to the detector is described by the lidar equation:

$$ATB(\lambda, z) = (\beta_{mol}(\lambda, z) + \beta_{part}(\lambda, z)) \times e^{-2 \int_{Z_{sat}}^{z} (\alpha_{mol}(\lambda, z\prime) + \eta \alpha_{part}(\lambda, z\prime)) dz\prime} \qquad (1)$$

where ATB stands for Attenuated Total Backscatter [$m^{-1}$ $sr^{-1}$], $\beta_{mol}(\lambda, z)$ and $\beta_{part}(\lambda, z)$ are the wavelength-dependent molecular and particulate backscatter coefficients [$m^{-1}$ $sr^{-1}$], $\alpha_{mol}(\lambda, z)$ and $\alpha_{part}(\lambda, z)$ are the extinction coefficients [$m^{-1}$], $Z_{sat}$ is the altitude of the satellite, $\lambda$ is the wavelength, and $\eta$ is a multiple scattering coefficient (e.g., Platt et al., 1973; Garnier

et al., 2015; Donovan, 2016).

For the HSRL lidar, one can write similar equations for the attenuated radiance backscattered from atmospheric particles and molecules (APB and AMB), respectively:

$$APB(\lambda, z) = \beta_{part}(\lambda, z) \times e^{-2 \int_{Z_{sat}}^{z} (\alpha_{mol}(\lambda, z\prime) + \eta \alpha_{part}(\lambda, z\prime)) dz\prime} \qquad (2)$$

$$AMB(\lambda, z) = \beta_{mol}(\lambda, z) \times e^{-2 \int_{Z_{sat}}^{z} (\alpha_{mol}(\lambda, z\prime) + \eta \alpha_{part}(\lambda, z\prime)) dz\prime} \qquad (3)$$

For cloud definition, we will also need to define the attenuated molecular backscatter for clear sky conditions

$$ATB_{mol}(\lambda, z) = \beta_{mol}(\lambda, z) \times e^{-2 \int_{Z_{sat}}^{z} \alpha_{mol}(\lambda, z\prime) dz\prime} \qquad (4)$$

The physical meaning of $\eta$ in Eqs. (1-3) is an increase in the number of photons remaining in the lidar receiver field of view besides the ones directly backscattered by the layer, and its value depends on the type of scattering media, FOV of the telescope, and laser beam divergence. The typical value of $\eta$ varies between 0.5 and 0.8 for commonly used lidars (Chiriaco et al., 2006;

Chepfer et al., 2008, 2013; Garnier et al., 2015; Donovan, 2016; see also Appendix B of Reverdy et al., 2015). Setting $\eta$ to 1 means no multiple scattering, and would correspond to an infinitely narrow FOV telescope combined with an infinitely small laser beam divergence. In CALIOP cloud products up to version 3, the $\eta$ was set to 0.6 for ice clouds, whereas for version 4.10 a temperature-dependent coefficient was used, which varied in between 0.46 and 0.78 (Young et al., 2018). For water clouds, the $\eta$ values are derived from the relationship developed in Hu et al., 2007 (also see Table 4 in Young et al., 2018). A detailed

modeling of $\eta$ for different cloud types observed by CALIOP and ATLID (Shcherbakov et al., 2022) shows that $\eta$ depends on the cloud thickness and type and that the ATLID values are somewhat higher than those of CALIOP. Based on these works, we set a fixed value of $\eta$ to 0.6 for CALIOP and to 0.75 for ATLID. This is an approximation and a more complex approach might be required for processing real data, but our tests show that the conclusions of the present work do not change if we vary $\eta$ within ±0.1 either for CALIOP or for ATLID (but not for both).


### 2.3 Cloud detection and cloud variables

To characterize the scattering properties of the atmosphere, it would be convenient to use some ratio of attenuated backscatter values (Eqs. 1−4), which would have a clear physical interpretation. Due to attenuation of AMB below the clouds, using it in the denominator is counterproductive, so the $ATB_{mol}(\lambda, z)$ is used instead, and the scattering ratio (SR) is defined as:

$$SR(532nm, z) = \frac{ATB(532nm, z)}{ATB_{mol}(532nm, z)} \tag{5}$$

Considering a single-pulse profile measurement, we declare a layer as cloudy if the following two conditions are met:

$$SR(532nm, z) > 5 \text{ and } ATB(532nm, z) - ATB_{mol}(532nm, z) > 2.5 \times 10^{-6} \, m^{-1} \, sr^{-1} \tag{6}$$

The second condition in Eq. 6 comes from the fact that the molecular backscatter in the upper troposphere is weak, and the fluctuation in ATB might cause a false cloud detection if only SR is used. With the second condition, the cloud detection is more robust. This definition is used in CALIPSO-GOCCP (e.g. Chepfer et al., 2008, 2010, 2013) and we suggest keeping it for other lidars to ensure the consistency between cloud products as discussed later.

In application to ATLID, this will mean using the recalculated to 532nm values of ATB, which will be estimated from (1) $\beta_{part}(355nm, z)$ and $\alpha_{part}(355nm, z)$ retrieved from the measurements (Eqs. 2 and 3) and $\beta_{mol}(532nm, z)$ and $\alpha_{mol}(532nm, z)$ retrieved or estimated from pressure-temperature profiles from reanalysis. In the numerical experiment below, we calculated $ATB_{mol}(532nm, z) = \beta_{mol}(532nm, z) \times e^{-2\int_{Z_{sat}}^{z} \alpha_{mol}(532, z')dz'}$ using the available pressure-temperature profiles and the formalism provided in (Feofilov et al., 2022). Here, we reproduce the Eq. 8 of this paper:

$$SR'(532nm, z) = \frac{(\beta_{mol}(532, z) + \beta_{part}(355, z)) \times e^{-2\int_{Z_{sat}}^{z}(\alpha_{mol}(532, z') + \eta_{355}\alpha_{part}(355, z'))dz'}}{\beta_{mol}(532, z) \times e^{-2\int_{Z_{sat}}^{z} \alpha_{mol}(532, z') \, dz'}} \tag{7}$$

In this conversion, we assume that the spectral dependence of particulate backscatter ($\beta_{part}(\lambda)$ and $\alpha_{part}(\lambda)$) is weak at the wavelengths used in this study. In (Beyerle et al., 2001) it is stated that this is generally true for cirrus. In (Voudouri et al. 2020), the values at two wavelengths agree within (relatively large) error bars. Therefore, we do not attempt to compensate for the spectral dependence. The only area where we noticed that this approach does not work for real data is the polar stratospheric region, where a direct application of Eq. 7 leads to an overestimation of polar stratospheric clouds, PSCs (Fig. 8b of Feofilov et al., 2022).

For the cloud properties, we use the same variables as in CALIPSO-GOCCP (Chepfer et al. 2010): cloud fraction CF(z), opaque cloud cover $C_{opaque}$, and opaque cloud height $Z_{opaque}$. If a given atmospheric layer was observed multiple times or if it was sampled vertically at several points, we define the cloud fraction profile CF(z) in a usual way:

$$CF(z) = \frac{N_{cld}(z)}{N_{tot}(z)} \tag{8}$$

where $N_{cld}(z)$ is the number of times the conditions of Eq. (6) are met and $N_{tot}(z)$ is the total number of measurements in this layer. The opaque cloud cover $C_{opaque}$ is used in long-time series and is defined over the 2°×2° latitude/longitude gridded data as follows:

$$SR(z) < 0.06; \quad C_{opaque} = \frac{N_{opaque\_prof}}{N_{total\_prof}} \quad\quad\quad (9)$$

where the first condition triggers the opaque cloud detection (Guzman et al., 2017), $N_{opaque\_prof}$ is the number of vertical profiles, for which an attenuation corresponding to a presence of opaque cloud was found and $N_{total\_prof}$ is the total number of measurements in 2°×2° grid box. For an individual lidar profile, $Z_{opaque}$ corresponds to an altitude of full attenuation of backscattered signal whereas for gridded data, $Z_{opaque}$ is an opaque-cloud-cover-weighted sum (Guzman et al., 2017).

## 3. The CLIMP short-term dataset

In this section, we search for useful cloud information for model evaluation that can be retrieved from the ATLID, but cannot be obtained from the CALIPSO data. For this purpose, we use high resolution cloud scenes (Sect. 3.1), simulate how they would be observed by ATLID and CALIPSO (Sect. 3.2), and compare the SR(z) profiles seen by the 2 lidars (Sect. 3.3) and the clouds detected by the 2 instruments (Sect.3.4). To address the comparability of clouds observed by two space-borne lidars, we used the existing methodology (Reverdy et al., 2015; Feofilov et al., 2022), but with a much finer-scaled cloud model, updated instrumental parameters of ATLID, and a new simulation chain, which estimates noise at the detector level and propagates it to cloud product level (the details are provided in Section 3.2.2 below). The main question we sought to answer in this section was whether ATLID can observe optically thinner clouds than CALIPSO in daytime, a useful capability to evaluate thin cirrus clouds in climate models (e.g. Berry et al., 2019). At the same time, we checked whether the chosen cloud detection parameters and instrumental properties affect the detection of highly inhomogeneous low-level thick clouds.

### 3.1 Cloud generating model

The 3DCLOUD model (Szczap et al., 2014) generates three-dimensional (3-D) spatial structures of stratocumulus, fair weather cumulus and cirrus that share some statistical properties observed in real clouds such as the inhomogeneity parameter $\rho$ (standard deviation normalized by the mean of the water content) and the Fourier spectral slope $\hat{\beta}$ close to $-5/3$ between the smallest scale of the simulation to the outer scale $L_{out}$ (where the spectrum becomes more flat). We assume that water content follows a gamma distribution. 3DCLOUD_V2 presented in Alkasem et al. (2017) is based on wavelet framework instead of Fourier framework. First, 3DCLOUD assimilates meteorological profiles (humidity, pressure, temperature and wind velocity) and solves drastically simplified basic atmospheric equations in order to simulate 3D water content. Second, the Fourier filtering method is used to constrain the intensity of mean water content, $\rho$, $\hat{\beta}$, and $L_{out}$, values provided by the user (Hogan and Illingworth, 2003; Kärcher et al., 2018).

Conditions of simulations to generate the stratocumulus in this study (see Fig. 1 and Fig. 1b) are identical to those used in Szczap et al. (2014) for the DYCOMS2-RF01 case (the first Research Flight of the second Dynamics and Chemistry of Marine Stratocumulus) for the marine stratocumulus regime (Stevens et al.,2005). We have only changed the number of voxels in the $x$, $y$ and $z$ direction to $N_x = N_y = 1000$ and $N_z = 50$, respectively. The corresponding spatial resolutions were set to $\Delta_x = \Delta_y = 100$ m and $\Delta_z = 24$ m, respectively. The vertical extension of the simulated area is still $L_z = 1200$ m, but the horizontal extensions for this study are $L_x = L_y = 100$ km.

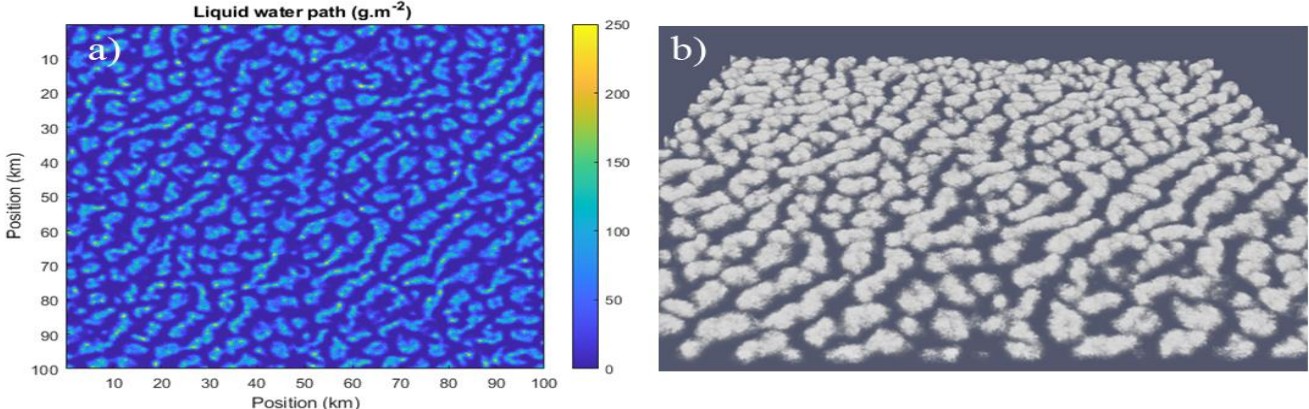

**Figure 1: Examples of the stratocumulus generated with 3DCLOUD : (a) 2-D ice water path and (b) its volume rendering.**

Ice water path (g.m⁻²)

c)

d)

**Figure 2: Examples of cirrus generated with 3DCLOUD_V3 : (a) ice water path and (c) its volume rendering; (c) IWC PDF ; (d) Mean 1-D power spectrum of IWP (red curves) and of IWC (blue curve) following *x,y* and z direction (solid, dash dot and dashed**

line, respectively). A theoretical power spectrum with spectral slope $\hat{\beta} = -5/3$ is added_ = −5/3 (dashed black line). A dotted vertical black line indicates the outer scale $L_{out} = 20$ km.

If the number of voxels is large, the 3DCLOUD and 3DCLOUD_V2 are very time-consuming (see Table 1 in Szczap et al., 2014) and cannot assimilate the fractional coverage for cirrus cloud. Therefore, we have developed 3DCLOUD_V3 that overcomes these two drawbacks for the cirrus cloud. This model will be published elsewhere. Here, we present only an outline
of the 3DCLOUD_V3 algorithm.

To increase the calculation speed in 3DCLOUD_V3, we generate clouds using modified statistic tools developed as part of stage 2 of 3DCLOUD. The first stage of 3DCLOUD (i.e. the step of solving simplified basic atmospheric equations, which is very time-consuming) is no longer carried out in 3DCLOUD_V3. Thereby, 3DCLOUD_V3 can be seen as a purely stochastic cirrus cloud generator. The user has to provide, in addition to $N_x$, $N_y$, $N_z$, $\Delta_x$, $\Delta_y$, and $\Delta_z$ , the mean Ice Water Path (IWP),
$L_{out}$ , the shape of the vertical profile of Ice Water Content (IWC), $\rho(z)$, $\hat{\beta}(z)$ , and of horizontal wind velocity components $u(z)$ and $v(z)$, and finally the cloud fraction $CF$. The shape of the vertical profile of IWC can also be stipulated (rectangular, upper triangle, lower triangle and isosceles trapezoid (Feofilov et al., 2015)). The algorithm works as follows:

(1) Generation of a 3D isotropic field with a Gaussian probability density function (PDF) from a 3D inverse Fourier transform assuming random phase for each Fourier amplitude and a 3D spectral energy density with 1D spectral slope $\hat{\beta}$ close to −5/3
between the smallest scale of $L_{out}$.

(2) Transformation of the 2-D Gaussian PDF to a 2-D Gamma PDF at each $z$ level, satisfying the values of IWC(z), $\rho(z)$, and $\hat{\beta}(z)$.

(3) Horizontal displacement, at each $z$ level, of 2-D IWC (to simulate fall streaks) computed from $u(z)$ and $v(z)$, based on the model of sedimentation proposed by Hogan and Kew (2005). In 3DCLOUD_V3, the user can choose the value of the
290 sedimentation velocity: either constant or function of IWC (see formula into Fig.12 in Heymsfield et al., 2017). Alternatively, the wind velocity vertical profile can be computed from a constant value of the vertical wind shear prescribed by the user; in this case, the user has also to provide the "generated-level height" as explained in Hogan and Kew (2005).

(4) Iterative modification of the vertical profile of the cloud cover in order to obtain the $CF$ value prescribed by the user.

Figure 2 demonstrates the examples of 2D IWP and the 3D IWC volume rendering of the cirrus generated with 3DCLOUD_V3,
where $N_x = N_y = 1000$, $N_z = 100$, $\Delta_x = \Delta_y = 100$ m and $\Delta_z = 20$ m. The mean IWP is set to 1 g m$^{-2}$. The IWC vertical profile shape is "rectangular". The geometric depth is 2 km. The outer scale is $L_{out} = 20$ km. We set the constant vertical wind shear to 5 m s$^{-1}$ km$^{-1}$ in the $x$ and $y$ directions and the generated-level height is 400 m under the cloud top. The inhomogeneity parameter of IWC is $\rho = 0.4$. The spectral slope β is equal to $-5/3$. Figure 2c shows the gamma-like PDF of the IWC (we ignored null values) and Fig. 2d shows the mean power spectra of IWP (and IWC) along $x$ and y directions (and
z direction), with 1-D spectral slope close to −2.0 (−1.3) between of $L_{out} = 20$ km and finest spatial resolution. As expected, values of spectral slope of IWP are smaller than those of IWC (i.e. IWP signal is "smoother" than IWC signal) because the IWP is the vertically integral quantity of the IWC. One can note that the IWC spectral slope is slightly smaller than the

prescribed theoretical value $\hat{\beta} = -5/3$ because of the many null values of the IWC; we plan to remove this bias in the final version of 3DCLOUD_V3.

## 3.2 Numerical chain to simulate of cloud observations by CALIOP and ATLID at high resolution

### 3.2.1 Creating pseudo-orbits

We performed the following numerical experiment, outlined in a flowchart in Fig. 3. First, we created a gridded global atmosphere from the output of the U.S. Department of Energy's Energy Exascale Earth System Model (E3SM) atmosphere model (EAM) version 1 (EAMv1; Rasch et al., 2019) for the conditions of autumn equinox in Northern Hemisphere. Since we wanted to address both high- and low-level cloud detection, we picked up only the tropical part of the orbit between 5°S and 5°N and used this data as a set of smooth "background" profiles. Since this model does not provide the small-scale variability needed for our experiment, we used the subgrid model described in Section 3.1, which generates realistic cloud profiles at grid comparable or finer than the distance between two consecutive footprints of studied lidars. To address the most challenging observation conditions, we picked up two cloud types: (1) thin cirrus with optical depths (τ) of about 0.03−0.1 per layer (Sassen and Comstock, 2001) and (2) stratocumulus clouds with their high horizontal variability and large optical depths (up to 30, but with about one third of semi-transparent clouds). These clouds were simulated using an updated 3DCLOUD_V3 model (see Sect. 3.3) and provided as gridded sets of ice water content (IWC) and liquid water content (LWC) values for cirrus and stratocumulus clouds, respectively. We do not consider another challenging case, a thin cloud layer above a highly reflective cloud, but the daytime noise estimated for stratocumulus scene will give an idea of what background noise will be interfering with the useful cloud signal in this case.

These gridded sets were converted to pseudo orbits by slicing them along the diagonal lines and arranging the slices into "lidar curtains", each comprising 20000 individual profiles and split to daytime and nighttime parts, 10000 profiles each. This way we got almost seamless cloud distributions, which followed the variability prescribed by 3DCLOUD_V3 model and at the same time resembled parts of real lidar orbits. We show the most representative parts of these pseudo orbits in Fig. 4 and 5 for cirrus and stratocumulus clouds, respectively, and we discuss them below.

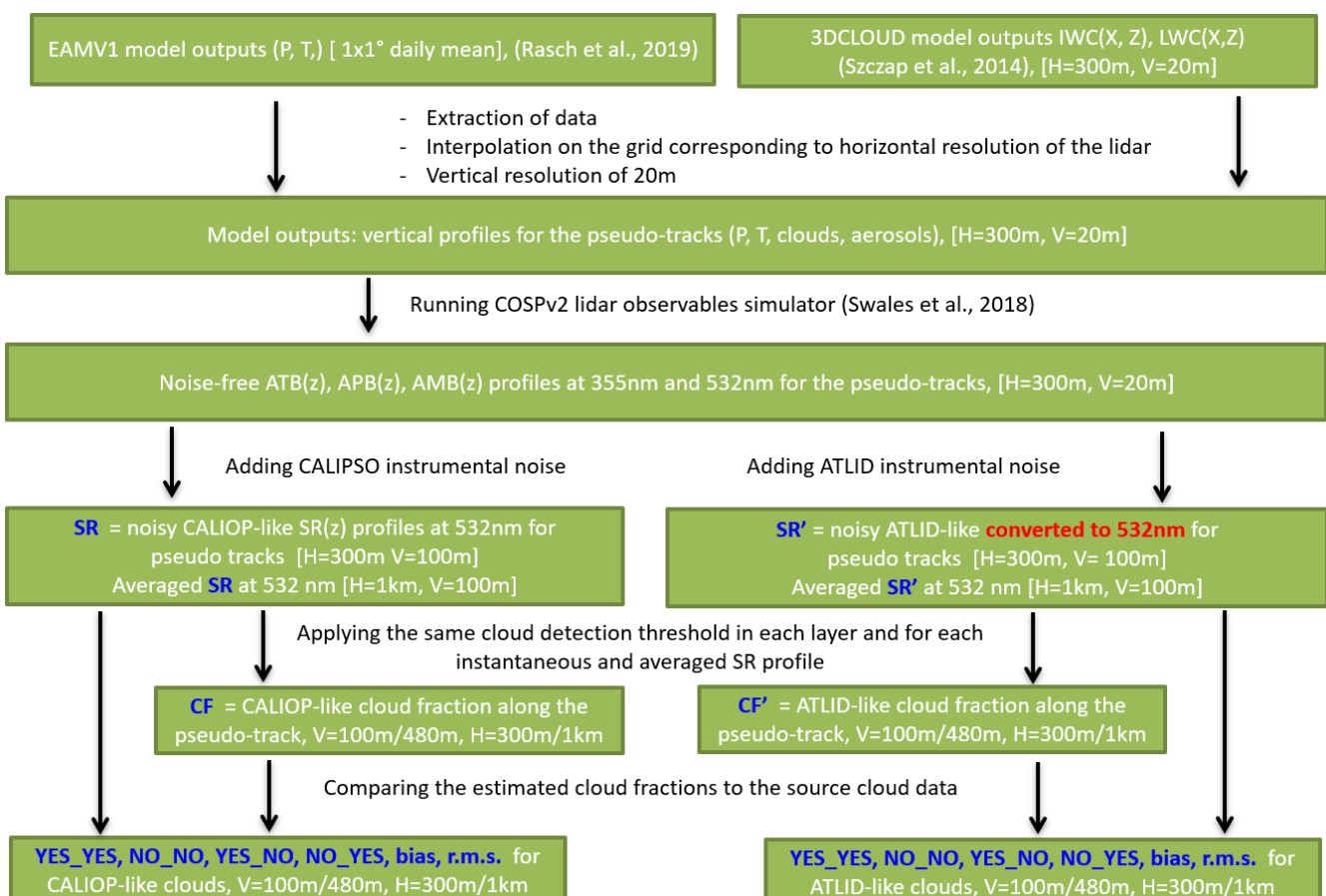

**Figure 3: Flowchart explaining the numerical experiment on comparing clouds retrieved from CALIOP and ATLID observations. Green boxes list the input and output data. Black text between boxes describes actions performed on each dataset. Blue text in the boxes marks the datasets used in the estimation. White text in square brackets in the boxes indicates horizontal (H) and vertical (V) resolutions of the datasets. Note that the ATLID SR' values are estimated at 532nm (see Sect. 2.3).**

With these two datasets covering both the daytime and the nighttime scenes, we performed a full series of simulations, explained in Fig. 3. Namely, we fed the high-resolution atmospheric inputs described above to the CALIOP and ATLID simulators (Chepfer et al., 2008; Reverdy et al., 2015) included into the Cloud Feedback Model Intercomparison Project Observational Simulator Package, v2 (COSP2) simulator (Swales et al., 2018). These simulators do not account for instrumental noise effects, so their outputs were processed by a third part of the simulation chain (Fig. 3), which estimates noise and its propagation in the lidar system.

### 3.2.2 Estimating lidar signals and noise

As mentioned above, the outputs of COSPv2 simulator are the noise-free $APB(355nm, z)$ and $AMB(355nm, z)$ profiles for ATLID and noise-free $ATB(532nm, z)$ profiles for CALIOP, both calculated at the horizontal resolution of 300 m and vertical resolution of 20 m. To estimate the noise for these profiles and to propagate it further to $SR(532nm, z)$ for CALIOP and to

recalculated $SR'(532nm, z)$ for ATLID, we calculated the signals from the scratch using the corresponding instrumental parameters introducing measurement noise as follows.

We start with a laser emission and estimate the number of emitted photons per sounding pulse, measured at the output of the sounding unit:

$$N_{em} = \frac{E_{pulse}}{hc/\lambda} \times \varsigma \tag{10}$$

where $E_{pulse}$ is the energy per laser pulse, $h$ is Planck's constant, $c$ is the speed of light, $\lambda$ is the wavelength, and $\varsigma$ is an effective coefficient of optical throughput of the emission path of the lidar. In the present work, it is assumed that there is no optical loss in the emission path and $\varsigma$ is equal to 1. The numerical solution of the equations (1)−(3) yields the number of photons per range gate $(t_i, t_i + \Delta t_i)$, which we will denote as $t_i$, coming through the CALIOP's receiver or through the ATLID's receiver before splitting to molecular and particulate components in HSRL module (compare to Eqs. (2) and (3)):

$$N_{rec}^{tot}(t_i) = N_{rec}^{mol}(t_i) + N_{rec}^{par}(t_i) \tag{11}$$

$$N_{rec}^{mol}(t_i) = N_{em} \times \beta_{mol}(i) \times \Delta z_i \times \Omega(z_i) \times \xi_{rec} \times e^{-2\sum_{j=0}^{i}(\alpha_{mol}(j)+\eta\alpha_{part}(j))\Delta z_j} \tag{12}$$

$$N_{rec}^{par}(t_i) = N_{em} \times \beta_{par}(i) \times \Delta z_i \times \Omega(z_i) \times \xi_{rec} \times e^{-2\sum_{j=0}^{i}(\alpha_{mol}(j)+\eta\alpha_{part}(j))\Delta z_j} \tag{13}$$

$$\Omega(z_i) = \pi \times \frac{(d_{tel}/2)^2}{z_i^2} \tag{14}$$

where $N_{rec}^{mol}(t_i)$ $and$ $N_{rec}^{par}(t_i)$ are the photons backscattered by molecules and particles, respectively, $\Omega(z_i)$ is an altitude-dependent solid angle with $z_i$ corresponding to time-of-flight $t_i$ between the satellite and the measured layer $i$, and $\xi_{rec}$ is the receiver's transmission.

For an HSRL lidar, the molecular and particulate components are supposed to be registered individually, but this separation is not ideal because of the cross-talk between the channels: a part of molecular backscatter comes at the same wavelength as the original laser radiance and it "contaminates" the particulate channel, which is centered at this wavelength. Overall, the HSRL system is characterized by four crosstalk coefficients, $C_{mm}$, $C_{pp}$, $C_{mp}$, and $C_{pm}$. The first two show a contribution of the molecular and channels to themselves, and in the ideal HSRL they should be equal to 1. The second pair shows how much energy "leaks" from a molecular channel to a particulate one and vice versa. In the ideal HSRL, these coefficients should be equal to 0. In the operational retrieval, these coefficients will be determined through a continuous calibration procedure performed on the orbit. For this exercise, we estimated these coefficients from the Fabry-Perot interferometer spectral curves (Cheng et al., 2013): $C_{mm} = 0.815$, $C_{pp} = 0.60$, $C_{mp} = 0.185$, and $C_{pm} = 0.40$. In the case of non-ideal HSRL, the number of photoelectrons produced by each detector per range gate is as follows:

$$N_{det}^{mol}(t_i) = \gamma \times \xi_{rec} \times \left(N_{rec}^{mol}(t_i) \times c_{mm} + N_{rec}^{par}(t_i) \times c_{pm}\right) \tag{15}$$

$$N_{det}^{par}(t_i) = \gamma \times \xi_{rec} \times \left(N_{rec}^{mol}(t_i) \times c_{mp} + N_{rec}^{par}(t_i) \times c_{pp}\right) \tag{16}$$

where $\gamma$ and $\xi_{rec}$ are the detector's quantum efficiency and transmittance of the optical path, respectively. To come back to "pure" $N_{rec}^{mol}(t_i)$ and $N_{rec}^{par}(t_i)$ used in the retrieval, one has to solve this system:

$$N_{rec}^{mol}(t_i) = N_{det}^{mol}(t_i) \times k_a + N_{det}^{par}(t_i) \times k_b \tag{17}$$

$$N_{rec}^{par}(t_i) = N_{det}^{mol}(t_i) \times k_c + N_{det}^{par}(t_i) \times k_d \tag{18}$$

$$k_a = \frac{c_{pp}}{\kappa}; k_b = \frac{-c_{pm}}{\kappa}; k_c = \frac{-c_{mp}}{\kappa}; k_d = \frac{c_{mm}}{\kappa}; \kappa = \gamma \times \xi_{rec} \times (c_{mm}c_{pp} - c_{pm}c_{mp}) \tag{19}$$

Besides the components related to atmospheric backscatter properties, the $N_{det}^{mol}(t_i)$ and $N_{det}^{par}(t_i)$ are affected by "parasite" solar backscattered photons during the daytime, which are not correlated with the laser shots. To estimate solar background add-on to $N_{det}^{mol}(t_i)$ and $N_{det}^{par}(t_i)$, one has to solve the radiative transfer equation for the radiation emitted by the Sun, backscattered by air and particles in the atmosphere and by the surface in the direction of the spaceborne lidar, and attenuated by the atmospheric layers:

$$N_{sol}(t_i) = \Delta t_i \times N_{sol}^{TOA}(\lambda) \times \left[R_{sol}^{atm} + R_{sol}^{surf}\right] \tag{20}$$

$$R_{sol}^{atm} = \int_{Z_{sat}}^{0} \left(\beta_m(z) \times \phi_m(SZA) + \beta_p(z) \times \phi_p(SZA)\right) \times \cos(SZA)^{-1} \times$$

$$\exp\left\{-2 \int_{z_{sat}}^{z} \left(\alpha_m(z') + \alpha_p(z')\right) \cos(SZA)^{-1} dz'\right\} dz \tag{21}$$

$$R_{sol}^{surf} = A_{surf} \times \phi_{surf}(SZA) \times \exp\left\{-\int_{z_{sat}}^{z} \left(\alpha_m(z) + \alpha_p(z)\right) \cos(SZA)^{-1} dz\right\} \times \int_{Z_{sat}}^{0} \left(\alpha_m(z) + \alpha_p(z)\right) dz \tag{22}$$

where $N_{sol}^{TOA}(\lambda)$ is a top of atmosphere solar flux at wavelength $\lambda$ and for filter width $\Delta\lambda$, $R_{sol}^{atm}$ and $R_{sol}^{surf}$ represent the proportion of the incoming solar radiance reflected in the direction of lidar, $\phi_m(SZA)$, $\phi_p(SZA)$, and $\phi_{surf}(SZA)$ are the scatter plots for the angle between the sun and the nadir view of lidar for molecular scattering, scattering on particles, and scattering from surface, respectively, $z_{sat}$ is the altitude of a satellite, and $A_{surf}$ is the surface albedo. We assumed Lambertian scattering from the surface with albedo equal to 0.08 for ocean and 0.15 for land (arbitrary values), we used Rayleigh scattering phase function for the molecular component, and we used the geometric optics phase function approximation for particulate scattering.

The solar photons pass through the optical system and HSRL, hit the detectors, and produce the "solar noise photoelectrons":

$$N_{det.sol.}^{mol}(t_i) = \gamma \times \xi_{rec} \times \xi_{mol.sol.} \times N_{sol}(t_i) \tag{23}$$

$$N_{det.sol.}^{par}(t_i) = \gamma \times \xi_{rec} \times \xi_{par.sol} \times N_{sol}(t_i) \tag{24}$$

where $\xi_{mol.sol.}$ and $\xi_{par.sol.}$ represent the convolution of the solar filter spectral curve with the interferometric spectral curve for a given channel (see the comment on solar filter bandwidth in Table 1). In addition to solar noise, there is always a dark current of the detector, $N_{dark}$, and readout noise, RON, which are added to the signal. Since the $N_{det.sol.}^{mol}(t_i)$ and $N_{det.sol.}^{par}(t_i)$, $N_{dark}$, and RON are registered along with $N_{det}^{mol}(t_i)$ and $N_{det}^{par}(t_i)$ during daytime, they enter the Eqs. (15) and (16) and affect the retrieval. For the non-HSRL lidar:

$$N_{det.sol.}(t_i) = \gamma \times \xi_{rec} \times \xi_{tot.sol.} \times N_{sol}(t_i) \tag{25}$$

where $\gamma$ and $\xi_{rec}$ stand for the corresponding parameters of non-HSRL lidar and $\xi_{tot.sol.}$ represents the transmission coefficient of a solar rejection filter, which is equal to a ratio of an integral of the spectral transmission curve of the filter to a full spectral width of the filter. A quick back-of-envelope estimate of the ratio of solar photons coming to the particulate detector of ATLID to number of solar photons reaching the surface of CALIOP's detector per same sampling interval is:

$$\frac{N_{sol}^{TOA}(355nm) \times d_{tel.ATLID}^2 \times d_{fp.ATLID}^2 \times Z_{CALIOP}^2 \times \xi_{rec.ATLID} \times \xi_{par.sol.}}{N_{sol}^{TOA}(532nm) \times d_{tel.CALIOP}^2 \times d_{fp.CALIOP}^2 \times Z_{ATLID}^2 \times \xi_{rec.CALIOP} \times \xi_{tot.sol.}} = \frac{(1162.8 \times 355) \times 0.6^2 \times 690^2 \times 29^2 \times 0.62 \times (0.35 \times 0.87)}{(1900.0 \times 532) \times 1.0^2 \times 393^2 \times 90^2 \times 0.67 \times (0.04 \times 0.85)} = 0.38 \tag{26}$$

so, at the first sight, the ATLID's retrieval should be less solar-contaminated than CALIOP. But, this ratio alone is not enough for such a conclusion because the solar photons should be compared to useful signal. Below, we show the results of simulations for two atmospheric scenarios which consider two-way radiative transfer both for solar radiance and for lidar sounding radiance and add the noise of the remaining detection path.

Now, when all the components of the signal are known, we can estimate the daytime and nighttime signal and noise and propagate them to the retrieved parameters. It is important to mention that the instruments compared in this work use the detectors of different types. Namely, CALIOP lidar uses a photomultiplier tube (PMT) whereas ATLID lidar detects the backscatter with the help of charge-coupled device (CCD). Besides different characteristics like gain or dark current (see Table 1), these detectors are not the same in terms of applicable noise statistics (Liu et al., 2006). Even though the incoming photon flux distributions for both instruments are Poisson, the photoelectrons produced by the PMT do not follow a strict Poisson distribution. It is known that for Poisson-distributed signals, a one-to-one relationship exists between the mean and the variance of the photocurrent. As Liu et al., (2006) show, the mean and the variance of the PMT photocurrent are also proportional, but not one to one, and the corresponding noise scale factor (NSF) has to be applied to estimate random errors for lidar systems using PMTs or avalanche detectors. The NSF is linked to an excess noise factor, ENF, but it is not equal to it. For the PMTs with identical gain factors $m$ for each dynode, the ENF is given by (Kingston, 1978; Liu and Sugimoto, 2002):

$$ENF = \frac{m}{m-1} \tag{27}$$

For the analog detection, the NSF in the multiplied-photoelectron domain can be either calculated from the detector's ENF and gain or estimated from the solar-noise dominated signals (Liu et al., 2006):

$$NSF = \frac{\sigma(N_{det})}{\sqrt{\langle N_{det} \rangle}} = \frac{\sqrt{var(N_{det})}}{\sqrt{\langle N_{det} \rangle}} \tag{28}$$

where $\sigma(N_{det})$, $var(N_{det})$, and $\langle N_{det} \rangle$ are a standard deviation, variance, and mean of the signal, respectively. The NSF value provided in the CALIPSO L1 version 4.10 files is equal to 5.14. However, using this value in synthetic noise calculations leads to an overestimation of the daytime noise, so for the calculations below we took a more conservative value $NSF = 3.16$. One can write the expressions for the variances of CALIOP and ATLID signals in the analog detection domain through the number of photoelectrons calculated for each channel:

$$var(N_{det}^{mol}) = ENF \times (N_{det}^{mol} + N_{dark} + N_{det.sol.}^{mol}) + RON \tag{29}$$

$$var(N_{det}^{par}) = ENF \times (N_{det}^{par} + N_{dark} + N_{det.sol.}^{par}) + RON \tag{30}$$

$$var(N_{det}^{tot}) = NSF^2 \times (N_{det}^{tot} + N_{dark} + N_{det.sol.}^{tot}) + RON \tag{31}$$

We draw the reader's attention to the fact that the detector's parameters in (Eq. 31) are not equal to those in (Eqs. 29, 30) and that for real calculations one has to use the values from Table 1 or similar source. The variances of molecular, particulate, and total incoming photon fluxes, which are finally used in the optical property retrievals, are estimated in accordance with the standard error propagation formulae applied to equations above:

$$var(AMB) = k_a^2 \times var(N_{det}^{mol}) + k_b^2 \times var(N_{det}^{par}) + 2 \times k_a \times k_b \times cov(N_{det}^{mol}, N_{det}^{par}) \tag{32}$$

$$var(APB) = k_c^2 \times var(N_{det}^{mol}) + k_c^2 \times var(N_{det}^{par}) + 2 \times k_c \times k_d \times cov(N_{det}^{mol}, N_{det}^{par}) \tag{33}$$

$$var(ATB) = \frac{1}{\kappa^2} \times var(N_{det}^{tot}) \tag{34}$$

where the $cov(N_{det}^{mol}, N_{det}^{par})$ represents the covariance of molecular and particulate channels. This term is required because the signals in the channels are coupled through a non-zero cross-talk.

When the variances are known, the original noise-free AMB, APB, and ATB profiles are modified by random noise, which is modulated by the standard deviation calculated from the variances, and the results are saved. If the horizontal or vertical signal averaging is involved, the noise is scaled inversely proportional to a square root of the number of samples within the averaging interval.

### 3.2.3 Useful lidar signals and their SNRs

To address the information content of the backscattered radiance, it makes sense to define a useful signal and to estimate the SNR for this signal. For CALIOP the useful signal is represented by $ATB(\lambda, z)$ (see Eq.1) whereas the ATLID can measure the molecular and particulate backscattered radiances separately, so it would be logical to call the $APB(\lambda, z)$ (see Eq. 2) a signal, which carries the information about the cloud, and look at its SNR. For the sake of simplicity, we do not discuss here the perpendicular channels of these two space lidars assuming that the backscattered depolarized radiance is detected the same

way, and adding the processing of this component to the formalism above would not change the conclusions of this work. Another aspect that we do not discuss here is the change in cloud microphysics, which can also affect the cloud detection and cloud radiative effects. We consider and model only the cloud occurrence, cloud cover, and cloud detectability.

For the simulated CALIOP signals, we estimate SR(z) at 532 nm and CF(z) according to Eqs. 6 and 7. The simulated ATLID signals are converted to equivalent 532-nm SR'(z) (see Sect. 2.2 and Feofilov et al., 2022). Then we calculate CF(z) for ATLID using the same Eqs. 6 and 7 with the same thresholds, and then we analyze the resulting cloud fraction.

To quantify the lidar cloud detection agreement and disagreement regarding the reference cloud dataset, we distinguish four cases: (1) when the lidar detects the actually cloudy layer as cloudy (YES_YES case), (2) when there is no cloud and the lidar does not detect a cloud (NO_NO), (3) when the lidar does not detect an existing cloudy layer (YES_NO or false negative), and (4) when the lidar detects a cloud whereas the layer does not contain a cloud (NO_YES or false positive). We will define their occurrence ratios as:

$$R_{YES\_YES}(z) = \frac{N_{YES\_YES}(z)}{N_{tot}(z)} \; ; \; R_{NO\_NO}(z) = \frac{N_{NO\_NO}(z)}{N_{tot}(z)} \; ; \; R_{YES\_NO}(z) = \frac{N_{YES\_NO}(z)}{N_{tot}(z)} \; ; \; R_{NO\_YES}(z) = \frac{N_{NO\_YES}(z)}{N_{tot}(z)} \quad (35)$$

The sum of all four ratios in (Eq. 35) yields unity. A perfect match between the cloud distribution in the atmosphere and the product retrieved from the measurement would be when $R_{YES\_YES}(z) + R_{NO\_NO}(z) = 1$ and $R_{YES\_NO}(z) = R_{NO\_YES}(z) = 0$.

### 3.3. Simulated ATLID and CALIPSO lidar profiles over cirrus and stratocumulus scenes

The most representative parts of pseudo orbits generated with the help of 3D_CLOUDV3 model (Section 3.3) are shown in Fig. 4 and 5 for cirrus and stratocumulus clouds, respectively. We arbitrarily split the "cloud curtain" generated from the output of this model (Sect. 3.2) to "daytime" and "nighttime" by setting the solar zenith angle (SZA) to 45° and 120°, respectively. These values are not linked with the cloud formation mechanisms in the 3D_CLOUDV3 model, they are just needed for a second half of the simulator chain (see noise-related boxes in Fig. 3). In Fig. 4ab, one can see a fine structure of modeled cirrus clouds. Looking at Fig. 4cd, one can say that the clouds are optically thin. This combination makes the detection of the clouds marked in Fig. 4ef challenging.

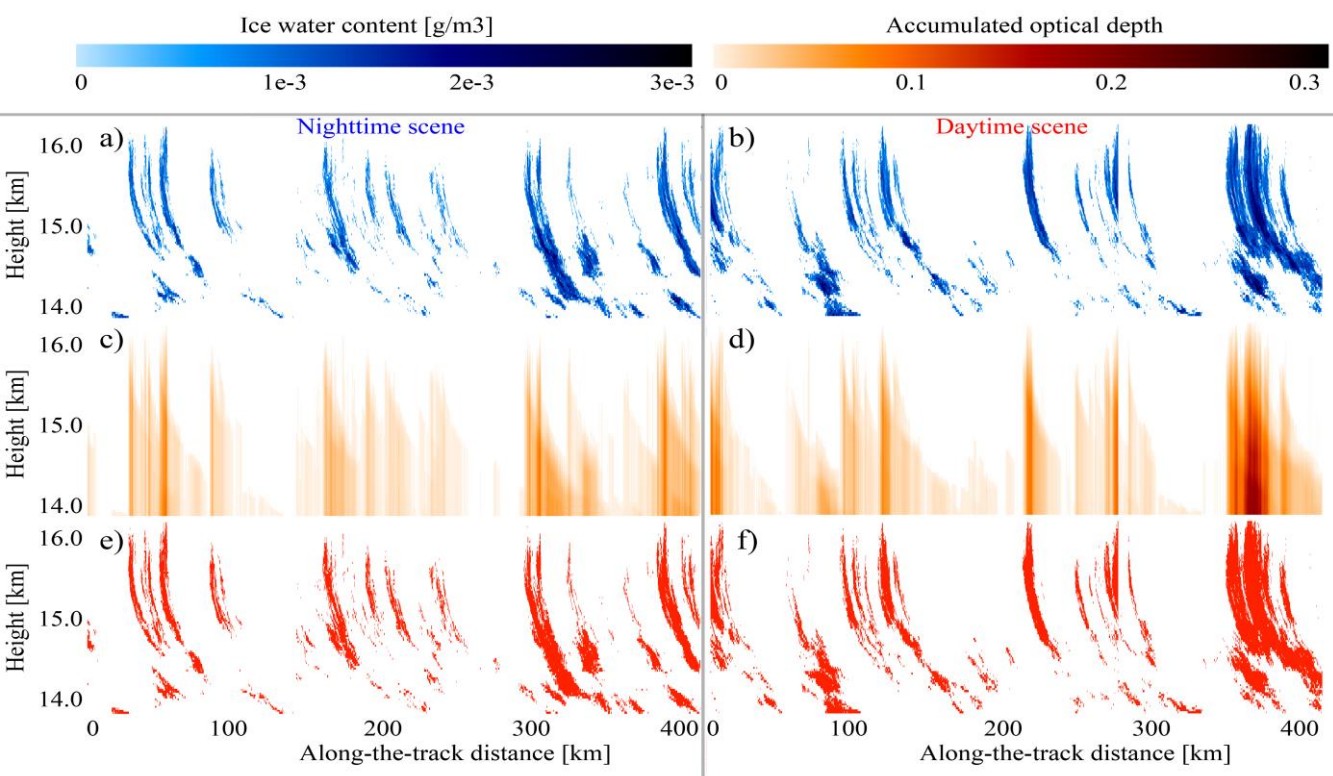

**Figure 4: Example of cirrus cloud (a) input data from 3DCLOUD model used in the simulation: Ice water content (IWC), night corresponds to one piece of orbit; (b) IWC, day corresponds to another piece of orbit; (c) accumulated optical depth starting from the cloud top, night; (d) same as (c), day; (e) cloud mask, night; (f) cloud mask, day. We set the cloud mask to 1 whenever IWC>0. The cloud masks presented here are called "reference dataset" in the rest of the paper.**


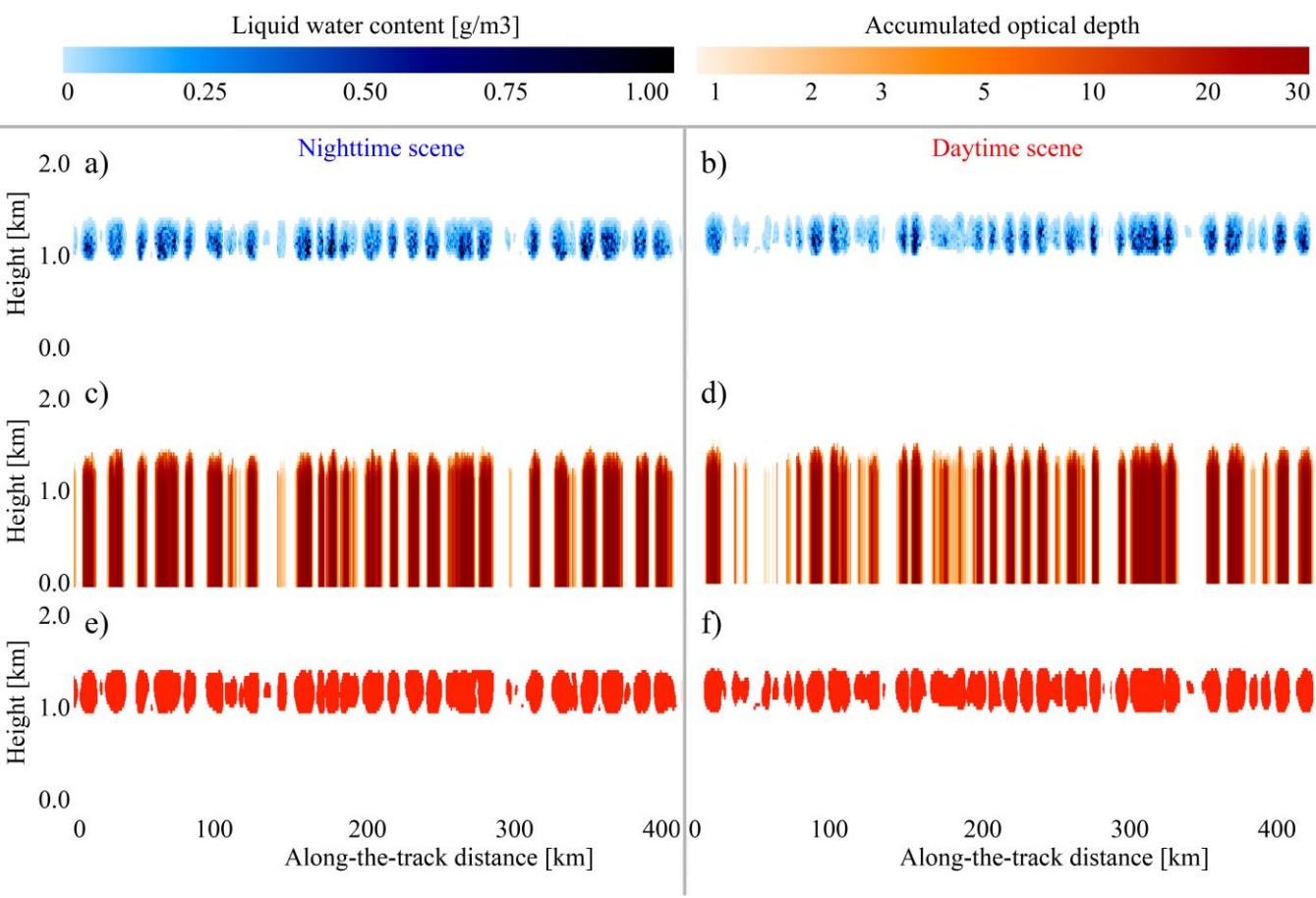

**Figure 5: Same as Fig. 4, but for stratocumulus cloud scenes. Note the color scale difference between Fig. 4 and Fig. 5.**

The stratocumulus clouds shown in Fig. 5 belong to another category of challenging observations. The clouds are closely spaced along the horizontal axis and at the same time they are optically thick: about two-thirds of the clouds have optical thickness larger than 3 (Fig. 5cd), but the scene also contains about one third of semi-transparent clouds like the ones that were reported in (Leahy et al., 2022). From the Fig. 5cd, one can conclude that at present there is no space-based measurement that can retrieve all the optical properties of cloud layers shown in Fig. 5ef. Another problem of these clouds is that their horizontal

averaging might bias the estimated cloud fraction (see e.g. Fig. 4 of Feofilov et al., 2022 and its discussion).

   In Fig. 6 and 7, we demonstrate the differences between two lidars for four scenes (cirrus/stratocumulus clouds, day/night) using the simulated backscatter signal. For the cirrus cloud scene (Fig. 6), both the $ATB(532nm, z)$ of CALIOP and the $APB(355nm, z)$ of ATLID show a detectable signal in the areas marked by a cloud mask in Fig. 6ef. But, if one defines the signal detection level as three sigmas, one will see that a part of thin clouds will be missing. This is not surprising since we

compare a "pure" modeled cloud with its noisy representation in the measuring system. What can be estimated from the image is the potential reliability of cloud detection from ATLID and CALIOP: according to the SNR values (Fig. 6gh vs Fig. 6cd), the $APB(355nm, z)$ signal from ATLID (Fig. 6ef) reaches higher SNR values than the $ATB(532nm, z)$ signal from CALIOP

(Fig. 6ab). This gives a hint that the cloud detection from this instrument might be somewhat better than from CALIOP, and that one can lower the detection threshold and still get the cloud instead of noise. This is a subject of one of the experiments described below. As for the daytime vs nighttime difference, we do not see a big change between the left-hand-side and right-hand-side panels for ATLID (Fig. 6e-h) whereas the CALIOP shows higher noise in Fig. 6bd. We note here that the calculations were performed for the cases when only a thin cirrus cloud was present in the atmospheric column, whereas the rest of it corresponded to clear sky conditions. In the real life, though, the second cloud layer beneath cirrus might increase the solar noise (see the right-hand-side panels of Fig. 7), and this will adversely affect the thin cloud detection, especially from the CALIOP measurements. This is explained by a larger field of view of CALIOP lidar (see Table 1). In our exercise, we wanted to estimate the best achievable results for a given cloud scene for each instrument and to compare the lidar performances. This way, the conclusions made below for the daytime scenes refer to the minimal differences between the two instruments.

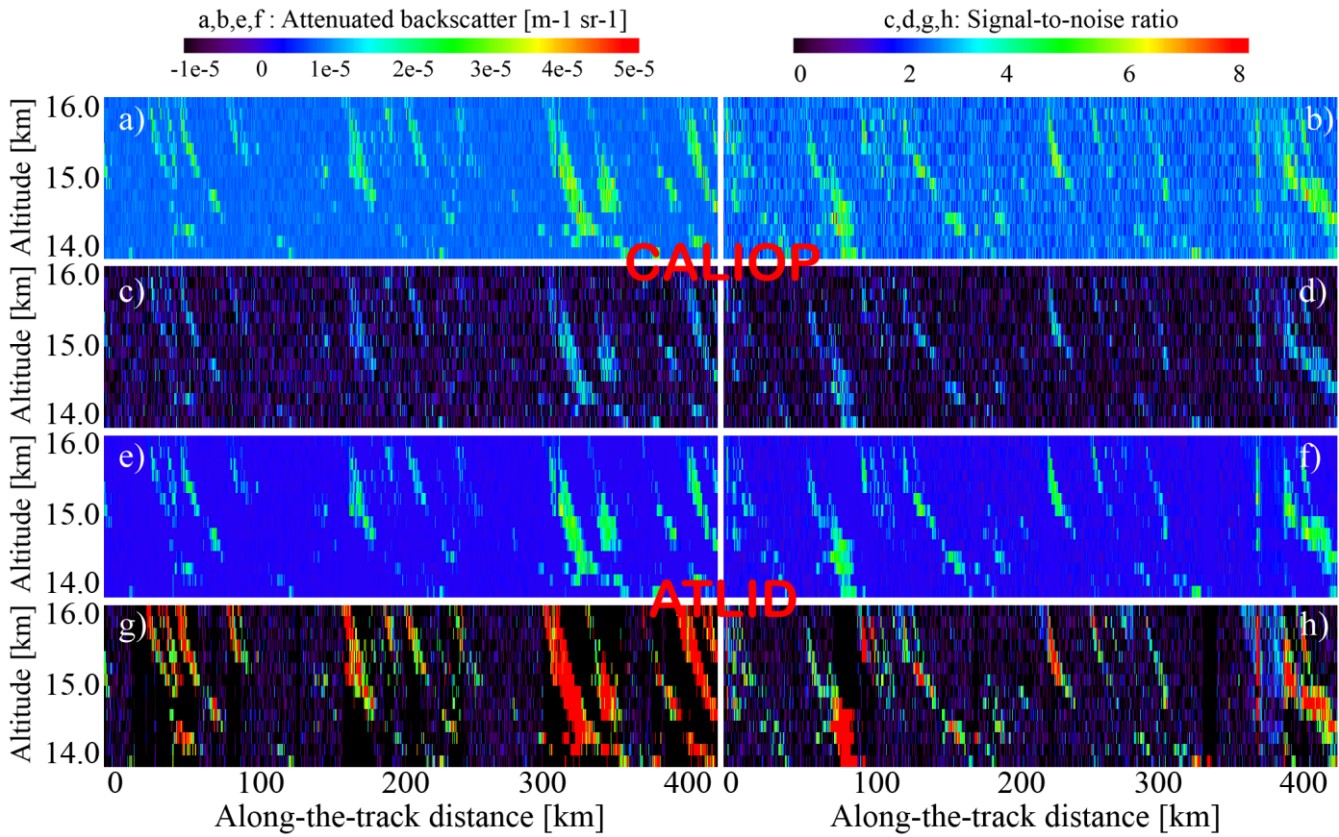

**Figure 6: Signals and signal-to-noise ratio for cirrus cloud scene. CALIOP: (a) ATB(532nm, z), night for one piece of orbit; (b) ATB(532nm, z) day for another piece of pseudo-orbit; (c) SNR, night; (d) SNR, day; ATLID: (e) APB(355nm, z), night; (f) APB(355nm, z) day; (g) SNR, night; (h) SNR, day. Note that the scene contains only these clouds and a clear sky below. For the reflective clouds beneath the cirrus layer, the daytime noise will be higher (see the right-hand-side panels of Fig. 7).**

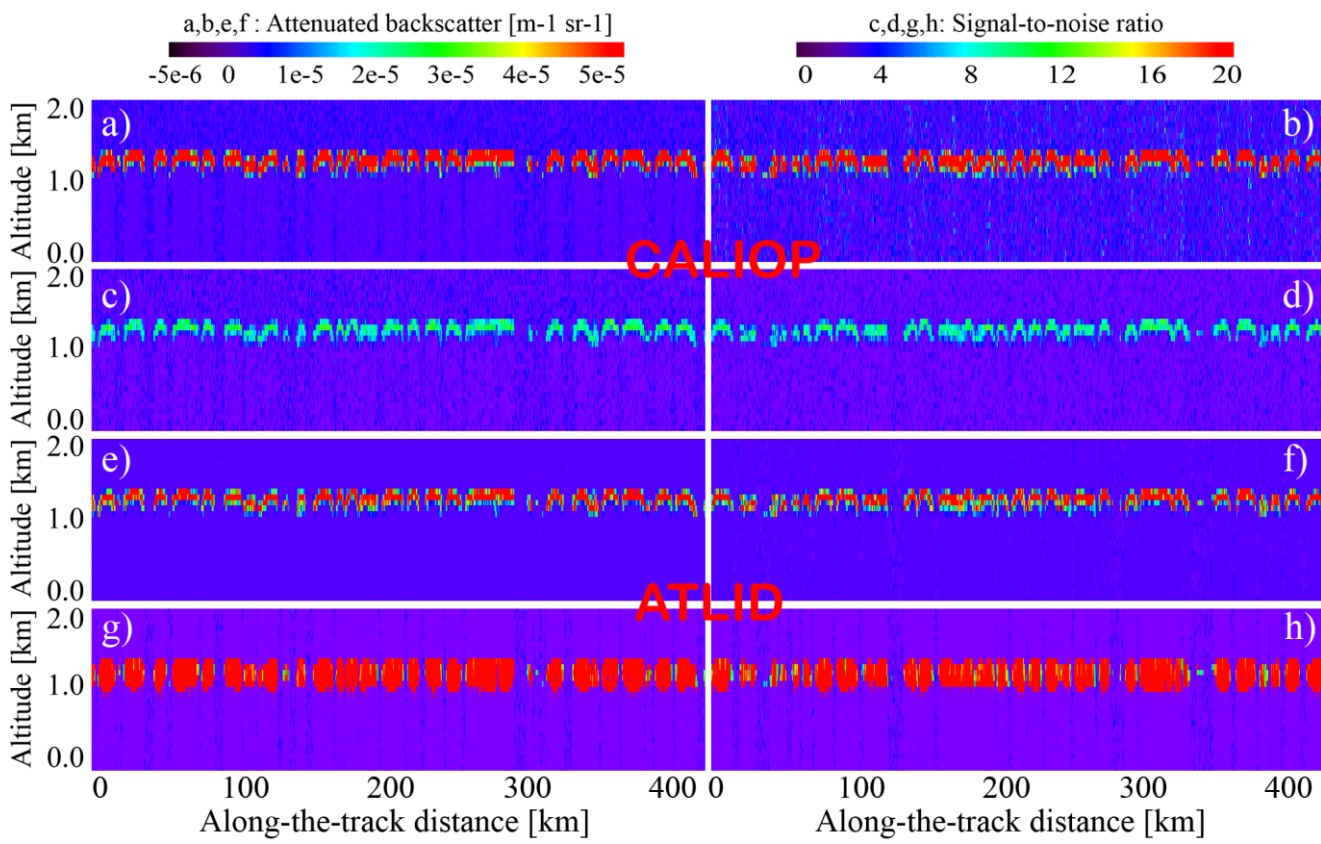

**Figure 7: Same as Fig. 6, but for stratocumulus cloud scenes.**

As for the stratocumulus clouds (Fig. 7), both the signals and SNRs are strong for both lidars, day and night. The altitudes beneath these clouds correspond to areas without useful signal: at these heights, the signal is already attenuated by a cloud above, and the attenuation is so strong that even the cloud base is not visible at optical wavelengths (e.g. Guzman et al., 2017). Another remarkable feature shown in this plot is higher daytime noise for CALIOP (Fig. 7bd). Even though this high noise level does not affect the stratocumulus cloud detection itself, it might affect the aforementioned higher-level cloud detection and, from this point of view, ATLID has an advantage over CALIOP.

Summarizing, one can say that the $ATB(532nm, z)$ signals of CALIOP and the $APB(355nm, z)$ signals of ATLID carry similar type information for the same cloud scenes, but their SNRs suggest that (a) the daytime cloud detection from ATLID should be more reliable and (b) that one can lower the detection threshold for this instrument without admixing numerous noise-triggered clouds. Let's now see how the signal quality transforms into the product quality and, in particular, to cloud detection quality.

### 3.4. Capability of ATLID to detect optically thinner clouds than CALIPSO

Here, we describe the test we performed seeking to answer whether the cloud detection limits (Eq. 6) defined in (Chepfer et al., 2010) could be lowered to detect thinner clouds. For this test, we followed the second half of the flowchart (Fig. 3) and

calculated the $SR(532nm, z)$ for CALIOP and the CALIOP-like $SR(532nm, z)$ for ATLID (Eqs. 2-3), but we changed the cloud detection thresholds of (Eq. 6) to the following ones:

$$SR(532nm, z) > 3 \text{ and } ATB(532nm, z) - ATB_{mol}(532nm, z) > 1.5 \times 10^{-6} \ m^{-1} \ sr^{-1} \qquad (36)$$

Then we estimated the cloud fractions and statistical agreement with the source cloud data (Eqs. 6,7). The threshold in the left-hand side of (Eq. 36) implies that the particulate backscatter in a layer, which we call a cloudy one, is twice the molecular one.

The threshold in the right-hand side of (Eq. 36) corresponds to the absolute values of $ATB(532nm, z)$ recalculated for $SR(532nm, z) = 3$ at the height of 8 km (Chepfer et al., 2010), but overall the rationale for selecting these very values is based on the SNR values levels we observed in the test simulations. Further lowering the threshold will lead to an increased number of false-positive cloud detections in ATLID.

    Since the "native" CALIOP profiles are averaged over 3 points above 8 km, we applied an averaging procedure over ~1 km

distance to all simulated signals and repeated the analysis. To compare apples to apples in terms of signal statistics, we averaged over 4 CALIPSO shots and over 2 effective ATLID shots, yielding the actual average over 1330 m and 1140 m, respectively. To reduce the number of plots, we do not show the instantaneous profiles without the averaging, but in Table 2 we provide the estimates for them (seek columns marked with Averaged=N).

    In Fig. 8ab, the $SR(532nm, z)$ has the same patterns as the $ATB(532nm, z)$ signals in Fig. 6ab. But, the daytime noise is more

pronounced in this presentation, partially because of the chosen color scale. However, not all the noise from Fig. 8b propagates to Fig.8d. This is because of a second condition of (Eq. 36): the variations are partially filtered out by imposing a condition on the $ATB(532nm, z)$ signals w.r.t. $ATB_{mol}(532nm, z)$. Still, the daytime scene contains a lot of false detections marked by red in Fig. 8d. The overall characteristics of CALIOP cloud detection for this scene estimated over the whole simulated cloud dataset can be found in the 2nd and 6th columns of Table 2. The bottom two lines of this table refer to the detectability of a

cloud in the whole layer: if some values of the $SR(532nm, z)$ triggered cloud detection, we calculated the cloud fraction similar to (Eq. 7) and then compared the resulting series of cloud fractions with the reference one defined from the source dataset. The "total score" line refers to the cloud detection statistics and is defined in the caption. As one can see, the strong daytime noise of CALIOP prevents the correct cloud detection, mostly due to large number of false positive cloud detections (NO_YES). The bias and the r.m.s. rows show the biggest change when passing from nighttime to daytime conditions.

The same analysis performed for ATLID (Fig. 9) shows less daytime noise (compare Fig. 9b with Fig. 8b), and the cloud detection quality for the clouds defined using (Eq. 36) is better than that of the CALIOP (compare Fig. 9d with Fig. 8d). The corresponding columns of Table 2 tell us that for ATLID the number of false detections during day and night is approximately the same, whereas for CALIOP using the Eq. 36 for the detection dramatically increases the amount of false detections during daytime. We should also stress here that the obtained result is a lower estimate because we used the scenes without underlying

clouds, which could reflect more solar radiance and further contaminate the observations. For these scenes, the difference between ATLID and CALIOP will be even larger.

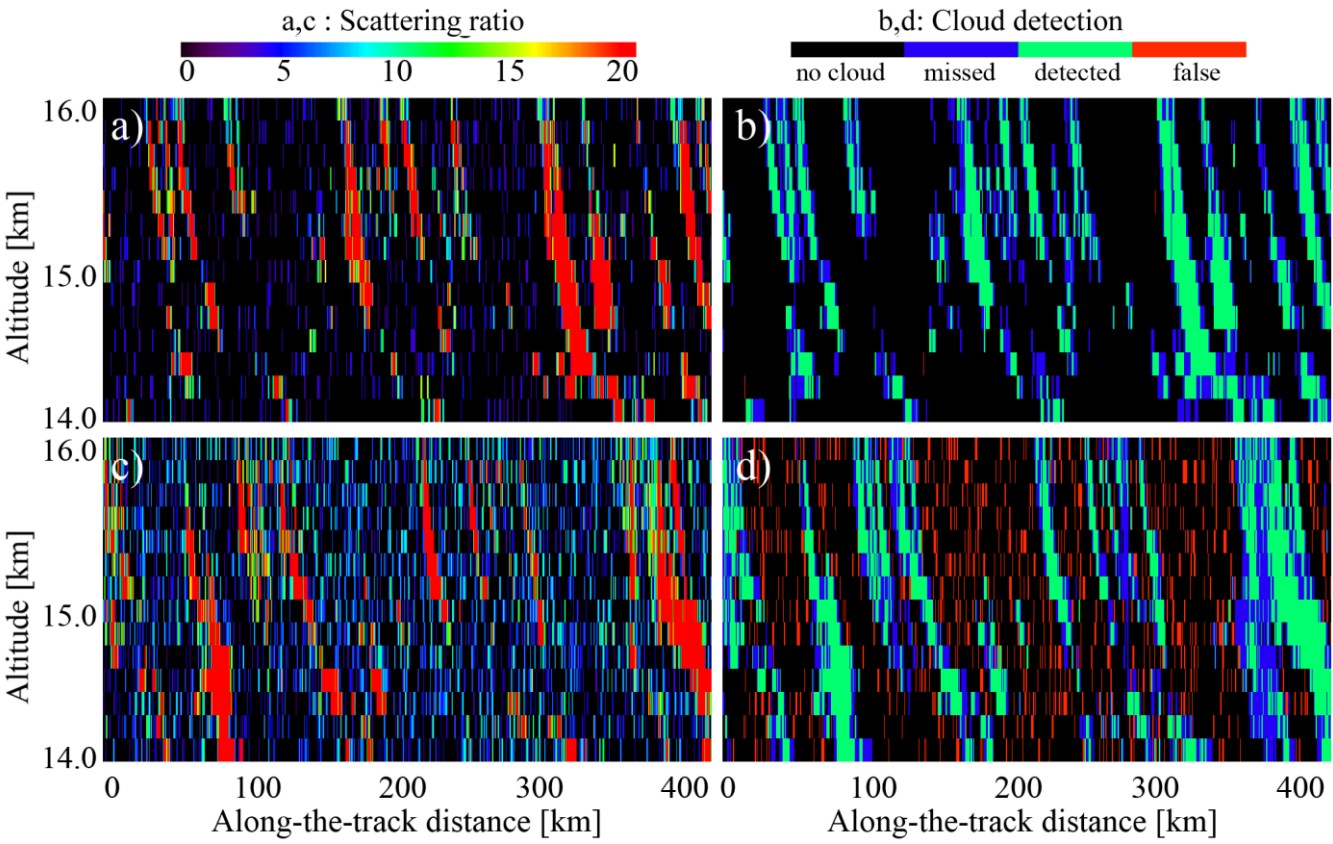

**Figure 8: Scattering ratio and cloud detection estimated for cirrus clouds observed by CALIOP using Eq. 36: (a) scattering ratio, night; (b) cloud detection, night; (c) scattering ratio, day; (d) cloud detection, day. Note the color scale difference for (ac) and (bd).**

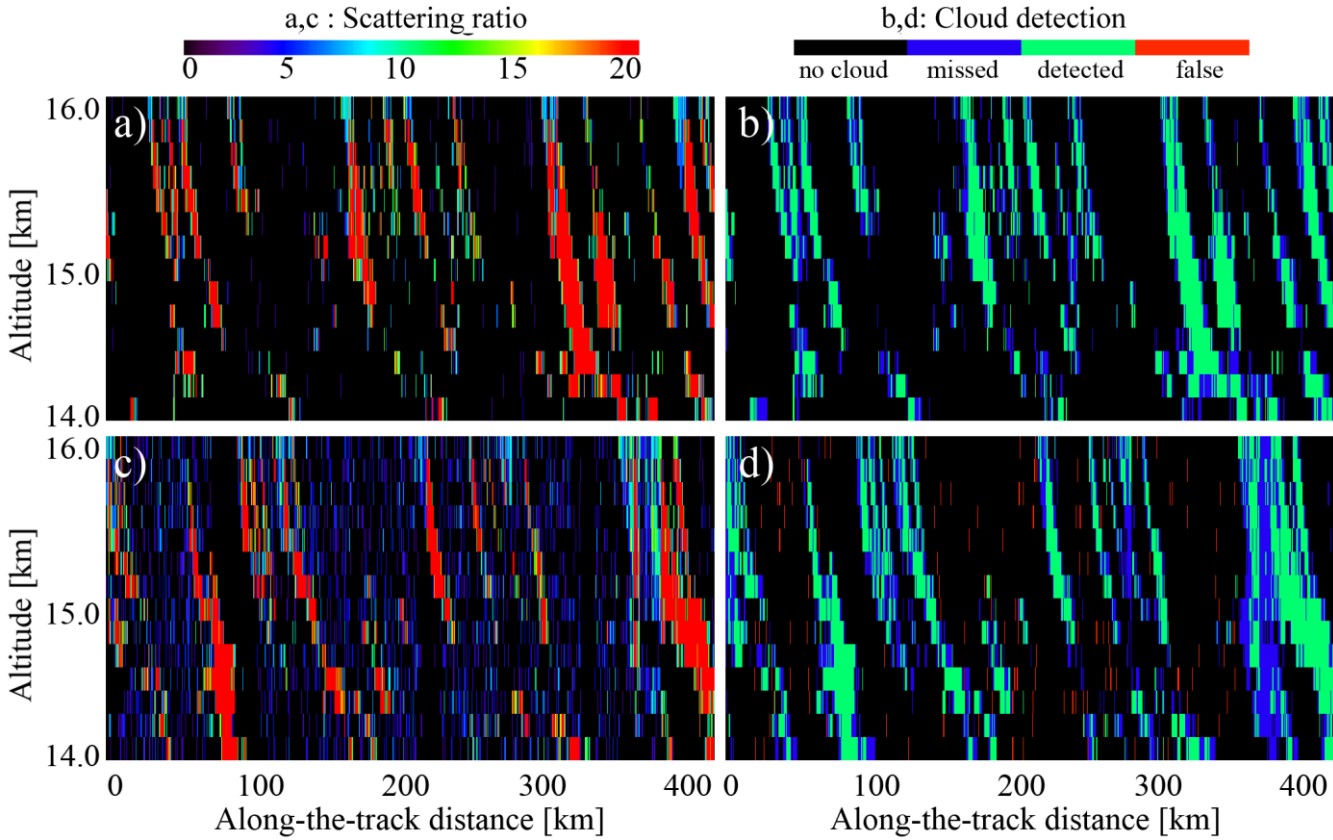


**Figure 9. Same as Fig. 8, but for ATLID**

| Lidar | CALIOP | | | | | | | | ATLID | | | | | | | |
|---|---|---|---|---|---|---|---|---|---|---|---|---|---|---|---|---|
| Day/Night | Night | | | | Day | | | | Night | | | | Day | | | |
| Cloud | Ci | | Sc | | Ci | | Sc | | Ci | | Sc | | Ci | | Sc | |
| Averaged | N | Y | N | Y | N | Y | N | Y | N | Y | N | Y | N | Y | N | Y |
| YES_YES | 7 | 7 | 8 | 8 | 8 | 7 | 10 | 9 | 8 | 8 | 8 | 8 | 7 | 7 | 8 | 8 |
| NO_NO | 80 | 78 | 82 | 86 | 69 | 72 | 68 | 80 | 83 | 81 | 86 | 86 | 78 | 78 | 86 | 86 |
| YES_NO | 10 | 14 | 6 | 6 | 13 | 18 | 4 | 5 | 9 | 11 | 6 | 6 | 13 | 15 | 5 | 5 |
| NO_YES | 2 | 0 | 4 | 0 | 10 | 3 | 18 | 6 | 0 | 0 | 0 | 0 | 2 | 0 | 0 | 0 |
| Tot. score | 85 | 85 | 88 | 93 | 56 | 71 | 69 | 87 | 91 | 89 | 94 | 93 | 81 | 83 | 94 | 94 |
| Bias | 11 | -4 | -1 | -6 | 21 | 11 | 16 | 2 | -5 | -8 | -6 | -6 | 8 | 2 | -5 | -5 |
| R.m.s. | 18 | 13 | 8 | 6 | 23 | 21 | 9 | 6 | 10 | 9 | 6 | 6 | 21 | 18 | 6 | 5 |

Table 2: Cloud detection statistics for CALIOP and ATLID when the cloud definition corresponds to as $SR(532nm, z) > 3$ and $ATB(\lambda, z) - ATB_{mol}(\lambda, z) > 1.5 \times 10^{-6}\ m^{-1}\ sr^{-1}$ (Eq. 36). The bias and r.m.s. values are defined for the clouds detected in the columns (see text) and we define the total score in % as $100\% \times (1 - (YES\_NO + NO\_YES)/(YES\_YES + NO\_NO))$.

The same type plots built for the stratocumulus clouds (Fig. A1 and A2 in the Appendix A for CALIOP and ATLID, respectively) show a different picture. Strong signals and large SNRs shown in Fig. 5 help to unambiguously identify the cloud. Large fraction of underestimated clouds shown in blue in Fig. A1cd and Fig. A2cd corresponds to cloud parts below the opaque cloud top layer, which are not accessible for the instruments observing the scene from above. As with cirrus clouds, the false detections rate is higher for CALIOP during daytime.

For CALIOP, the 1 km averages reduce the number of false detections and improve the total score for daytime simulations for cirrus. For ATLID with its lower daytime noise, the averaging procedure does not change the cloud detection quality that much. For the stratocumulus clouds, the averaging procedure is not required for ATLID since sometimes it can lead to overestimate the cloud fraction (e.g. Chepfer et al. 2008, Feofilov et al. 2022). For CALIOP, it improves the score because of suppression of sporadic-noise-induced "clouds" above the real cloud layer (Fig. A1d).

Overall, the ATLID-related columns in Table 2 demonstrate more consistency between daytime and nighttime cloud amounts and reference data than the CALIOP-related ones, and ATLID daytime cloud quality is better than that of CALIOP whereas the nighttime results are comparable. Our tests show that if the CALIOP-like solar filter were used in ATLID, one could lower the thresholds of Eq. 36 down to $SR(532nm, z) > 2$ and $ATB(532nm, z) - ATB_{mol}(532nm, z) > 1.0 \times 10^{-6}\ m^{-1}\ sr^{-1}$ without losing the quality of cloud retrievals, whereas the same thresholds applied to CALIOP would give completely unacceptable results for daytime conditions.

Of course, the examples considered in this section do not cover the whole range of high-, middle-, and low-level clouds, but they draw a line between the threshold values that can be used for cloud definition for CALIOP and ATLID and show that the

difference is linked to noise characteristics of the instruments. This result suggests that ATLID should be able to observe optically thinner clouds than CALIOP in daytime at full horizontal resolution.

To illustrate this point, we used the available data set for cirrus and estimated the minimal detectable backscatter (MDB) for ATLID in terms of equivalent $ATB(532nm, z)$ for comparison with CALIOP values obtained for 5 km horizontal averaging of cirrus measured at 15 km height (McGill et al., 2007). For this numerical experiment, we used noisy $APB$ and noise-free

$AMB$ to keep the consistency with our approach of cloud detection using only one noisy component, the particulate one. For this horizontal averaging, we obtained MDB=$3.0\pm1.0$x$10^{-7}$ m$^{-1}$ sr$^{-1}$ for the nighttime and MDB=$4.0\pm1.0$x$10^{-7}$ m$^{-1}$ sr$^{-1}$ for the daytime in equivalent $ATB(532nm)$ values, whereas for CALIOP we obtained MDB=$4.0\pm2.0$x$10^{-7}$ m$^{-1}$ sr$^{-1}$ for the nighttime and MDB=$1.3\pm0.2$x$10^{-6}$ m$^{-1}$ sr$^{-1}$ for the daytime in its native $ATB(532nm)$. The daytime value estimated for CALIOP is in good agreement with the measured one (McGill et al., 2007) whereas the estimated nighttime value is somewhat lower than

the measured MDB=$8.0\pm1.0$x$10^{-7}$ m$^{-1}$ sr$^{-1}$. From this comparison, we cannot conclude that the ATLID will provide better sensitivity to thin clouds during nighttime, but we can conclude that its daytime thin cloud detection at 5 km averaging capacity should be comparable to that of CALIOP for the nighttime, and this will be an important achievement for daytime vs nighttime cloud comparison. Using the cloud detection thresholds defined by Eq. 36 and refined for the real data flow using the methodology outlined above, the CLIMP short-term product will be produced.

**4. The CLIMP long-term dataset**

**4.1. Capability of CLIMP and CALIPSO-GOCCP to detect the same clouds**

One of the overarching goals of our study is to develop a method for merging the data from several space borne lidars into a continuous cloud record to detect long-term changes and get a seamless cloud climatology. Since the low threshold tested in

the previous section revealed the sensitivity mismatch between the two instruments, we had to test whether the cloud detection thresholds developed for CALIOP (Chepfer et al., 2010) are applicable to ATLID, and whether the clouds retrieved using these thresholds are consistent between the two lidars. For this exercise, we followed the same scheme as in the previous section, but this time the clouds were defined in Eq. 6 as in Chepfer et al. (2010) and the follow-up works (e.g. Cesana et al., 2019; Guzman et al. 2017).

Figure 10 demonstrates the daytime and nighttime scattering ratios above the detection thresholds (Eq. 6) and the corresponding cirrus cloud detection statistics for CALIOP. The $SR(532nm, z)$ in Fig. 10ab demonstrates the same patterns as the $ATB(532nm, z)$ signals in Fig. 6ab. As expected, this time the daytime noise is less pronounced (compare Fig. 10b to Fig. 8b). Still, the daytime scene contains a certain number of false detections marked by red in Fig. 10d. The same analysis performed for ATLID (Fig. 11) also shows somewhat less noise in daytime (compare Fig. 11b with Fig. 9b). The cloud

detection quality of ATLID is like that of the CALIOP (see Table 3). In this setup, the ATLID is just slightly better than CALIOP with its somewhat higher rate of false detections during the day (compare the "c" and "d" panels of Fig. 8, 9, 10, and 11 and the corresponding columns in Table 3). For stratocumulus clouds (Fig. 1 and Fig. 5), with their strong signals, the

agreement between CALIOP and ATLID is also better than for the clouds defined by Eq. 36 (compare the "c" and "d" panels of Fig. A1, A2, A3, and A4). The 1 km averaging further improves the agreement between the data sets (Table 3).

Summarizing, using the thresholds (Eq. 6) to define the clouds makes the cloud data sets from CALIOP and ATLID comparable. Further adjustment will be needed for real ATLID data to compensate the effects of diurnal cycle (Noel et al., 2018; Chepfer et al., 2019; Feofilov and Stubenrauch, 2019). Other compensations might be required when the real ATLID data become available. Since there is a high chance that there will be no overlapping period for these two satellite instruments, an intercalibration procedure will be required. For this, one can use the average cloud amount for low, middle, and high clouds

in different zones (tropics, mid-latitudes, and polar) to track the changes and to introduce feedback to cloud detection algorithm. This way, the number of cases measured for each zone will be high, and the uncertainty will be low. The daytime and nighttime observations should be considered separately to address the diurnal cycle and daytime noise issues. In the sections below, we assume that the intercalibration has been performed and that the cloud datasets agree.

| Lidar | CALIOP | | | | | | | | ATLID | | | | | | | |
|---|---|---|---|---|---|---|---|---|---|---|---|---|---|---|---|---|
| Day/Night | Night | | | | Day | | | | Night | | | | Day | | | |
| Cloud | Ci | | Sc | | Ci | | Sc | | Ci | | Sc | | Ci | | Sc | |
| Averaged | N | Y | N | Y | N | Y | N | Y | N | Y | N | Y | N | Y | N | Y |
| YES_YES | 7 | 6 | 7 | 8 | 7 | 6 | 9 | 8 | 7 | 6 | 7 | 7 | 6 | 6 | 7 | 8 |
| NO_NO | 82 | 77 | 86 | 86 | 72 | 73 | 79 | 85 | 82 | 80 | 86 | 86 | 79 | 77 | 86 | 86 |
| YES_NO | 11 | 17 | 6 | 7 | 15 | 20 | 5 | 6 | 10 | 14 | 7 | 7 | 15 | 18 | 6 | 6 |
| NO_YES | 0 | 0 | 0 | 0 | 6 | 1 | 7 | 1 | 0 | 0 | 0 | 0 | 0 | 0 | 0 | 0 |
| Tot. score | 88 | 81 | 93 | 93 | 67 | 75 | 85 | 93 | 89 | 85 | 93 | 93 | 83 | 80 | 94 | 94 |
| Bias | -2 | -12 | -6 | -7 | 13 | -2 | 3 | -5 | -8 | -9 | -7 | -7 | -4 | -10 | -6 | -6 |
| R.m.s. | 14 | 11 | 6 | 6 | 22 | 20 | 7 | 5 | 9 | 9 | 6 | 6 | 18 | 17 | 6 | 5 |

Table 3: Cloud detection statistics for CALIOP and ATLID in the case when the cloud is defined as $SR(532nm, z) > 5$ and $ATB(\lambda, z) - ATB_{mol}(\lambda, z) > 2.5 \times 10^{-6} \, m^{-1} \, sr^{-1}$ (Eq. 6).

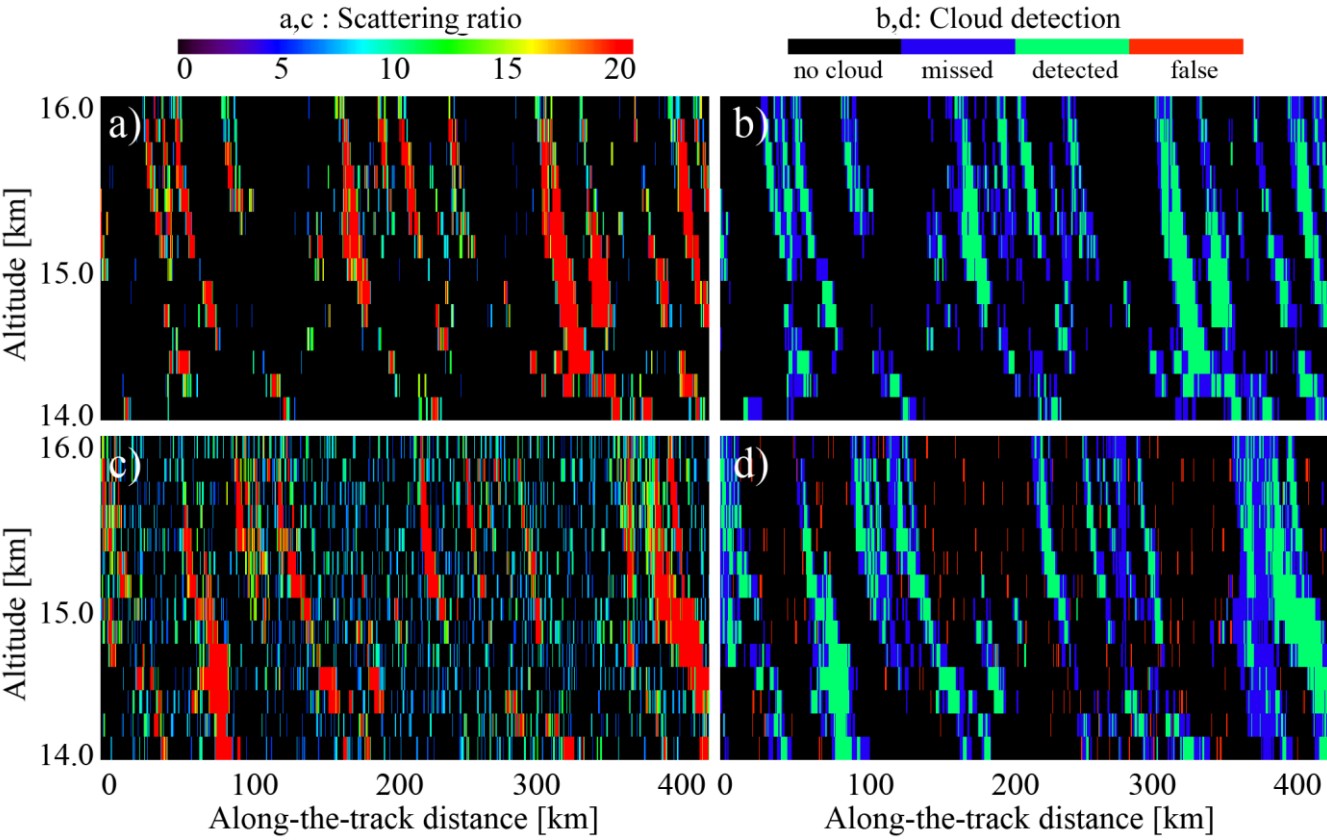

**Figure 10: Scattering ratio and cloud detection statistics estimated for cirrus clouds observed by CALIOP using Eq. 6: (a) scattering ratio, night; (b) cloud detection, night; (c) scattering ratio, day; (d) cloud detection, day. Note the color scale difference for (ac) and (bd).**

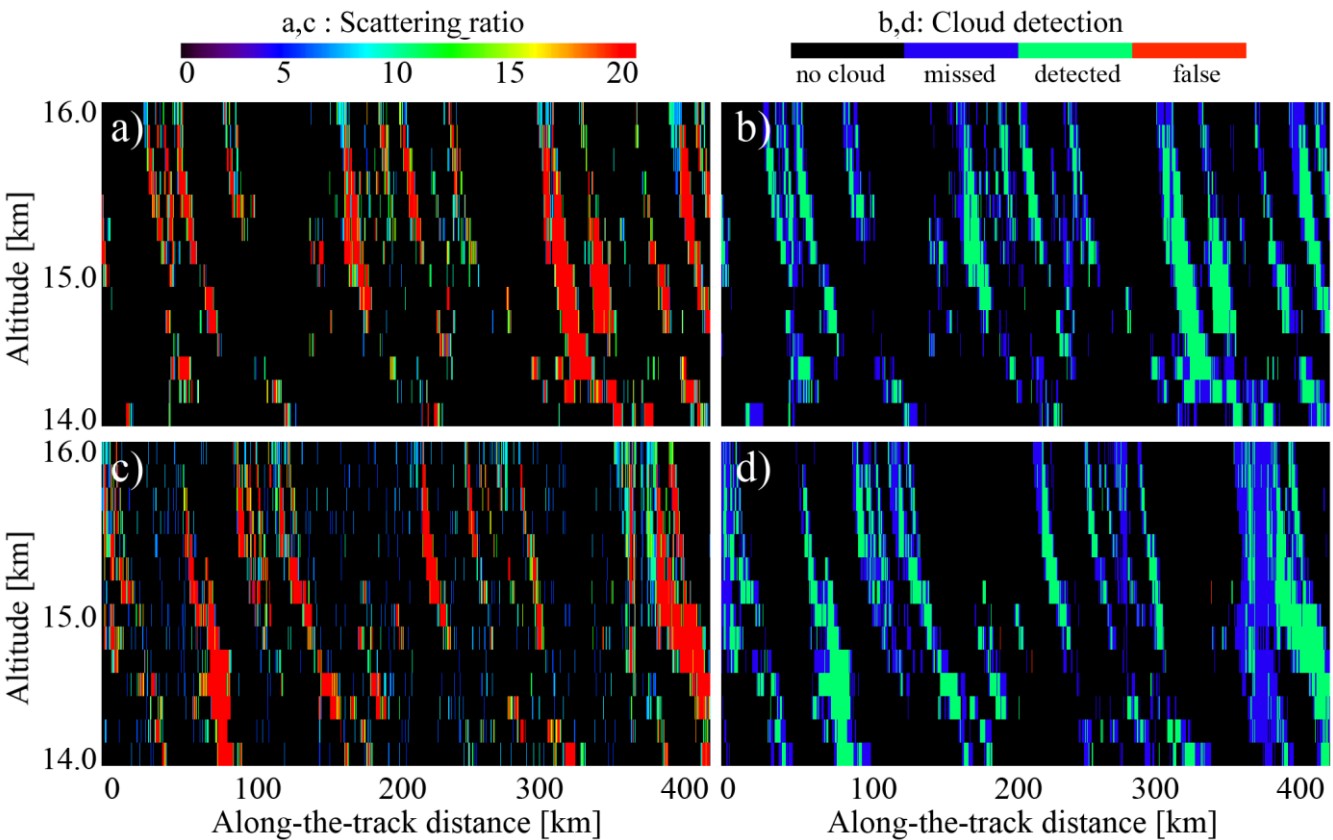

**Figure 11: Same as figure 10, but for ATLID**


## 4.2 Numerical chain to simulate long-term lidar record and method to quantify time of emergence

The previous section shows that ATLID and CALIOP data may be merged to build a long-term dataset, even though their instrumental or orbital differences might necessitate further reconciliation. Here we suppose perfect reconciliation will eventually be reached, and we build a long-term space lidar synthetic dataset spanning more than 30 years, to examine when a change in cloud properties attributable to human-induced warming would be detectable in the lidar cloud record according to climate model simulations. This approach is directly inspired by the one pioneered in (Chepfer et al. 2018), and later expanded in Perpina et al. (2021).

We use climate predictions from IPSL-CM6 (Boucher et al. 2020) and CESM2 (Community Earth System Model, Hurrell et al. 2013), two ocean-atmosphere-coupled GCMs which took part in the Climate Model Intercomparison Project (CMIP) phase 6 (Eyring et al., 2016). We use predictions that start in 2008 and end in 2034, and follow the RCP8.5 scenario, which tracks the observed $CO_2$ emissions closely (Schwalm et al., 2020). Predictions are provided as monthly grids with spatial resolutions of 1.27°x2.5° on 79 vertical levels (IPSL-CM6) and 1.25°x0.94° on 40 vertical levels (CESM). On these predictions of atmospheric conditions, we apply the COSP1.4 lidar simulator (Sect. 3.2), which generates on similar spatial grids the monthly-averaged cloud properties that would be observed by a spaceborne lidar flying over the simulated atmosphere. In addition to the simulation steps described in Sect. 3.2, here as a first step of the simulation, for each grid box of the GCM-created atmosphere an ensemble of subgrid-scale profiles are stochastically generated by the Subgrid Cloud Overlap Profile Sampler (Klein and Jakob, 1999). Each of these profiles is fed to the COSP simulator, which generates a synthetic lidar profile, on which cloud detection is performed. All subgrid-scale cloud detection profiles are eventually averaged to generate a single vertical profile for each grid box (see Chepfer et al., 2008 for details).

From the synthetic cloud properties, we considered two climate diagnostics whose trend should be related to climate change: first the fraction of opaque clouds $C_{opaque}$, defined as the number of lidar profiles in which an opaque cloud is detected in a given lat/lon grid box, divided by the total number of profiles sampled in the same grid box. Opaque clouds are responsible for the majority of cloud radiative effect in the Tropics (Vaillant de Guélis et al., 2017) and the cloud amount has been identified as one of the main drivers of cloud feedbacks on climate (Zelinka et al., 2016), thus the fraction of opaque clouds should be closely tied to climate change. Second, we considered the altitude of full attenuation $Z_{opaque}$ (Guzman et al. 2017), averaged over all opaque profiles in every grid box. The vertical distribution of clouds is closely linked to their longwave radiative impact and to climate change (Vaillant de Guelis et al., 2018), and their altitude is expected to increase by several hundred meters per century (Richardson et al., 2022). Altitude is among the cloud properties whose change is expected to be detectable the earliest using active remote sensing (Chepfer et al., 2014; Takahashi et al., 2019; Aerenson et al., 2022).

From the GCM predictions, the COSP lidar simulator generates monthly grids of $C_{opaque}$ and $Z_{opaque}$, that we spatially average over the Tropics (30°S-30°N) to get monthly time series. We deseasonalize those time series to get their monthly anomalies over the 2008−2034 period. For any time $t$ along these time series, the record length is equivalent to the period between 2008-01-01 and $t$, and we computed the trend $w(t)$ as the linear regression of the time series of anomalies over that period. The

uncertainty $\sigma_w(t)$ in the trend $w(t)$ at a time $t$ was computed, as in Chepfer et al. (2018), as $\sigma_w(t) = \sigma_N \sqrt{\frac{1+\varphi}{1-\varphi}} n^{-\frac{3}{2}}$, with n the

number of years of the record at time t, $\varphi$ the lag-1 autocorrelation coefficient of the series between 0 and t, and $\sigma_N$ the standard deviation of the noise remaining in the series between 0 and $t$ once it has been deseasonalized and the auto-correlated part removed.

The following analysis focuses on the tropical regions (30°S-30°N), where the atmospheric circulation will be impacted by the weakening of the Hadley and Walker circulations expected in the upcoming century by most climate predictions (Davis and
Rosenlof, 2012; Su et al., 2014; Kjellsson, 2015; Chemke, 2021). These changes will have important effects on the spatial distribution of tropical clouds (Su et al., 2014), which provide the basis for our climate diagnostics. Cloud opacity is one of the cloud properties most closely linked to their radiative impact (Zelinka et al. 2012), which explains why our diagnostics are based on the properties of opaque clouds (as in Perpina et al. 2021). The results below assume it will be possible to process ATLID measurements in such a way that CLIMP and GOCCP cloud properties are consistent.

**4.3. How many years of ATLID observation are required in addition to CALIPSO to evaluate climate model prediction of cloud changes?**

Figure 12 shows how the uncertainty in the retrieved trend for $C_{opaque}$ changes with the length of the record of lidar-based cloud properties, starting in 2008, according to predictions from IPSL-CM6 (blue) and CESM1 (orange). The uncertainty is generally the largest and fluctuates most when the record is short, and decreases and stabilizes as the record gets longer. At any time $t$,
if we require a 95% confidence level in the prediction and assume trends are normally distributed, the real trend will lie in the as $w(t) \pm 2\sigma_w(t)$ interval. The sign of the trend will be robust once $\left|\frac{w(t)}{\sigma_w(t)}\right| > 2$. This is when the uncertainty of the trend becomes small compared to the trend itself, and marks the time of emergence of cloud change induced by anthropogenic warming. This occurs earlier for strong, stable trends, and might never occur for very small trends or trends whose sign changes over time. Times of emergence in the $C_{opaque}$ time series are indicated in Fig. 12 with triangles for three confidence levels (50,
70 and 95%). Reaching a reliable sign requires a longer record if the required confidence level is strong.

According to predictions from IPSL-CM6 (blue), a reliable trend should emerge from the natural variability at a 50 to 70% confidence level between 2030 and 2032. In other words, IPSL-CM6 predicts that revealing a reliable long-term trend in the fraction of opaque clouds would require an uninterrupted spaceborne lidar record of 22 years, which would be achievable if EarthCARE operates for at least 7 years. Reaching 95% confidence levels on the retrieved trend would require extending the
record beyond 25 years, most probably through another spaceborne lidar mission further in time. CESM1, meanwhile, predicts that a reliable long-term trend in the fraction of opaque clouds (at similar confidence levels between 50 and 70%) would be reached between 2025 and 2027, requiring 2 to 4 years of EarthCARE operation. A highly reliable trend (95% confidence levels) would be detectable in 2029, after 6 years of EarthCARE operation. In summary, if a 50% confidence level is acceptable, detecting a reliable trend would either be possible within the EarthCARE nominal operation timeframe (2 years

after launch), according to CESM1, or would require EarthCARE to operate 4 years beyond its planned lifetime, according to IPSL-CM6.

If we consider the $Z_{opaque}$ diagnostic (Fig. 13), the IPSL-CM6 model now predicts a trend will be detectable at high 95% confidence levels in 2024, i.e. one year into EarthCARE's nominal operation period. Meanwhile, according to CESM1 predictions, detecting a reliable trend (even at a modest 50% confidence level) would require EarthCARE operating for eight

years, 5 years beyond its nominal operation timeframe. This very fast detection of a reliable $Z_{opaque}$ trend predicted by IPSL-CM6 is consistent with how this model expects important and fast changes in the vertical distribution of opaque clouds in the Tropics (Perpina et al. 2021).

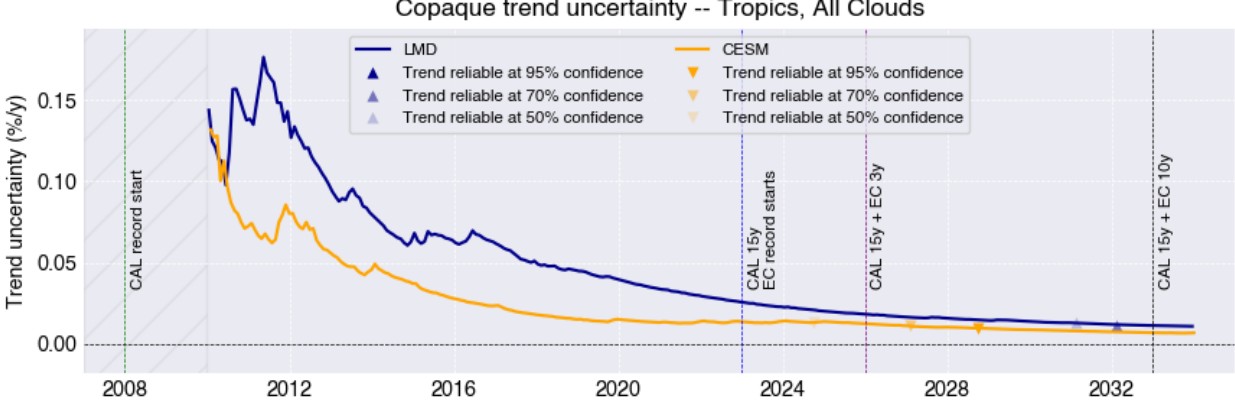

**Figure 12: Evolution of the uncertainty in the C$_{opaque}$ trend as a function of the length of the spaceborne lidar record, according to**
**atmospheric conditions predicted by IPSL-CM6 (blue) and CESM (orange) in the period between 2008 and 2034 following the RCP85 scenario. The first two years of the record (2008-2010) are considered in the analysis, but trend uncertainties during that period are very large, and are masked in the figure to improve the legibility of later years. CALIPSO's planned end of operation (2023) is marked by a vertical blue line. Supposing EarthCARE begins operation right afterward, its nominal 3-years operation point is marked by a vertical purple line, and an optimistic 10-years operation point is marked by a vertical black line.**


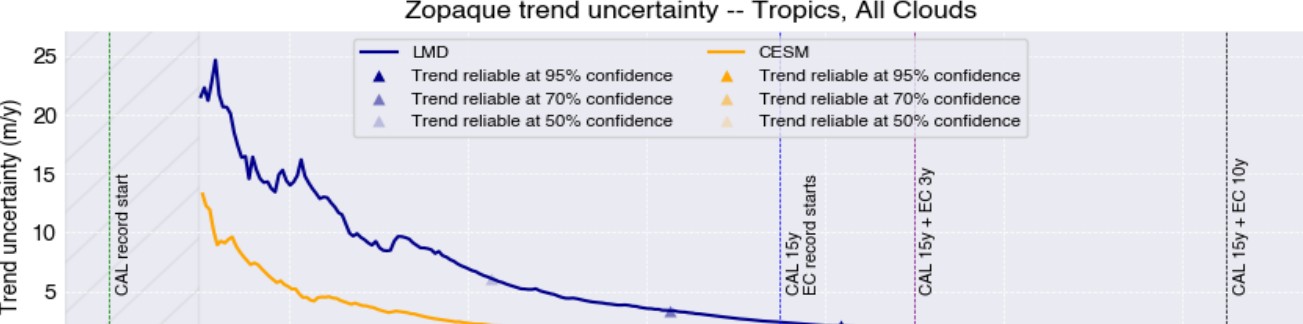

**Figure 13 : Same as Fig. 12, but for the altitude of opacity $Z_{opaque}$ instead of $C_{opaque}$.**

We sum these results up in Table 4, which in addition provides similar record lengths to detect reliable trends when considering grid boxes dominated by either low or high clouds. Tropical low opaque clouds include sparse shallow cumulus (Konsta et al., 2012) and optically thicker stratocumulus along the West coasts of continents (Guzman et al., 2017), both confined to the boundary layer and most frequent in subsidence regions. By contrast, tropical high opaque clouds are more localized and strongly correlated with deep convection. Since both kinds of clouds are driven by very different processes, it is not unreasonable to assume they will probably evolve differently in the upcoming century, which justifies their separate studies. In practice, we identified grid boxes dominated by low clouds as those where $Z_{opaque}$ was below 3 km, and high-cloud grid boxes as those where $Z_{opaque}$ was above 3 km. The results, shown in Table 4, suggest that the nominal ATLID/EarthCARE operation will be enough to validate or invalidate the trends in opaque tropical low clouds predicted by CESM. It will be possible to validate or invalidate others model-based cloud predictions only if EarthCARE performs beyond its nominal lifetime (which is not impossible, as CALIPSO demonstrated), or if measurements from a follow-up spaceborne lidar mission after ATLID are included in the cloud profile record. These results are consistent with the trends, uncertainties and times of emergence found when conducting a relatively simpler comparison of HadGEM2-A predictions in current vs +4K conditions (Chepfer et al., 2018). Needless to say, that the treatment of any follow-up mission (like AOS or Aeolus-2) will require the compensation for all the differences between the lidars like it is done in this work.

| | $C_{opaque}$ | | $Z_{opaque}$ | |
| --- | --- | --- | --- | --- |
| | IPSL-CM6 | CESM | IPSL-CM6 | CESM |
| All clouds | 2030 (7 years) | 2027 (4 years) | 2021 | 2034 (11 years) |
| Low clouds only (<3 km) | No trend | 2024 (1 year) | No trend | 2025 (2 years) |
| High clouds only (>3 km) | 2027 (4 years) | 2031 (8 years) | 2018 | No trend |

Table 4: When will a spaceborne lidar record starting in 2008 be long enough to enable a reliable detection (at 70% confidence level) of $C_{opaque}$ or $Z_{opaque}$ trends according to predictions from IPSL-CM6 or CESM. The required years of EarthCARE operation are shown in parentheses, supposing they begin in 2023. The monthly evolution of the trend uncertainties for low and high clouds are provided in Figures B1 and B2 of the Appendix B.

As stated upfront, these results depend on rather strong hypotheses of perfect continuity and perfect intercalibration between the consecutive spaceborne lidars that provide the measurements from which the cloud properties are derived. Imperfect continuity would occur if, for instance, EarthCARE starts operation later than CALIOP stops. The missing years in the record would delay the detection of a reliable trend by at least the same time period (Chepfer et al., 2018). Perfect intercalibration supposes the effects of instrumental differences in technical specifications (wavelengths, pulse energy, field of view, etc.) and orbital characteristics (local time of overpass, altitude) on lidar measurements are reconciled somehow. For instance, ATLID operates at 355nm and CALIOP at 532nm, and the impact this has on measurements can be reconciled by converting ATLID signal at 532nm as done in the current study, but the costs of this conversion are not completely understood and will require

re-examination when actual ATLID data will be available. Imperfect intercalibration could lead to offsets in one spaceborne lidar's record compared to the other, and would increase the uncertainties of the retrieved trends. Increased delays between the operation of both instruments would complicate their intercalibration. The different local times of overpass (01:30/13:30 local Solar time, LST, for CALIPSO, 06:00/18:00 LST for EarthCARE) are also quite problematic, since each instrument will sample clouds at a different phase of their diurnal cycle (Noel et al., 2018; Chepfer et al., 2019; Feofilov and Stubenrauch,

2019). In particular, this will impact high clouds related to deep convection that exhibit a marked diurnal cycle. It is out of the scope of the present work to evaluate how this change could bias the retrieved long-term trends. The same applies to a follow-up lidar mission, which may or may not operate at the same orbit with the same LST of overpass and may or may not measure the depolarized backscatter.

Finally, the times of emergence presented here must not be understood as definite but as predictions by climate models. It is

worth noting, for instance, that, according to predictions from IPSL-CM6, a reliable trend should already be readily detectable in the existing record of $Z_{opaque}$ that is today only built on CALIOP/CALIPSO (Table 4). Such a trend has not been identified yet. This is consistent with the fact that in current climate conditions IPSL-CM6 overestimates the altitude of opaque clouds in tropical convective regions, and brings them significantly higher (+2 km) near the end of 21st century (Perpina et al., 2021). Such rapid changes are not present in CESM predictions. These important model differences highlight the crucial need for

continued long-term cloud lidar observations able to monitor the actual cloud changes, and disambiguate model predictions.

## 5. Conclusions

This study presents the physical basis for the ATLID Cloud CLIMate Product named CLIMP. This product builds on previous work on CALIPSO, a space lidar dedicated to cloud and aerosol observations like ATLID. CALIPSO data have been being used for 16 years to evaluate the description of clouds in climate models using a dedicated product named GOCCP and a

dedicated lidar simulator named COSP/lidar. The present work also builds on recent work on AEOLUS, a space lidar with HSRL capability operating in the UV, like ATLID. Based on this legacy, we have defined the CLIMP short-term (ST) and CLIMP long-term (LT) products, both dedicated to cloud climate studies. Both contain the same variables as GOCCP (see Table D1 in Appendix D) on the same horizontal and vertical resolutions, but CLIMP-ST and CLIMP-LT have different cloud detection thresholds because they aim to tackle slightly different science objectives.

The CLIMP-ST product is designed to make full use of ATLID capability to evaluate cloud description in climate models. CLIMP-ST is expected to contain optically thin cloud detected in daytime conditions at full resolution that were not observed by former space lidars at such high spatial resolutions during daytime. This new information, if confirmed in actual data, will help make progress on our current understanding of processes tied to thin ice clouds in the climate system. It will help evaluate the description in climate models of optically thin clouds in regions where they are frequent and important for climate, for

example in the tropics and polar regions.

The CLIMP-LT product is designed to detect the same clouds as CALIPSO-GOCCP. Merging CLIMP-LT with GOCCP will allow building a multi-decade cloud profile record, useful to monitor the cloud inter-annual natural variability and cloud changes induced by human-caused climate warming. This record, if quality is sufficient, will be useful to evaluate climate prediction of cloud changes and to help reduce uncertainties in model-based climate feedbacks and climate sensitivity.

To design CLIMP-ST and CLIMP-LT, we examined the differences between CALIOP and ATLID, space lidars that operate at different wavelengths and use different observation techniques and detectors. We sought to answer two questions: (1) Can the HSRL capability of ATLID help reconcile its cloud retrievals with the CALIOP record? (2) Does the cloud product retrieved from ATLID observations compare well with the one retrieved from CALIOP observations, and if so, how many years of ATLID observations are needed to detect trends in opaque cloud cover or altitude of opaque clouds, assuming ATLID

operation will follow CALIOP without a gap?

To answer these questions, we coupled the outputs of the 3DCLOUD model with the COSP2 simulator and added instrumental noise for two cloud scenes, thin cirrus clouds at ~15 km in the tropics and stratocumulus clouds at ~1 km height. CALIOP and ATLID orbits over these cloud scenes were simulated both for nighttime and daytime conditions, at full vertical and horizontal (1/3 km) resolution and at 1 km horizontal resolution. Then, we applied a wavelength conversion algorithm to ATLID

observations to convert UV lidar profiles into 532nm lidar profiles and added synthetic noise generated for each instrument in accordance with its characteristics.

We addressed the first question for CLIMP-ST. We showed that the lower daytime noise of ATLID allows applying more sensitive thresholds for cloud detection $\{SR(532nm, z) > 3; ATB(532nm, z) - ATB_{mol}(532nm, z) > 1.5 \times 10^{-6} \ m^{-1} \ sr^{-1}\}$ than for CALIPSO at full spatial resolution in daytime without introducing a bias. This suggests that ATLID

may provide new information on optically thin clouds at daytime conditions at full spatial resolution.

We addressed the second question for CLIMP-LT. We search for consistency between ATLID and CALIPSO-GOCCP in cloud detection, therefore we applied the same cloud detection threshold $\{SR(532nm, z) > 5; ATB(532nm, z) - ATB_{mol}(532nm, z) > 2.5 \times 10^{-6} \ m^{-1} \ sr^{-1}\}$ to both instruments, then their nighttime cloud products are comparable, whereas the daytime CALIOP clouds are characterized by somewhat higher false detection rate. This suggests ATLID and

CALIPSO might observe the same clouds, with some adjustment in the cloud detection scheme. Then we analyzed 24 years of predictions from two general circulation models (IPSL-CM6 and CESM2) in the RCP85 scenario, coupled with the COSP lidar simulator. We show that IPSL-CM6 predicts the opaque cloud cover trend detection will require 7 years of ATLID operation besides the existing CALIOP cloud data set, whereas CESM2 predicts the opaque cloud cover trend can be detected in 4 years. For the clouds above 3 km altitude, these numbers change to 4 and 8 years, respectively, and for the altitudes below

3 km the IPSL-CM6 clouds indicate no trend and CESM cloud trend detection will require one year of ATLID operation. These differences in climate predictions highlight the need for a multi-decade cloud lidar record.

The current results rely on a comparison of exactly the same atmospheric scenes "virtually observed" by two space lidars, and they were obtained in the framework of comparing the cloud detection capabilities of these two instruments. However, the comparison of the actual ATLID measurements with actual CALIOP ones will face with an uncompensated difference linked

to the local solar time sampling by CALIOP and ATLID. The difference in the diurnal cycle will bias the detected cloud amount and height. This is a separate issue that should be compensated for, and this should be a subject of a separate work. Moreover, the comparison of actual ATLID measurements with CALIOP ones will probably face unexpected differences other than the ones foreseen in this paper. Therefore, the CLIMP algorithm will require an adjustment after ATLID launch to take those into account.

That being said, this study suggests that it is likely that ATLID will provide new information useful to help evaluate cloud description in climate models beyond the existing space lidar observations. Moreover, merging the ATLID data with the CALIOP data will probably provide important information on cloud response to climate warming.

**Appendix A**

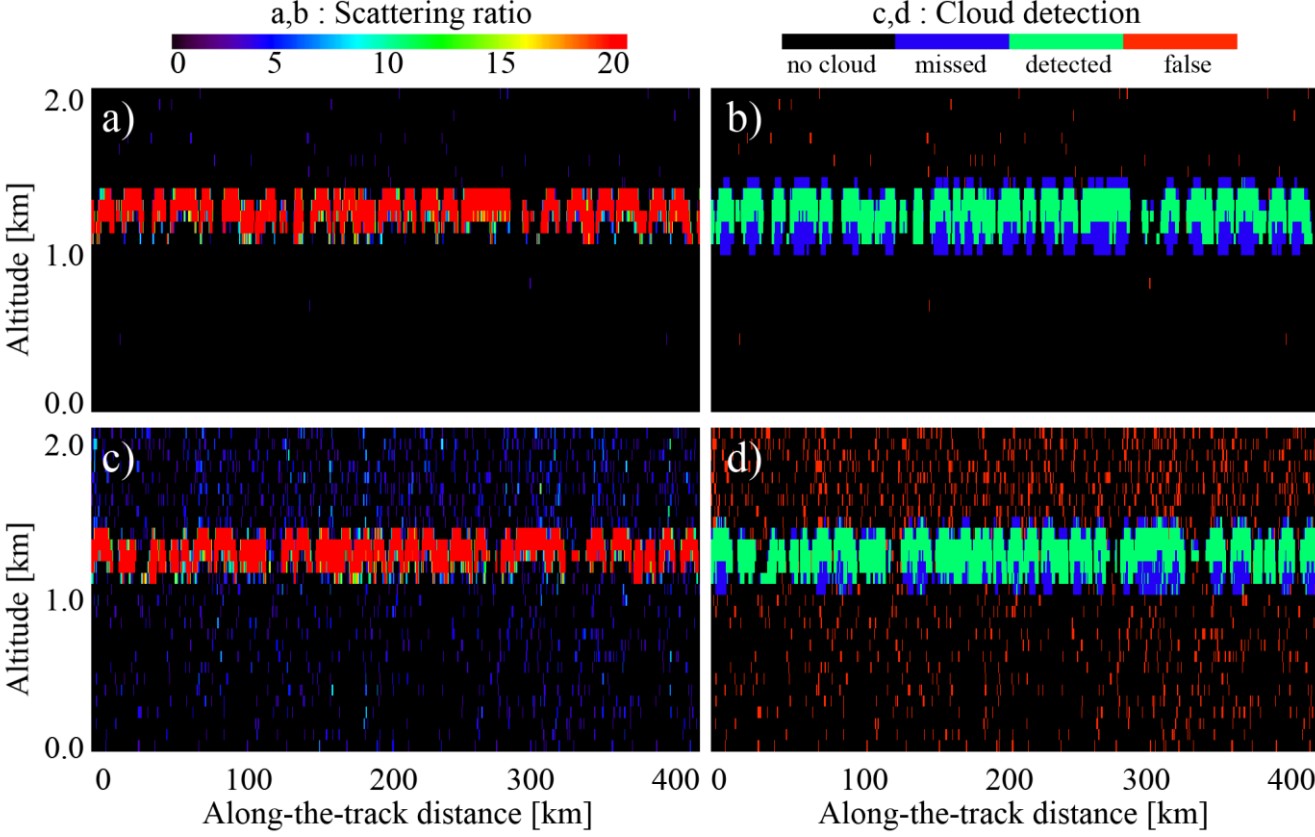

**Figure A1: Scattering ratio and cloud detection statistics estimated for stratocumulus clouds observed by CALIOP using Eq. 36: (a) scattering ratio, night; (b) cloud detection, night; (c) scattering ratio, day; (d) cloud detection, day.**

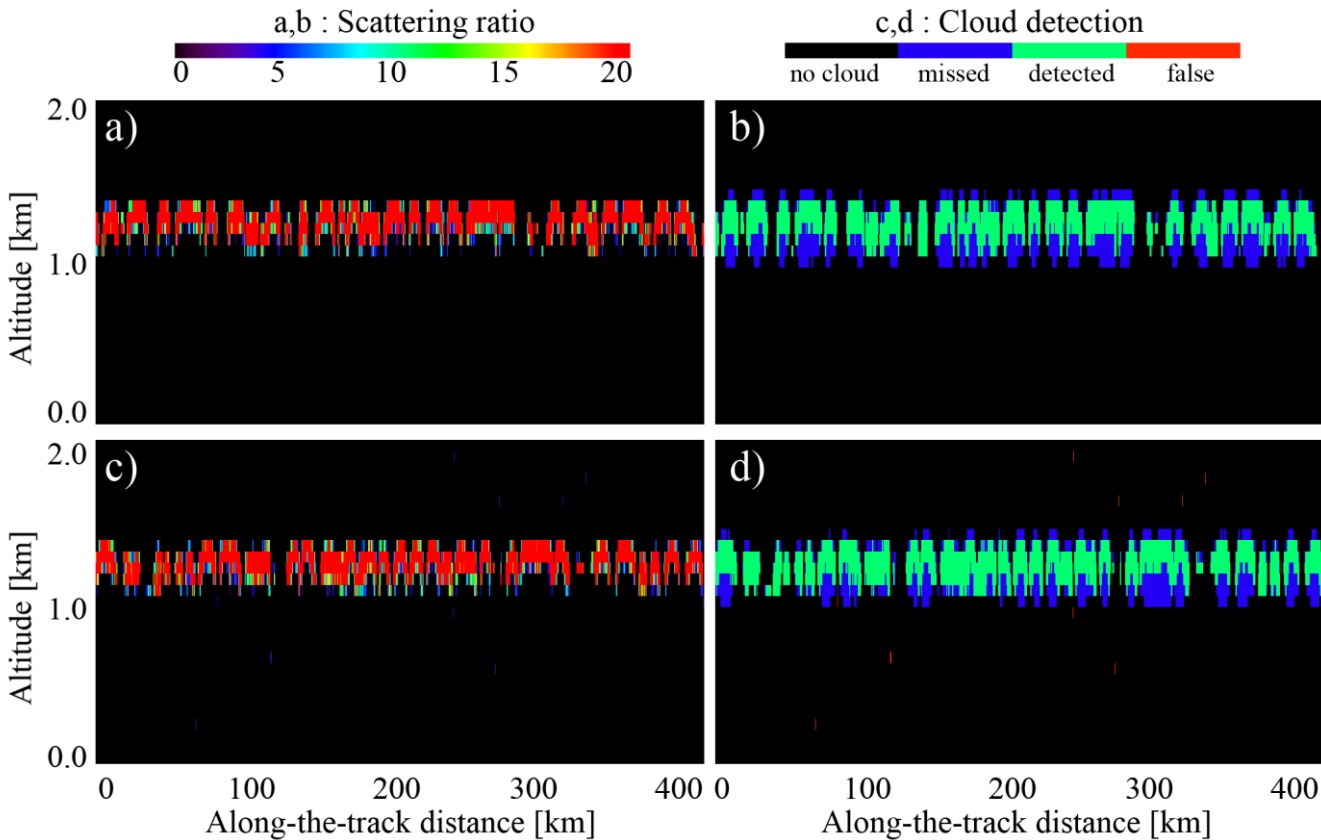

**Figure A2: Same as Fig. A1, but for ATLID.**

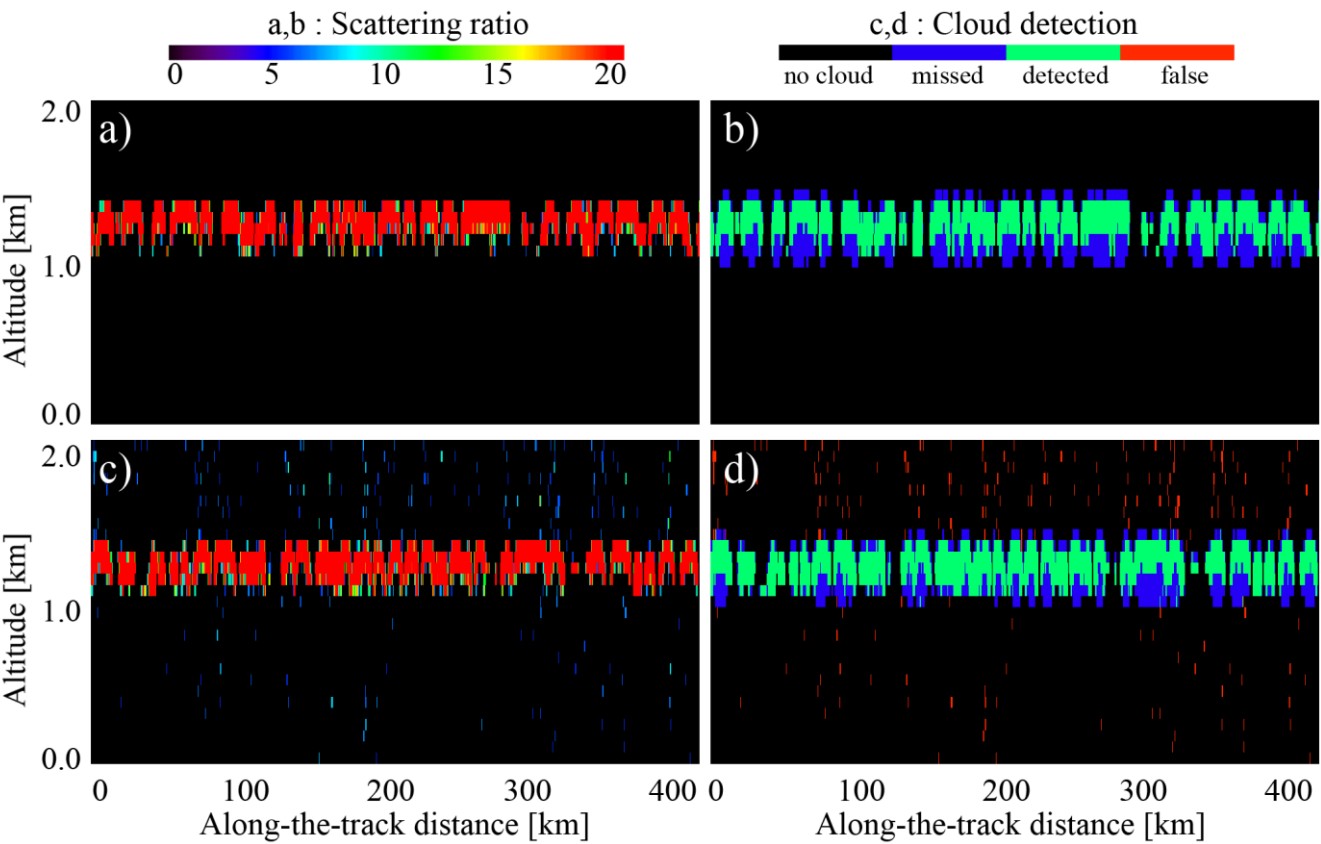

**Figure A3: Scattering ratio and cloud detection statistics estimated for stratocumulus clouds observed by CALIOP using Eq. 6: (a) scattering ratio, night; (b) cloud detection, night; (c) scattering ratio, day; (d) cloud detection, day.**

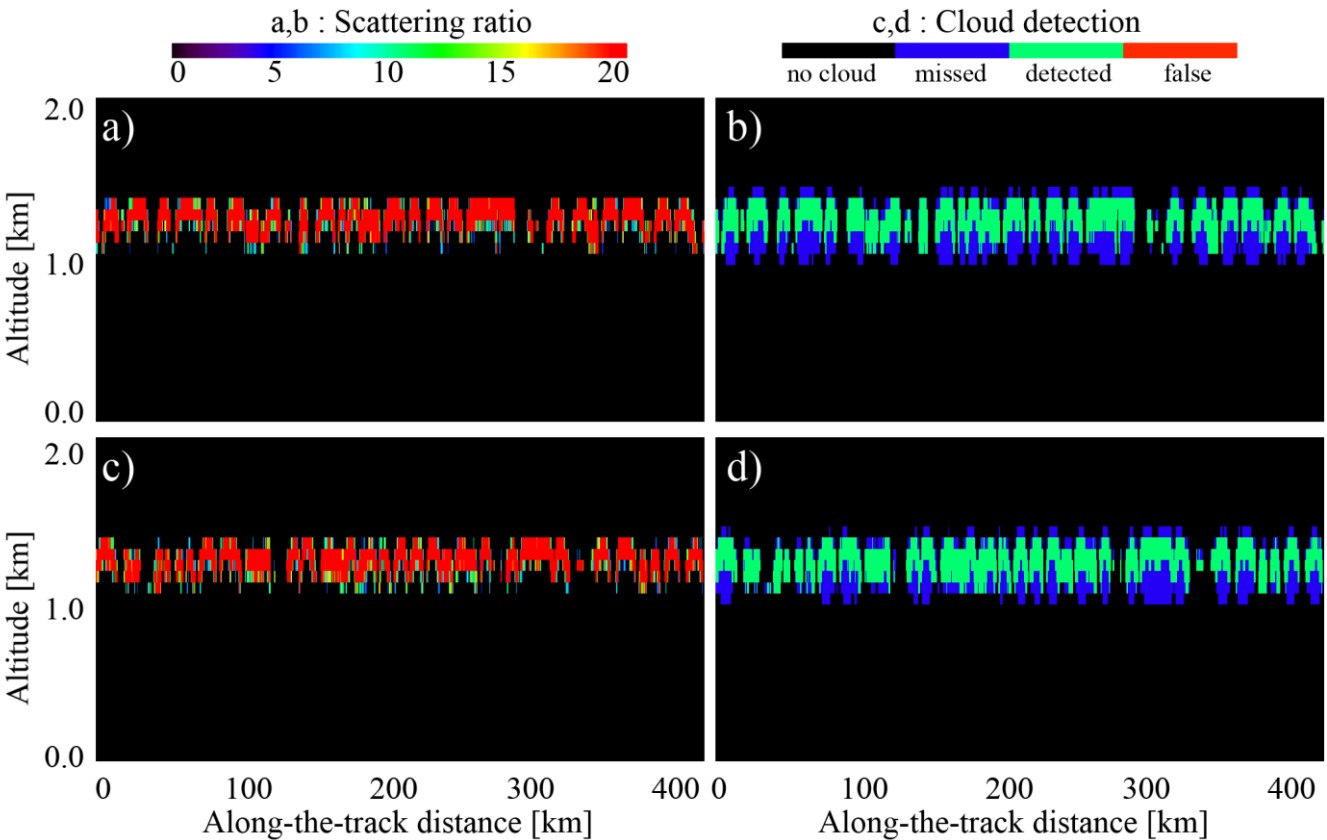

**Figure A4: Same as Fig. A3, but for ATLID.**

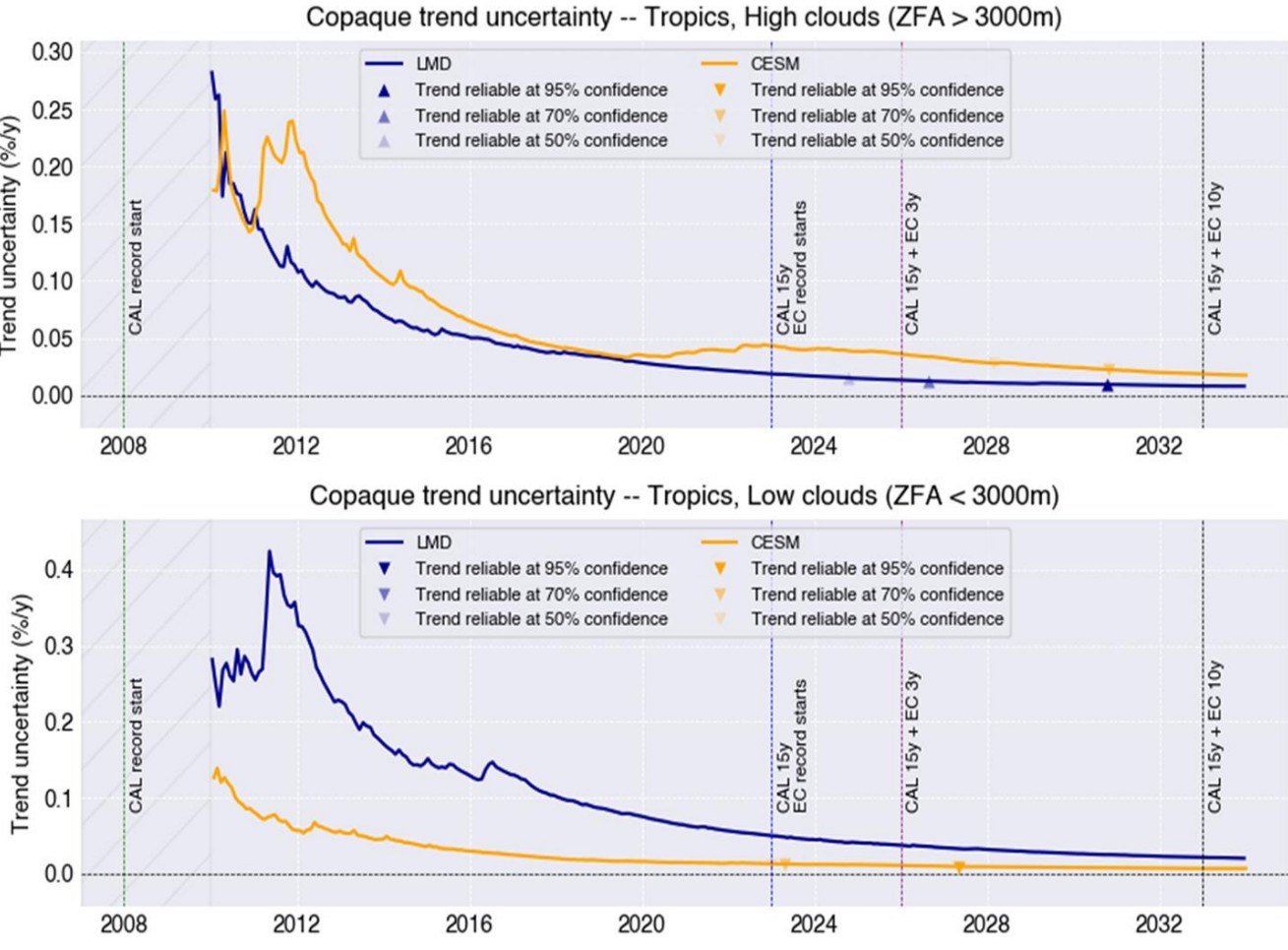

**Figure B1: Same as Fig. 12, but with a separate analysis of high-levels clouds (top) and low-level clouds (bottom)**

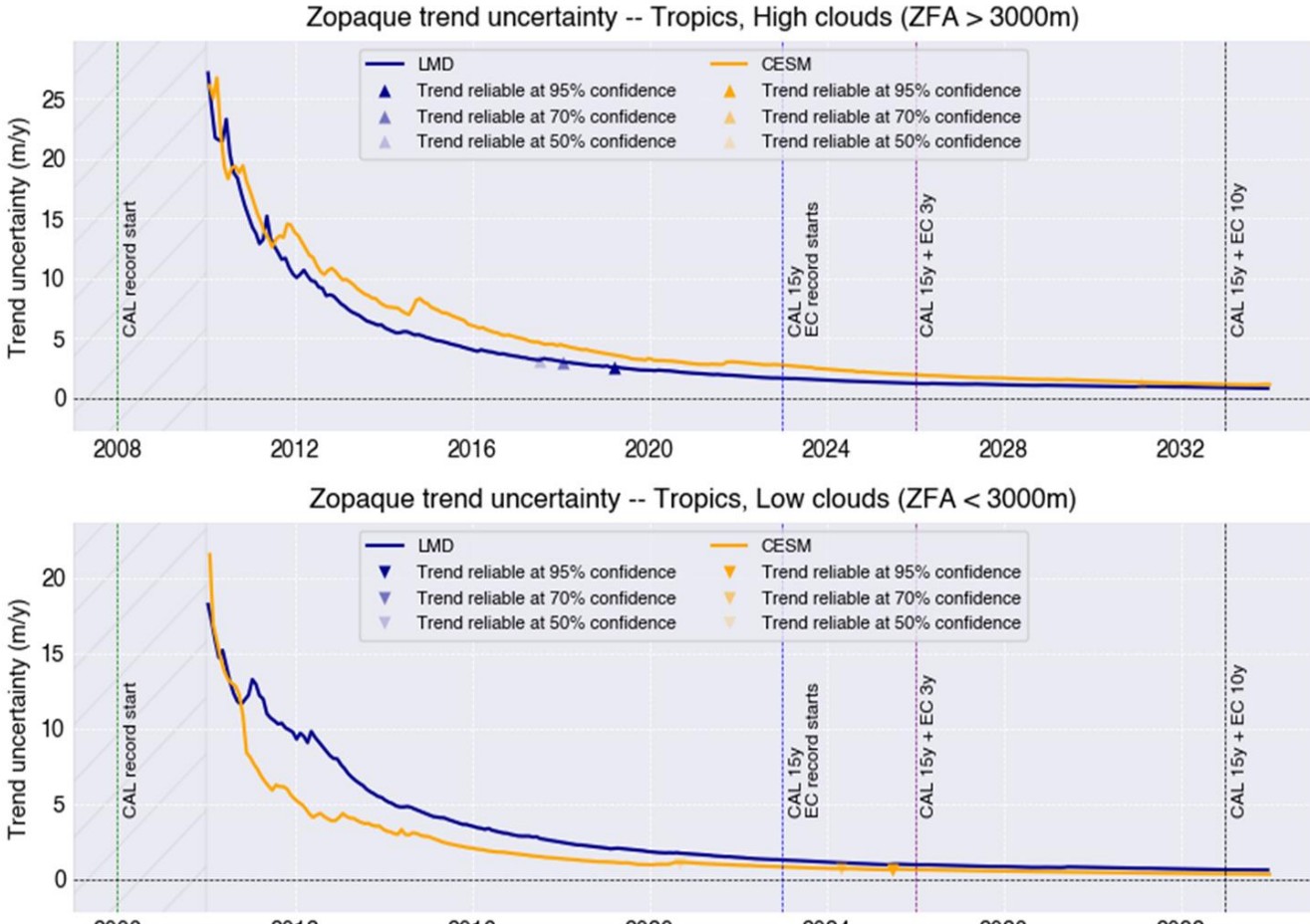

**Figure B2: Same as Fig. 13, but with a separate analysis of high-levels clouds (top) and low-level clouds (bottom)**

**Appendix D**

| Variable Name | Unit | Dimension | Remarks |
|---|---|---|---|
| Time_UTC | Seconds | Ntime | same unit as in ATLID L1B file |
| Altitude | Meters | NZ | |
| Levels | Meters | Nlev (4) | |
| Flags | Unitless | Nflags (6) | |
| Lon | Degree | Ntime | |
| Lat | Degree | Ntime | |
| Surface_elevation | Meters | Ntime | from DEM and/or lidar ground return |
| Temperature | Degree C | Ntime x NZ | From ECMWF in ATLID L1B |
| Pressure | hPa | Ntime x NZ | From ECMWF in ATLID L1B |
| Scattering_ratio | Unitless | Ntime x NZ | |
| Layer_identification_mask | Unitless (int8) | Ntime x NZ | See Table D2 |
| Quality_flags | 0/1 (int8) | Ntime x NZ x Nflags | See Table D3 |
| Cloud_presence | 0/1 (int8) | Ntime x Nlev | nlev cloud flag at specific vertical levels |
| | | | nlev=0 - anywhere in the profile |
| | | | nlev=1 - at low levels |
| | | | nlev=2 - at mid levels |
| | | | nlev=3 - at high levels |

Table D1: Variable definitions for ATLID cloud product. Variables are of type Real (float64) unless specified otherwise. Shaded variables are used as dimensions.


| Bin | Corresponding SR values |
|---|---|
| 0 | Fully attenuated region: SR < SR_bins[0] (default 0.01) |
| 1 | Clear-sky region: SR_bins[0] < SR < SR_bins[1] (default value 1.2) |
| 2 | Unclassified region: SR_bins[1] < SR < SR_bins[2] (default value 3.0) |
| 3 to 11 | Cloud region: SR > SR_bins[2]. The actual bin number provides information on SR intensity within the cloud, with 3 = weakest signal and 11=strongest signal. Defaults: |

| 3 | 4 | 5 | 6 | 7 | 8 | 9 | 10 | 11 |
|---|---|---|---|---|---|---|---|---|
| 5 | 7 | 10 | 15 | 20 | 25 | 30 | 40 | 50 |

Table D2: Layer identification mask description

| Flag value | Explanation |
|---|---|
| 0 | Missing or unreliable data, according to cross-talk information from ATLID level 1b. If Mie, Rayleigh, Geo-localization or atmospheric quality are not good enough, the profile will be rejected and be considered as missing or unreliable. |
| 1 | Data located below the surface elevation |
| 2 | Noisy data, according to molecular calibration. If the calibration R is not within range, the entire profile is flagged as noisy. |
| 3 | Conflicting cloud detection indicators in the upper troposphere SR<3 and $\Delta$ATB>1.5e-6 m$^{-1}$ sr$^{-1}$. |
| 4 | Presence of very bright clouds (SR> 50) anywhere in the profile |
| 5 | Negative SR (SR<0). Can appear in fully attenuated cloud mask (SR < 0.01) |

Table D3: Quality flag indicator

**Author contribution**

HC, VN, and AF: conceptualization, investigation, methodology, and validation; FS: calculating cloud distributions; AF: data curation and formal analysis; AF: writing original draft; AF, HC and VN: review and editing.

**Competing interests**

The authors declare that they have no conflict of interest.

**Code/data availability**

With this work, we do not deliver the codes or the data other than those presented in the tables and figures of the article.

**Acknowledgements**

This work was supported by the European Space Agency (ESA) under ESA Contract No. 4000xxxxxx/221/I-DT-lr. This work was also supported by the Centre National de la Recherche Scientifique (CNRS) and by the Centre National d'Études Spatiales (CNES) through the Expecting EarthCARE, Learning from A-Train (EECLAT) project as well as by the Office National 860 d'Etudes et de Recherches Aérospatiales (the French Aerospace Laboratory, ONERA). The authors want to thank Institut Pascal at Université Paris- Saclay and the organizers of the "4-week Atmospheric Cycle Workshop" for inspiring this study (French program "Investissements d'avenir" ANR-11-IDEX-0003-01"). Special thanks to Dr. Valery Shcherbakov (Laboratoire de Météorologie Physique, Université Clermont Auvergne, Aubière, France) for discussing the multiple scattering coefficient in relation to CALIOP and ATLID. Lastly, the authors express their gratitude to the reviewers, Dr. Sebastien Bley 865 (ESA-ESRIN/LITR) and Dr. Mark Vaughan (NASA Langley) for their thorough reviews and valuable comments.

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
