# Peer review of "Incorporating EarthCARE observations into a multi-lidar cloud climate record: the ATLID Cloud Climate Product"

_EGUsphere, 2022_

## Referee Comment (RC1)

**Referee comment on egusphere-2022-1187**

**General comments:**

The manuscript by Feofilov et al. aims to demonstrate the cloud detection capabilities of ATLID including the advantages in contrast to other space borne lidars, particularly CALIOP. The authors show that ALTID will be more sensitive to optically thin clouds compared to CALIOP. This finding is used to build the short-term ATLID cloud climate product. To generate a long-term cloud climate product, however the thresholds are chosen to be consistent with CALIOP in order to merge CALIOP and ATLID observations. It is finally discussed how long ATLID onboard EarthCARE needs to be operated in order to calculate reliable trends in opaque cloud cover, which is assumed to be highly influence by human-induced climate warming.

The manuscript is well written and addresses important aspects for cloud climate studies. I mainly have minor comments and technical corrections and I am convinced that after taken these into account the paper can be published. I would suggest some slight restructuring of the chapters (see specific comments) and some more details in the final discussion (also with respect to the case when there are no CALIOP and ATLID measurements available at the same time). I think it is a very good paper that just needs a bit more work before being published.

**Specific comments:**

1. The manuscript is structured in 'Definitions', 'Short-term cloud dataset and 'Long-term cloud dataset'. The 'simulated lidar profiles over cirrus and stratocumulus clouds' part is an important input dataset for the full analysis, but it is only a subsection of 'Short-term cloud dataset'. I would therefore suggest to add an additional chapter between Definitions and Short-term dataset called something like 'Simulated lidar profiles', because it is also part of the long-term dataset.

2. I would like to see some more discussion for the case that EarthCARE starts operation later than CALIOP stops. Could be a work around to use some typical cloud scenes characterized by CALIOP and later with EarthCARE to find the same cloud regimes to tune the long-term cloud dataset without intercalibration between the instruments operating at the same time?

3. It is stated that a long-term cloud record can be produced when using a kind of less sensitive cloud detection threshold (based on SR and the attenuated total backscatter) which improves the agreement between CALIOP and ATLID. —> But in that case, you are missing some thin clouds which ATLID would be capable to detect. Could ATLID help quantifying how CALIOP underestimated the global cloud coverage in past datasets?

4. Climate models have large uncertainties as shown in Perpina (2021) —> therefore a long space borne lidar record is essential to better quantify trends and understand the inter model differences. If ATLID cannot fill the long-term gap after CALIOP because it is likely not going to operate as long as CALIOP. How could upcoming satellite missions Aeolus-2 or AOS) help overcoming this long-term challenge after ATLID? You are mentioning this aspect in L 566, but could go in some more detail.

**Technical corrections:**

L 1-18: You should try shorten the introduction part of the abstract. The whole abstract is way too long. Parts of the motivation and introduction can be addressed in detail in the Introduction chapter.

L 31: ATLID-ST: Please define. Or do you mean CLIMP-ST?

L 56: clouds properties —> Shouldn't it be 'cloud properties?

L 106: Rephrase to: "Avoid overestimation of the cloud fraction… "

L 110: Averaging le lidar signal. Should be "averaging the lidar signal"

L 121: optically thinner "cloud". Through the text, you always write 'cloud', but it should be clouds.

L 129: Chapter 2: Definitions (rather Methods? See Specific comment above)

L 199: Rephrase to "Or if it was sampled"…

L 274: Rephrase to "tropical part of the orbit"

L 320: Voluntarily split the -> voluntarily seems not the write phrase here (maybe better artificially??)

L 329: We set the cloud mask to 1 whenever IWC>0. Shouldn't be the instrument sensitivity be taken into account here? Very small IWC values (<0.001) could be model specific, but does not represent what the lidar would see.

L 337: Better rephrase to "Fig. 4 and 5. demonstrate..."

Fig. 4: Please improve the labelling. What is CALIOP and what is ATLID becomes not really clear here. (a), (b), (c) and (d) are explained doubled, (e)-(h) are missing. I would suggest to contrast the two instruments, always CALIOP left and ATLID right would make the differences more visible.

 L 381: particulate backscatter ?? Particle backscatter is more common, also in Line 383.

L 394: "the detectability "

L 397: "the daytime noise increases the disagreement with the reference cloud dataset". This is unclear what is meant. Increases proportional to the disagreement to the reference dataset...?

L 401: during day and night

L 415: Table 2: What is the end of the caption? It is unclear where the free text continues.

L 539: Any idea what makes the big difference in the observation requirement (with respect to EarthCARE operation lifetime) between IPSL-CM6 and CESM1?

L 575: What would be the required overlap period between CALIOP and EarthCARE for optimal intercalibration? Could Aeolus help overcoming the problems when CALIOP and EarthCARE are not flying at the same time (due to CALIOP and Aeolus operating synchronously and Aeolus having a HSRL lidar more comparable to ATLID than CALIOP and ATLID?)

L 650: probably be helpful is too weak in my opinion! "The results in this study using simulations indicate that a merged dataset between CALIOP and ATLID will provide important information… " would be more sound. In general the final paragraph of the conclusions could be phrased a bit stronger showing the benefit of this study.

---

## Referee Comment (RC2)

**Review of "ATLID Cloud Climate Product"**
**by A. G. Feofilov, H. Chepfer, V. Noël, and F. Szczap**

Reviewed by Mark Vaughan (mark.a.vaughan@nasa.gov)

As indicated by the manuscript title, the authors are proposing the creation of a multi-decade lidar-derived cloud climate record that fuses 16+ years of CALIOP observations with new measurements to be acquired during the upcoming EarthCARE mission. To do this, they have adapted the GOCCP layer detection algorithm, designed for CALIOP's elastic backscatter measurements, for use with the high spectral resolution lidar (HSRL) data that will be acquired by ATLID. With this scheme they intend to produce two new products: a short-term climate record that will be used to "evaluate the description of cloud processes in climate models, beyond what is already done with existing space lidar observations" and a long-term record "to evaluate the cloud climate variability predicted by climate models". Part 1 of the manuscript describes the methods that will be used for combining the CALIOP and ATLID measurements. Part 2 discusses what sorts of things we might expect to learn by probing this extended data record.

The authors have quite obviously put a huge amount of work into this manuscript and I expect it to eventually make a valuable contribution to the scientific literature on cloud trends in a changing climate. Unfortunately, in my opinion it is not yet ready for publication, as details essential for fully evaluating the comparisons presented are too often elided. My primary concerns are as follows.

a.  The authors do not enumerate the uncertainties inherent in the ATLID measurement technique, nor do they address the potential effects of these uncertainties in their cloud detection scheme. For example, assuming a perfect measurement system, deriving 532 nm attenuated scattering ratios (i.e., the authors' equation (5)) from ATLID's 355 nm HSRL measurements can be done as follows:

$$SR(532, z) = \left(1 + R_m(z)\left(\frac{Eq.\ 2(355, z)}{Eq.\ 3(355, z)}\right)\right)\left(\frac{Eq.\ 3(355, z)}{Eq.\ 4(355, z)}\right)$$

$$= \left(1 + R_{p,532}(z)\right)\exp\left(-2\int_{z_{sat}}^{z} \eta\, \alpha_{p,355}(z')\, dz'\right)$$

where $R_m(z) = \frac{\beta_{m,355}(z)}{\beta_{m,532}(z)} \approx 5.33$. Per item b. below, this development assumes that $\beta_{p,355}(z) = \beta_{p,532}(z)$ and $\alpha_{p,355}(z) = \alpha_{p,532}(z)$. Potential biases in this formulation will come from imperfect knowledge of

1.  the electro-optic gains for the HSRL molecular and particulate channels and the perpendicular channel;

2.  the particulate vs. molecular signal crosstalk in the two HSRL channels (e.g., the interferometer contrast ratio, as in Burton et al., 2018; based on table 1 in do Carmo et al., 2021, I'm guessing the ATLID contrast ratio is ~3.4); and

3. the parallel vs. perpendicular signal crosstalk in the perpendicular channel and the HSRL particulate channel.

At this stage in the development of ATLID, I would assume that reasonably accurate estimates of these calibration coefficients and crosstalk parameters have been developed by the instrument and algorithm engineers. With these values, presumably the authors could develop a rough order of magnitude estimate of potential detection biases.

b. Some fundamental assumptions are not identified. In particular, the transformation of the ATLID attenuated scattering ratios from 355 nm to 532 nm requires that both the cloud backscatter and extinction coefficients are spectrally independent. Beyerle et al. (2001) suggest that this is generally true for cirrus. On the other hand, Voudouri et al. (2020) offer evidence to the contrary.

c. Related to the above, the authors do not make a convincing case for converting ATLID 355 nm measurements to their approximate realizations at 532 nm. Assuming the electro-optic gains (AKA, calibration coefficients) are accurately known for all three channels, one can derive the authors' equation (1) at 355 nm by summing equation (2), equation (3), and the perpendicular channel analog of equation (3). This approach obviates the need for accurate knowledge of the crosstalk parameters. (My guess is that imprecise knowledge of crosstalk (i.e., in computing the ratio between equation (2) and equation (3)) could be the dominant source of detection uncertainty.) Invoking the assumption in item b. (i.e., that $\beta_{p,355}(z) = \beta_{p,532}(z)$), one can readily derive the 355 nm threshold that is equivalent to the GOCCP threshold of 5 used at 532 nm ($1 + 4/5.33 \approx 1.75$).

(I note that CALIOP layer detection can also be biased by imprecise knowledge of calibration coefficients; e.g., underestimating the 532 nm calibration coefficient leads to increases in false positive detections.)

d. On several occasions the authors assert that ATLID signals will suffer much less than CALIOP from solar background noise. What's missing from the manuscript are explanations for this improved daytime performance and how the simulated cloud fields were converted to lidar measurements. I'd very much like to see Section 3 expanded to include

   o high level descriptions of the differences between the ATLID and CALIOP detectors

   o some comments on how synthetic noise is generated for each instrument (note: per Liu et al., 2006, Poisson-distributed noise is not quite correct for the "analog-mode APD and PMT detectors" used by CALIOP.)

   o an accounting of the differences in solar background rejection between ATLID and CALIOP

   o diagnostic comparisons of simulated CALIOP signals and the real thing; one informative example would be the SNR computed for 1500 km along track in clear skies between 19 km and 20 km.

On last idle question about noise characteristics: will ATLID be affected by the South Atlantic Anomaly in the same way that CALIOP is? (Hunt et al., 2009)

e. Lots of work obviously went into constructing the simulated cloud data. I just wish I had a handy dandy metric for evaluating the representativeness of these simulations with respect to things a lidar directly measures (e.g., attenuated backscatter coefficients and volume

depolarization ratios as functions of temperature/altitude).  How well do these simulations capture the natural variability of cloud backscatter?  What is the distribution of optical depths for stratocumulus clouds?  What is the lower limit of the attenuated backscatter coefficients within the simulated cirrus?  And how do these characteristics of the simulated data sets compare to real-world values harvested from CALIOP measurements?

f.  Figures 4 and 5

Comparisons of the ATLID and CALIOP measurements shown in these figures are more difficult to interpret than one would like simply because they show two different quantities: attenuated *particulate* backscatter for ATLID and attenuated *total* backscatter for CALIOP.  I believe these plots would be much more useful to readers if they showed apples-to-apples comparisons of the signals that are actually used the detection algorithm; i.e., either 532 nm attenuated scattering ratios or (perhaps better?) 532 nm attenuated backscatter coefficients.  I also believe they would be much more informative if the colors were rendered using a log scale.

Here's a specific example illustrating both points. Line 365 says, "Another remarkable feature shown in this plot is higher daytime noise for CALIOP (Fig. 5bd)".  First, I think the CALIOP day–night differences might be better illustrated by drawing the readers' attention to panels 5c and 5d (i.e., not 5b and 5d) and pointing out the clearly lower SNR in panel 5d, which is most readily visible in the molecular atmosphere above the cloud.  But second and more important, this same kind of day–night comparison will not be useful for ATLID, simply because the ATLID panels (5g and 5h) show attenuated *particulate* backscatter.  Since there is no molecular contribution to the ATLID signal, the SNR is close to zero in both cases, irrespective of the magnitude of the noise.  This fact, combined with the very dark colors used at the low end of the SNR scale, make it exceedingly difficult to distinguish any changes in the ATLID day versus night noise levels.  (Because the in-cloud signals saturate the high end of the color scale in all eight panels, I cannot discern any meaningful day versus night or CALIOP versus ATLID SNR differences.  Perhaps readers with more acute vision will do better.)

In addition to the remarks above, I am also attaching an annotated version of the manuscript into which I have inserted a fairly large number of additional comments, questions, and suggestions. These notes both amplify some of the concerns mentioned above and surface a few new, perhaps more detail-oriented issues. Please accept my apologies in advance for any repetition that may be encountered while perusing the annotated manuscript.

[revised manuscript text omitted]

---

## Author Comment (AC2)

First of all, we would like to thank Dr. Vaughan for his thorough analysis and in-depth review of our manuscript. In the new version, we tried to address all the issues identified in his review. The responses to major and minor comments are given below. We marked the reviewer's and the author's comments by "RC:" and "AC:", respectively.

We would like to preface our answers with the following introduction, which partially covers some of the comments and at the same time draws attention to one of the key elements of the methodology presented in the manuscript. It is traditionally considered that the advantage of an HSRL lidar is its capability to measure two backscatter components independently. Indeed, the possibility to split and measure the molecular and particulate backscatter signals is invaluable, but, and we want to stress it here, both signals are not noise-free. At the same time, the amount of information about the clouds is much higher for the particulate channel than for the molecular one. If there is a method of cloud detection, which makes the cloud retrieval at 355 nm consistent with the existing one at 532 nm and if the noise of the useful signal could be reduced by replacing the noisy measured molecular backscatter by a noise-free calculated one, then we should use this method even though it is equivalent to using the measured molecular backscatter in a noise-free case. All this does not make the molecular channel measurement redundant, but we suggest to exclude it from cloud detection.

**Major comments**

RC: The authors do not enumerate the uncertainties inherent in the ATLID measurement technique, nor do they address the potential effects of these uncertainties in their cloud detection scheme. … Potential biases in this formulation will come from imperfect knowledge of (1). the electro-optic gains for the HSRL molecular and particulate channels and the perpendicular channel; (2). the particulate vs. molecular signal crosstalk in the two HSRL channels, and (3). the parallel vs. perpendicular signal crosstalk in the perpendicular channel and the HSRL particulate channel.

AC: We agree that we did not discuss this in the manuscript. Moreover, we did not consider the perpendicular channel in our simulations since this would have added an extra layer of complexity, which is not related to the comparison of cloud detection capacity of the instruments. The overarching goal of this study was to determine whether the clouds retrieved from both lidars are comparable and whether one can use them to create a continuous, self-consistent, and long-term cloud record. We assume that the clouds depolarize the same way at 355 nm and 532 nm and that one can always get back to total backscatter by summing the parallel and perpendicular components. Therefore, we did not consider the perpendicular channel separately. As for the (2)nd point, we considered the whole signal chain, including error propagation in our estimated of noise in cloud detection. Indeed, the calibration issues leading to biases in cross-talk coefficients will lead to biases in cloud detection. The same way, any issue with CALIOP calibration will also lead to biases. But, the calibration is performed onboard and we assume that on average all the calibration coefficients are known with an accuracy, which is beyond the cloud detection accuracy requirements. However, the single-shot noise of a given spaceborne lidar cannot be reduced, and our task was to compare how these two instruments compare in terms of signal-to-noise when it comes to cloud detection. We have added corresponding explanations to the end of the Introduction.

RC: Some fundamental assumptions are not identified. In particular, the transformation of the ATLID attenuated scattering ratios from 355 nm to 532 nm requires that both the cloud backscatter and extinction coefficients are spectrally independent. Beyerle et al. (2001) suggest that this is generally true for cirrus. On the other hand, Voudouri et al. (2020) offer evidence to the contrary.

AC: On the one hand, this is true and to generalize the conversion one has also add a color ratio to the corresponding backscatter and extinction terms. On the other hand, this color ratio is not well-known for an arbitrary cloud particle. In addition, the continuation of the Aeolus cloud retrieval activity (Feofilov et al., 2022) showed that it is enough to compensate for depolarized component missing in ALADIN's measurements, and for diurnal cycle to get a good agreement between the clouds retrieved from CALIOP and ALADIN (see the description of the method in presentation Feofilov et al., 2023), so we assume that color ratio play a secondary role compared to other terms. Moreover, we had a look at the numbers provided in Voudouri et al. (2020) and we did not find a strong wavelength difference of backscatter and extinction coefficients, considering the error bars provided in this article. Still, we added a short discussion and the suggested references after Eq. 7 in Section 2.3.

RC: Related to the above, the authors do not make a convincing case for converting ATLID 355 nm measurements to their approximate realizations at 532 nm. Assuming the electro-optic gains (AKA, calibration coefficients) are accurately known for all three channels, one can derive the authors' equation (1) at 355 nm by summing equation (2), equation (3), and the perpendicular channel analog of equation (3). This approach obviates the need for accurate knowledge of the crosstalk parameters. (My guess is that imprecise knowledge of crosstalk (i.e., in computing the ratio between equation (2) and equation (3)) could be the dominant source of detection uncertainty.) Invoking the assumption in item b. (i.e., that $\beta_{p,355}(z) = \beta_{p,532}(z)$), one can readily derive the 355 nm threshold that is equivalent to the GOCCP threshold of 5 used at 532 nm ($1 + 4/5.33 \approx 1.75$)

AC: We agree that the conversion should work both ways and instead of converting the optical properties themselves one could have converted the threshold. But, the way we do it has two advantages over the "inverted" procedure suggested by the reviewer: (1) in our approach, the molecular backscatter and attenuation at 532nm are used both in the numerator and denominator and in the difference term of Eq. 5, and the SR becomes less noisy because the only remaining source of noise is the particulate backscatter. As we discuss in the article, the information about the clouds is carried by the photons backscattered from the particles and if we can get an extra source of noise-free information on molecular backscatter and attenuation, we should use it instead of measured molecular backscatter noisy signal; (2) only one set of rules is applied to SR and ATB and one can easily change them without further recalculation.

RC: On several occasions the authors assert that ATLID signals will suffer much less than CALIOP from solar background noise. What's missing from the manuscript are explanations for this improved daytime performance and how the simulated cloud fields were converted to lidar measurements.

AC: We do not state that the solar noise in CALIOP observations is significantly stronger than that of the estimated ATLID observations. Still, we claim that the daytime SNR is in favor of ATLID and this allows to put the cloud detection threshold somewhat lower than for CALIOP. To answer this question and associated sub-questions below, we added the whole chain of signal generation and error estimate, see "3.2.2 Estimating lidar signals and noise" in the updated version of the manuscript.

RC: I'd very much like to see Section 3 expanded to include: high level descriptions of the differences between the ATLID and CALIOP detectors

AC: In the new Section 3.2.2, we provided the formalism necessary to estimate the noise in ATLID and CALIOP channels.

RC: some comments on how synthetic noise is generated for each instrument (note: per Liu et al., 2006, Poisson-distributed noise is not quite correct for the "analog-mode APD and PMT detectors" used by CALIOP.)

AC: This is a strong point and we admit that we did not consider some of the effects mentioned in this article. In the new version of the manuscript, we provide more details about the noise generation, and, indeed, it is worth more discussion, especially the NSF (noise scale factor) issue. As the aforementioned work of Liu et al., (2006) shows, the photomultiplier tubes (PMT) are characterized not only by the enhanced noise factor (ENF), which is related to variation of the gain, but also by the NSF, which represents the effects of the convolution (not the sum!) of several stochastic processes in the PMT. As a result of these processes, the noise still can be considered as Poisson-distributed, but it should be scaled in accordance with NSF. However, there's a controversy regarding the value of NSF, which should be used in the calculations. For example, Eq.9 o this article tells us that NSF=sqrt(ENF x gain) = sqrt(1.5 x1.5e6)=1500 that makes little sense. If we assume that there's a typo and that it's not a total gain like it is said right after Eq.6, but a single-stage gain, which is about $4.15 = 1.5e6^{1/10}$, then it makes more sense: NSF=sqrt(1.5x4.15)=2.45. At the same time, there's a NSF=5.14, which is retrieved from the variability of the real signal and saved along with CALIPSO L1 data. According to Eq. 22 of (Liu et al.,2006), there's no compensation like square root or whatsoever, the NSF goes straight to the retrieved quantities and is supplemented with the second combination term in Eq. 22, so it's easy to get high noise in the simulated product if the NSF is off. To avoid this, we performed a series of tests, the results of which are shown in Fig. 1 and are summarized in Table 1. As one can see, using purely ENF in NSF calculation (NSF=sqrt(1.5)=1.22) like we did in the original version of the manuscript is not enough whereas using the value retrieved from atmospheric observations is too much, so we ended up with NSF=3.16, which reproduces the daytime noise and at the same time does not spoil the nighttime noise.

[Figure]

Fig. 1. A comparison of cloud- and aerosol-free tropical ATB profile measured by CALIOP and simulated with our lidar model for different NSF values.

Table 1. Noise r.m.s. values for the 23-28km vertical interval of measured and simulated ATB profiles, [km$^{-1}$ sr$^{-1}$]

| Day/night | CALIOP | Simulated, NSF=1.22 | Simulated, NSF=3.16 | Simulated, NSF=5.14 |
|-----------|--------|---------------------|---------------------|---------------------|
| Night | 1.5e-4 | 1.7e-4 | 1.8e-4 | 1.9e-4 |
| Day | 1.8e-3 | 6.2e-4 | 1.5e-3 | 2.4e-3 |

Because of this update, we had to switch from showing the instantaneous profiles in Fig. 6-9 to 1 km averages, otherwise the daytime figures become too noisy. This change, however, does not affect the conclusions of the article.

RC: an accounting of the differences in solar background rejection between ATLID and CALIOP

AC: In the new section, we provide the formulae for solar noise and we do a "back-of-envelope" estimate of a ratio of solar photons reaching the detector's surface for ATLID and CALIOP per same range gate. According to this estimate, the ratio is in favor of ATLID (~2.5 times less solar photons come per unit time). This is not the end of the story because this number has to be compared to number of "signal" photons, and the back-of-envelope estimate is more complex, but in any case, the signal-to-solar-noise ratio is in favor of ATLID.

RC: diagnostic comparisons of simulated CALIOP signals and the real thing; one informative example would be the SNR computed for 1500 km along track in clear skies between 19 km and 20 km.

AC: Please, see our comment to synthetic noise generation two comments above. With an adjusted NSF, our nighttime and daytime noise values in the aerosol-free stratosphere are comparable.

RC : On last idle question about noise characteristics: will ATLID be affected by the South Atlantic Anomaly in the same way that CALIOP is? (Hunt et al., 2009)

AC : this question is difficult to answer in the framework of our modeling, so we can only speculate here. As far as we understand, the SAA region affects the operation of lidar in two ways: by affecting the measurement path (detector and amplifiers) and by affecting the laser pulse generation (coronal arcing of Q-switch, leading to lower pulse energy). Since the ATLID does not use PMTs, which amplify any electron emitted by photocathode, including those provoked by high energy particles, we would assume that the electronic noise of ATLID should be less affected by SAA. As for the laser radiance generation, it depends on the peculiarities of the laser system design, and this question is beyond our competence.

RC: Lots of work obviously went into constructing the simulated cloud data. I just wish I had a handy dandy metric for evaluating the representativeness of these simulations with respect to things a lidar directly measures (e.g., attenuated backscatter coefficients and volume depolarization ratios as functions of temperature/altitude). How well do these simulations capture the natural variability of cloud backscatter? What is the distribution of optical depths for stratocumulus clouds? What is the lower limit of the attenuated backscatter coefficients within the simulated cirrus? And how do these characteristics of the simulated data sets compare to real-world values harvested from CALIOP measurements?

AC: This major remark is related to the following minor remarks: Line 222, Line 245, and Line 280, so we give a general answer both to the major and to the three minor remarks here. Please, also see the individual replies to these comments below. The reviewer is right: the development, improvement and the validation of the fast, flexible and realistic three

dimensional cloud generator 3DCLOUD (Szczap et al., 2014 ; Alkasem et al., 2017), and the last version 3DCLOUD_V3 needed a huge amount of work. Applications of a such cloud generator are numerous. We have carried out sensibility analysis of 3D cloudy atmosphere radiative properties to 3D microphysical and optical properties and have quantified 3D radiative effect on the measurements and cloud retrievals in the framework of IIR/CALIPSO (Fauchez et al., 2014, 2015), of CALIOP/CALIPSO (Alkasem et al., 2017), of MODIS/AQUA (Fauchez et al., 2016, 2017, 2018a,b) and POLDER/PARASOL (Cornet et al., 2010, 2013, 2018). The development of 3DCLOUD_V3 was initiated two years ago (funding by ONERA - The French Aerospace Lab French) in order to provide ice water content of 3D cirrus cloud for the radiative transfer code MATISSE that can help to design/realize optronic sensor and develop detection algorithm.

When the reviewer asks "but was it really necessary ?", we answer that 3DCLOUD (Szczap et al., 2014 ; Alkasem et al., 2017) and 3DCLOUD_V3 fulfilled the requirements requested to carry out this study (i.e.realistic 3D water content of cirrus and stratocumulus at very high spatial resolution $\Delta x \approx \Delta y \approx 100$ m and $\Delta z \approx 20$ for 100 km large horizontal extension).

The reviewer suggested to use "CALIOP/CATS/GLASS backscatter profiles" or "cirrus profile measurements acquired by an airborne HSRL" because "doing this should capture "natural variability" in cloud scattering intensity". We think it is a good alternative to use such kind of measurements in order to retrieve information or cloud 2D vertical profiles of water content or 3D water content with assumptions about horizontal spatial redistribution. For example, as point out on Fig.3 in Fauchez et al. (2014), The scale-invariant properties of cirrus are controlled by a $-5/3$constant spectral slope at all the scales and altitude levels according to the cirrus backscattering coefficient at 532 nm measured at different altitudes by the lidar CALIOP/CALIPSO. But the retrieval of full 3D cirrus or cumulus water content from 2D lidar measurement is a very difficult task for us. Actually, and for this study, we had 3DCLOUD and a robust version of 3DCLOUD_V3.

Therefore, as we are experts of 3DCLOUD and of 3DCLOUD_V3, we prefer to use 3DCLOUD and of 3DCLOUD_V3 for this study. We also chose to present quickly 3DCLOUD_V3 in this paper. 3DCLOUD_V3 will be published in detail elsewhere.

RC: Figures 4 and 5. Comparisons of the ATLID and CALIOP measurements shown in these figures are more difficult to interpret than one would like simply because they show two different quantities: attenuated particulate backscatter for ATLID and attenuated total backscatter for CALIOP. I believe these plots would be much more useful to readers if they showed apples-to-apples comparisons of the signals that are actually used the detection algorithm; i.e., either 532 nm attenuated scattering ratios or (perhaps better?) 532 nm attenuated backscatter coefficients.

AC: We see the point, but we have deliberately chosen particulate backscatter for the same reason as we have chosen 355 to 532 nm conversion of optical properties instead of 532 to 355 nm conversion of detection threshold. Please, see our answer to the third major comment above. We would like to draw the reader's attention to the fact, that the information content of these two instruments is different in the sense that the molecular component measured by CALIOP along with particulate one, does not carry information about the cloud. It is just adding magnitude to the signal, but it is not useful per se. On the other hand, the ATLID offers a component, which is directly related to the observed quantity, and it is interesting to show the properties of this very component.

RC: I also believe they would be much more informative if the colors were rendered using a log scale.

AC: We agree that the log scale is sometimes helpful. Unfortunately, we cannot apply it here because of the noise, which sometimes makes the values negative. We tried log scale with an offset of initial data, but the log compression is low in this approach, and the rendered plots are not more informative than the initial ones. Nevertheless, we changed the offsets to me the plots more readable.

RC: Here's a specific example illustrating both points….But second and more important, this same kind of day–night comparison will not be useful for ATLID, simply because the ATLID panels (5g and 5h) show attenuated particulate backscatter. Since there is no molecular contribution to the ATLID signal, the SNR is close to zero in both cases, irrespective of the magnitude of the noise.

AC: We did not get why the "same kind of day–night comparison will not be useful for ATLID". This channel is affected by solar radiance in (almost) the same way as the molecular one, so the day-night difference makes perfect sense. We deliberately do not mix up two sources of noisy data (see above) because we believe that one should use only the particulate backscatter in the analysis whereas the molecular one, needed for cloud definition, might be (and should be) taken from a noise-free source.

RC: In addition to the remarks above, I am also attaching an annotated version of the manuscript into which I have inserted a fairly large number of additional comments, questions, and suggestions.

AC: We transferred these remarks to this document to improve the readability and to answer them point by point in a "conventional" manner. Please, see the "Minor comments" section below.

**Minor comments**

RC, Line13 : a picky point, perhaps, but one might argue that "the era of space-borne optical active sounding of the Earth's atmosphere" actually began with the flight of LITE on NASA's space shuttle; see https://science-data.larc.nasa.gov/LITE/.the first Earth-observing atmospheric lidar to fly on a dedicated satellite was GLAS; see https://attic.gsfc.nasa.gov/glas/ and https://agupubs.onlinelibrary.wiley.com/doi/toc/10.1002/(ISSN)1944-8007.ICESAT1;

AC: Strictly speaking, this is true, but it was not a long-term campaign. To keep the abstract short, we just added "long-term" before the "space-borne".

RC, Line 25: these two lidars use different wavelengths and that fact should be noted at the outset. so consider revising this sentence as follows: "virtual ATLID (at 355 nm) and CALIOP (at 532 nm) measurements"

AC: Fixed, thanks.

RC, Line 28: how well do the CALIOP simulated daytime measurements compare to the real thing? I hope this is discussed in some detail in the main body of the paper.

AC: Please, see the answer to the comment on synthetic noise generation. Also see the Fig. 1 and Table 1 above.

RC, Line 30: since a particulate scattering ratio ($\beta$particulate / $\beta$ molecular) of X at 532 nm will be reduced to a bit less than 0.2 X at 355 nm, claiming improved daytime detection of faint targets by ATLID implies that the solar background noise would be reduced by more than a factor of 5 relative to CALIOP. I hope this too is explored in detail in the main body of the paper.

AC: This is not that straightforward, given that we exclude the noise in molecular channel. What we estimate with a help of a simple ratio is that the number of solar photons hitting the surface of the detector is ~2.5 times less for ATLID.

RC, Line 31: what is the ATLID-ST? how is it different from the CLIMP-ST? (or is it?)

AC: Indeed, the original phrasing could lead to a confusion. We have rewritten these two sentences.

RC, Line 32: in the current literature, there are hundreds of publications that use CALIOP data to investigate optically thin clouds and PSCs, so it's not clear what specific advances the authors envision.

AC: In this paragraph, we discuss the possibility of lowering the detection threshold that should be a step forward in definition of the cloud and in cloud detection. Let's imagine that there is a non-zero amount of the clouds, which we miss. Wouldn't it be interesting to get a reliable distribution of thin clouds, which we miss, and estimate their radiative effect, for example?

RC, Line 32: if "ice polar clouds" actually refers to PSCs, then improved daytime detection won't be highly relevant, as PSCs are a nighttime phenomenon.

AC: we agree that the improved daytime performance will not change the detection of these clouds, so we removed this part of the phrase.

RC, Line 33: I don't understand the motivation for this conversion. why not work directly with the ATLID 355 nm signals?

AC: Please, see the answer to the 3$^{rd}$ major comment.

RC, Line 39: is ATLID expected to achieve a 4-to-7 year lifetime?

AC: The initial lifetime of the mission is 3 years, but, as the Aeolus shows, it can be expanded beyond the planned end of life date. At the moment, we cannot say anything definite about it, so we added a short comment in the brackets.

RC, Line 59: Winker et al. 2010 (https://doi.org/10.1175/2010BAMS3009.1) might be a better reference, as it provides a more comprehensive science overview than the 2009 algorithm paper

AC: We have replaced the reference.

RC, Line 81: why not? surely the general principles apply, yes? (e.g., a simple thresholding algorithm applied to profiles of attenuated scattering ratio, as described in Chepfer et al., 2013) so one might expect that a change of threshold level (necessary to accommodate the larger molecular scattering at 355 nm) would be sufficient.

AC: Please, see the answer to the 3$^{rd}$ major comment.

RC, Line 85: given the focus on long-term measurement records, I think it's essential that the authors mention ATLID's design lifetime somewhere in his paper.

AC: Please, see the comment to line 39.

RC, Line 94: are the authors referring to spatial changes in cloud occurrence/coverage or to changes in cloud microphysics or both? the discussion thus far seems to be focused entirely on spatial properties.

AC: We mean only the cloud occurrence/coverage here. Detection of trends in microphysics is another challenging task. For the sake of simplicity, we do not precise anything in this paragraph, but we added a short disclaimer at the beginning of Section 3.2.3.

RC, Line 108: not averaging horizontally reduces the sensitivity to high thin clouds (e.g., subvisible cirrus). likewise, vertical averaging to 480 m reduces the sensitivity to geometrically thin layers (e.g., subvisible cirrus again). have the authors considered implementing two-dimensional retrieval schemes such as those proposed by Hagihara et al. 2010 (https://doi.org/10.1029/2009JD012344) and Vaillant de Guelis et al., 2021 (https://doi.org/10.5194/amt-14-1593-2021), which can mitigate some of these shortcomings by implicitly accounting for cloud aspect ratio? I realize that doing this would require a major update to the GOCCP algorithm. but would the improved detection of thin and/or attenuated layers be worth the effort?

AC: This can be explored and eventually implemented to the algorithm, but this is out of the scope of the present work, and this won't change the conclusions on the comparability/sensitivity of the instruments.

RC, Line 108: but CLIMP-ST doesn't need to maintain consistency with CloudSat, does it? shouldn't it instead be designed to be consistent with the cloud profiling radar resolution? (42 m???)

AC: This is correct, CLIMP-ST doesn't need to be 480m vertically as stated in the same paragraph: "This value of 480m can be different in CLIMP as it can be changed in COSP/lidar, but averaging le lidar signal vertically before cloud detection should remain the way to increase ATLID SNR when needed for climate mode evaluation".

Note that CLIMP is not the only cloud product from ATLID (there is another one that makes full use of ATLID capability and not dedicated to climate model evaluation). But, CLIMP-ST is dedicated to climate model evaluation, therefore we can only average the signal vertically to increase the SNR (we cannot average horizontally as this is not consistent with COSP subgridding module that transfer the modeled cloud at the GCM grid size to smaller scales). Keeping 480m vertical resolution in this paper is a first attempt, if after launch ATLID SNR is better than expected, we will reduce this vertical resolution (with always keeping full horizontal resolution).

RC, Line 115: I find this statement confusing, as the noise characteristics of daytime space-based lidar data are very definitely scene-dependent. since daytime background levels (and hence profile SNR) are so strongly scene dependent, maximizing correct detections while minimizing false detections during daytime operations would seem to require a threshold that *is* scene-dependent. sampling biases are inherent in either approach. e.g., per Chepfer et al., 2010, using a constant (i.e., scene-independent) threshold ends up rejecting ~30% of all daytime profiles. (though perhaps this value has changed in subsequent algorithm updates?) and these are not randomly distributed, but instead occur in profiles measured above especially bright scattering targets (e.g., dense water clouds and snow/ice surfaces). the question is, which biases have the most pernicious impacts on the science conclusions derived from the resulting data sets?

AC: From a pure observation perspective, this comment is right. But, for model evaluation, we need to use a scene-independent threshold because we need to use the same threshold in observations and in COSP, otherwise the clouds are not defined the same way in model and observations, and the model clouds cannot be evaluated against observations anymore. Applying the same set of scene-dependent thresholds in both the observations and in COSP would work if the model had already very realistic clouds (e.g. water cloud at the right

location and time and with the right optical depth every day), which is not the case. In these conditions using the same set of scene dependent thresholds in both the observations and in COSP would lead to confusing model-vs-observation comparison no more useful to evaluate the cloud description in climate models as we would not be able to say if differences are due to default in model cloud description or differences in the threshold use at this location this day for this cloud.

The rejection of 30% of day-time profiles in (Chepfer et al. 2010) is mostly due to not allowing horizontal averaging before cloud detection, this is not pernicious as soon as we know, which clouds are missed in daytime observations (Chepfer et al. 2013).

RC, Line 132: nominal orbit altitude at launch was 705 km, but was lowered to 688 km in September 2018 to maintain formation flying with CloudSat.

AC: We added this information, thanks.

RC, Line 133: define acronym

AC: Fixed, thanks.

RC, Line 136: Figure 1 in Hunt et al., 2009 shows a block diagram of CALIOP. Is there a publication showing a similar diagram for ATLID? If so, the authors should provide pointers to both of these references somewhere in this section. (Figure 5 in do Carmo et al., 2021 is close but not quite as detailed (perhaps because ATLID is a more complex instrument?))

AC: We added a reference to Fig. of (Hunt et al., 2009) and to Fig. 2 (not Fig. 5) of (do Carmo et al., 2021) to the corresponding paragraphs. We believe that the latter plot is sufficient to get a general understanding of the operation of this space-borne lidar/

RC, Line 142: and, like CALIOP, ATLID is polarization-sensitive

AC: fixed, thanks.

RC, Line 145: this is a badly outdated reference (e.g., lidar ratios for "desert aerosols" are given as "between 17 and 25 sr for 532 and 1064 nm" which we now know are wildly incorrect). maybe instead consider this CALIPSO-HSRL comparison (https://doi.org/10.5194/amt-7-4317-2014) or this new AMTD preprint (https://doi.org/10.5194/amt-2022-306) or some of the many references therein.

AC: We opted to take (Rogers et al., 2014), thanks.

RC, Line 148, Table 1: given as 705 km on line 132 above. perhaps explain the discrepancy (i.e., due to CALIOP's orbit altitude change) in the table caption?

AC: we added this information both to the text and to the table caption.

RC, Line 148, Table 1: 1 m; see Hunt et al., 2009 (https://doi.org/10.1175/2009JTECHA1223.1)

AC: Fixed, thanks.

RC, Line 148, Table 1: 480 m seems like an awkward choice for comparing the the two sensors. why not 300 m instead? can the current COSP/GOCCP models be run at ~300 m resolution?

AC: We agree that the number might seem to be strange, but it's a tradeoff related to noise characteristics of CALIOP lidar. Comparison with the model forbids horizontal averaging (Chepfer et al., 2010, Feofilov et al., 2022), so the only option is vertical averaging, and we already know that 480 m assures reliable SNR.

RC, Line 148, Table 1: quantum efficiency? do Carmo et al., 2021 says 0.79 for the molecular and co-polarized particulate channels and 0.75 for the cross-polarized particulate channel

AC: Fixed, thanks. The calculations have been redone with these numbers

RC, Line 148, Table 1: 0.11 @ 532 nm; see Hunt et al., 2009 (https://doi.org/10.1175/2009JTECHA1223.1)

AC: Fixed, thanks.

RC, Line 149, I believe these data should be taken instead from tables 1 and 2 in Hunt et al., 2009 (https://doi.org/10.1175/2009JTECHA1223.1)

AC: Fixed, thanks.

RC, Line 149: this needs much more explanation. table 1 in the do Carmo 2021 reference gives a per pulse energy of 35 mJ @ 51 Hz. I find no mention of doubling the energy at half the rep rate in that work.

AC: These are the values for the laser itself, but the data will be collected, averaged, and downloaded for two pulses, so it is fair to halve the effective repetition rate and to double the effective pulse energy. We explain this in the caption and we provide both the original and the modified values, so that the reader can compare him- or herself.

RC, Line 153: I don't think this is correct. since scattering ratios calculation require using the ATLID molecular channel and the molecular channel has a wider solar filter bandwidth, I'd think the noise in the molecular channel would dominate the "clear sky" scattering ratios and hence be the dominant factor when separating weakly scattering cloud signals from the ambient noise.

AC: This would be true if we used measured molecular backscatter in our calculations. However, in our approach we use only the information from the particulate channel whereas the noisy molecular backscatter and extinction are replaced with smooth profiles estimated from the concentration.

RC, Line 178: this discussion should make a clear distinction between multiple scattering for ice clouds and multiple scattering for water clouds. the present discussion only discusses ice clouds.

AC: We added this information to the discussion, thanks.

RC, Line 173: while I appreciate the authors' desire to cover "only the basic definitions needed for understanding", I still think more foundation is necessary here. in particular, since ATLID is a polarization-sensitive system, the authors should acknowledge that fact by providing equations describing all three measurement channels and by briefly explaining how the parallel and perpendicular channel measurements of particulate backscatter are combined.

AC: In the updated version of the manuscript, we provide a polarization-related "disclaimer" in the Introduction. The manuscript in its current form is already saturated with the equations, and adding another set of lines for perpendicular component will make it more difficult for understanding without changing the main message.

RC, Line 182: it's important to recognize that this statement applies to ice clouds only. CALIOP's mulitple scattering factors for water clouds are derived from the relationship developed in Hu et al., 2007 (https://doi.org/10.1364/OE.15.005327; also see Table 4 in Young et al., 2018)

AC: We added this information to the paragraph, thanks.

RC, Line 187: this statement is perhaps open to misinterpretation.  if the CALIOP and ATLID multiple scattering factors can be varied independently by ±0.1 without changing the conclusion, then presumably using CALIOP = 0.675 (i.e., 0.6 + 0.075) and ATLID = 0.675 (0.75 - 0.075) give results that are largely identical to CALIOP = 0.6 and ATLID = 0.75. since I doubt this is what is actually meant, some clarifications of the text would be most welcome.

AC: In the new version, we specify "the conclusions of the present work do not change if we vary η within ±0.1 for CALIOP or for ATLID."

RC, Line 190: please add a sentence explaining the purpose of this additional constraint.  off the top of my head, I believe it will limit the detection of weakly scattering clouds at higher altitudes by, in effect, enforcing a larger scattering ratio for detection in regions where the molecular attenuated backscatter is weaker.  is that actually its intended function?

AC: We added an explanation after new Eq. 6. The purpose of this additional criterion is to get rid of false positive cases in the upper troposphere.

RC, Line 193: Since the need for this conversion is not immediately apparent, the authors should clearly explain their motivation for using "recalculated 532 nm values of ATB". Assuming sufficiently accurate radiometric calibrations and corrections for interferometer crosstalk have been applied to the measured data (and that equation (2) represents the sum of both parallel and perpendicular contributions), the sum of equations (2) and (3) divided by equation (4) yields the same quantity as equation (1).  And since the denominator in equation (5) is readily available from model data, instead of converting the 355 nm measurements to "recalculated 532 nm values of ATB", why not simply convert the threshold to a value appropriate for use at 355 nm (i.e., ~1.75)?  If this simple conversion is not a feasible approach, please explain why.  What is the rationale for the additional computational complexity?  and what subsequent benefits are derived by using this (seemingly convoluted) approach.  if this is discussed in depth in some other publication, please also recapitulate the salient points here.  (having not yet read ahead in the paper, at this point I can only speculate that it has to do with multiple scattering differences and the estimated penetration depths into opaque layers.)

AC: Please, see the answer to the 3rd major comment.

RC, Line 197: I suspect equation (8) in Feofilov et al., 2022 is what's actually used in the detection scheme.  my suggestion is to minimize potential reader confusion by reproducing that same equation in this manuscript.  the development given here could be challenging for readers who are not already well versed in the authors' previous publications.

AC: We copied the Eq. 8 to this article.

RC, Line 199: please explain how opaque layers are detected.  is opacity inferred from the failure to detect a surface return in any profile?  if so, is the search of a surface return also conducted using data averaged vertically to 480 m?

AC: We added a definition to new Eq. 9.

RC, Line 203: the wording here is somewhat unclear. is Copaque also reported at single profile resolution?  i.e., for each profile, we not only know CF(z), we also know whether the profile contained an opaque layer. (or maybe Copaque is something to be inferred from Zopaque? e.g., Zopaque is a fill value when no opaque layer is detected?)

AC: The OPAQ algorithm of (Guzman et al., 2017) operates on instantaneous profiles, and we write about it in this paragraph "For an individual lidar profile, …"

RC, Line 216: please provide a concise overview of this methodology. references are fine for readers who want to explore details, but the broad outlines of the technique(s) should also be summarized in this manuscript.

AC: The updated version of the manuscript contains all necessary details, so we just added a reference to Section 3.2.2 in this paragraph.

RC, Line 212: can be retrieved from … but cannot be obtained from

AC: We changed the wording, thanks.

RC, Line 219: consider adding some references that describe the importance of these clouds in modulating the Earth's energy balance (e.g., maybe Berry et al., 2019; https://doi.org/10.1175/JCLI-D-18-0693.1)

AC: we agree with the suggested reference, and we added it to the text.

RC, Line 222: this sounds like a huge and impressive amount of work. but was it really necessary? could the same effects be obtained by ingesting CALIOP (or CATS or GLAS) backscatter profiles? or perhaps cirrus profile measurements acquired by an airborne HSRL? (or, for transparent layers, by an uplooking HSRL.) doing this should capture "natural variability" in cloud scattering intensity (which is what really matters for detection) about as well as could be possibly be done.

AC: Please, see the answer to a corresponding major remark.

RC, Line 223: only strato-cu and cirrus? or are other cloud types also simulated?

AC: 3DCLOUD (Szczap et al., 2014) and 3DCLOUD_V2 (Alkasem et al., 2017) can generate stratocumulus, fair weather cumulus and cirrus cloud. It is not possible to control the cloud coverage of cirrus with 3DCLOUD and 3DCLOUD_V2. During a master's internship, we modified 3DCLOUD in order to simulate Arctic mixed phase cloud (this work is not yet published). 3DCLOUD_V3 (used and shortly present in his paper) is devoted to the generation of cirrus cloud with the control of the cloud coverage. In the updated version of the manuscript, we added the following to this line: "… spatial structures of stratocumulus, fair weather cumulus and cirrus…"

RC, Line 224: this seems like a limited and esoteric list. one hopes that more mundane properties such as top and base heights, vertical and horizontal extents, and optical depths are also realistically rendered in the simulated data set. it would be helpful to have some remarks in the text to explain how these essential parameters are derived.

AC: In 3DCLOUD and 3DCLOUD_V2, 3D water content of cloud are generated in two distinct steps (see Fig.1 in Szczap al., 2014). During the first step, mean vertical profiles of 3D cloud water content (and consequently cloud top and base heights) are driven by the humidity, pressure and temperature vertical profiles provided by the user. Theses meteorological vertical profiles and cloud coverage are assimilated, and basic atmospheric equation are resolved, that provide cloud general 3D shape. The maximum horizontal extension, which mainly depends on the horizontal spatial resolution of the pixel, must not exceed a few tens of km. The vertical extension generally does not exceed 2 km.

We have demonstrated the robustness of 3CLOUD for DYCOM2-RF01 BOMEX case for stratocumulus and for cirrus by comparison with the work of Starr et al. (2000) and Hogan et Kew (2005).

In the second step, stochastic processes, based on Fourier framework, are used to ingest the scale invariant properties observed in the real cloud, such as the power spectra of the logarithm of their microphysical or optical properties , that typically exhibits a spectral slope

$\hat{\beta}$ of around −5/3 (Davis et al., 1994, 1996, 1997, 1999; Cahalan et al., 1994; Benassi et al., 2004; Hogan and Kew, 2005; Hill et al., 2012; Fauchez et al., 2014) from small scale (a few metres) to the "integral scale" or the outer scale $L_{out}$ (few tenths of a kilometer to one-hundred kilometers), where the spectrum becomes flat (i.e. decorrelation occurs). The spectral slope $\hat{\beta}$ value characterizes spatial organization of clouds patterns and/or spatial correlation of cloud variability. But these techniques do not strictly quantify intensity of cloud variability. From practical point of view, an interesting definition of the measure of cloud variability is the "relative variance" as proposed by Davis et al. (1997) or the so-called "cloud inhomogeneity parameter" $\rho$ as defined in Szczap et al. (2000a, b, c). For a variable $X$, the cloud inhomogeneity parameter is defined as the standard deviation of $X$ normalized by its mean value such as $\rho_X = \sigma_X / \bar{X}$. This "cloud inhomogeneity parameter" is also called the "fractional standard deviation" (Shonck et al., 2010 ; Hill et al., 2012, 2015; Bouttle et al., 2014 ; Ahlgrimm and Forbes, 2016, 2017) or the "relative dispersion" (Liu and Daum, 2000 ; Huang et al., 2014) or the "relative standard deviation" (Los and Duynkerke, 2001). The square of the cloud inhomogeneity parameter is called the "fractional variance" in Hogan and Illingworth (2003).

If the number of voxels is large, the 3DCLOUD and 3DCLOUD_V2 are very time-consuming (see Table 1 in Szczap et al., 2014) and cannot assimilate the fractional coverage for cirrus cloud. Therefore, we have developed 3DCLOUD_V3 that overcomes these two drawbacks for the cirrus cloud. 3DCLOUD_V3 do not need anymore the vertical profiles of the pressure, temperature and humidity but more simply the vertical profile of the ice water content. This model will be published in detail elsewhere.

In a general way, 3DCLOUD, 3DCLOUD_V2 and 3DCLOUD_V3 generate water content, not the optical depth. Therefore, we have to do assumption in order to simulate 3D optical properties from 3D water content. In 3DCLOUD, 3DCLOUD_V2, optical depth $\tau$ of stratocumulus is simply derived from the formula $\tau = 3LWP/2\,\rho R_{eff}$, where LWP is the liquid water path, $\rho$ is the density of water and $R_{eff}$ the effective radius. We have to note that the exact tailoring of the cloud parameters to reproduce all the peculiarities of a certain cloud scene was not the goal of using 3DCLOUD in this work. The main goal was to have a noise-free cloud dataset at the resolution finer than the distance between consecutive shots of the lidar, which shares some statistical properties observed in real clouds. Then we launch the same simulator for the same high-resolution scenes with the parameters corresponding to two different lidars and compare their performances.

RC, Line 225: are there some special symbols that are missing here?

AC: No symbol is missing. It was a typo, which has been corrected for the present version of the manuscript.

RC, Line 225: in earlier sections of the paper beta is also used to denote backscatter coefficients. to minimize the potential for confusion, I suggest changing this instance of beta to some other symbol.

AC: We understand that this might be a source of confusion, but the spectral slope symbol is also an established one. In the revised paper, the spectral slope "beta" or "β" is now written as $\hat{\beta}$ to avoid the confusion.

RC, Line 227: perhaps add a citation demonstrating that this assumption is reasonable? (not being a cloud modeler, I don't have a clue)

AC: From the 18-month midlatitude 94-Ghz radar dataset, Hogan and Illingworth (2002) found that PFD of IWC is well represented by lognormal or gamma distribution. From aircraft

measurement of tropical and extratropical cirrus, Kärcher et al (2018) found that observed PDF of total water content are reasonably well approximated by Gamma distribution. In the revised paper, we added both aforementioned references.

RC, Line 245: are arbitrary shapes accommodated? e.g., could one extract IWC profiles from CALIOP level 2 profile data to use in 3DCLOUD_V3?

AC: Yes, they are. In 3DCLOUD_V3, the user can provide, as input of 3DCLOUD_V3, either the 3D mean IWC and the shape model of vertical profiles among the rectangular, upper triangle, lower triangle and isosceles trapezoid model, as proposed in (Feofilov et al., 2015) based the analysis of collocated satellite or either its own vertical profile of the IWC, which could be extracted from CALIOP level 2 profile data. In the revised paper, we moved the sentence "The shape of the vertical profile of IWC can also be stipulated (rectangular, upper triangle, lower triangle and isosceles trapezoid (Feofilov et al., 2015) » from line 251 of the old paper to the line discussed in this comment.

RC, Line 245: does the model include an explicit relationship between IWC and extinction? how are backscatter and extinction coefficients and depolarization ratios specified?

AC: In a general way, all 3DCLOUD versions generate only water content. In order to compute optical properties (extinction, single scattering albedo, phase function), the user has to define the nature (liquid or solid water) of particles, the particle size distribution PDF and the shape of particle (generally spherical for droplets). In order to compute lidar backscatter and depolarization ratio profiles, one need a lidar forward model. In this study, extinction, single scattering and phase function are computed in the lidar simulator (see the new section dedicated to signal calculations).

RC, Line 262: suggestion: either move figure A2 into the main body of the text or (much less desirable) move this paragraph of text into the appendix. the goal is to have the description very close to the images so readers (and reviewers!) don't have to jump back and forth over ~20 pages of manuscript real estate to carefully track the correspondences between the two. readers (and reviewers) interested in how well the simulations capture the natural variability of the cloud fields will appreciate having the description and the images close together.

AC: We opted to move the figures from Appendix A to the main text

RC, Line 274: what's the relevance of picking the equinox if your focus is on the tropics?

AC: There was no special reason to pick up this very atmosphere, this was just the question of data availability and we felt obliged to describe the data we used. We believe that the results in the tropics will be the same for the summer solstice or any other season.

RC, Line 278: I'm afraid it's not entirely clear to me what's being done in this simulation. are the authors constructing a two-layer scene – i.e., thin cirrus over dense stratus? that would certainly qualify as one of "the most challenging observation conditions". and it would also be genuinely interesting, especially for low SES angles that would generate very high levels of solar background noise and make daytime cirrus detection especially challenging. or are they instead building two different scenarios; i.e., (1) thin cirrus in otherwise clear skies and (2) dense stratus in otherwise clear skies. this pair of scenes would provide the necessary baseline for comparing retrieval performance in the two-layer scene. but neither of these single layer scenes would be as challenging at the two-layer option. please clarify the text to eliminate potential confusion about the intended simulation scenes.

AC: For the sake of simplicity, we do not consider a two-layer scene even though it is technically possible. To clarify the text, we added the following phrase to this paragraph: "We do not consider another challenging case, thin cloud layer above a highly reflective cloud, but

the daytime noise estimated for stratocumulus scene will give an idea of what background noise will be interfering with the useful cloud signal in this case."

RC, Line 280: what's the minimum optical depth of the simulated stratocumulus? according to Leahy et al., 2012 (https://doi.org/10.1029/2012JD017929) over half the low clouds over oceans are transparent to CALIOP. does the simulated distribution agree with this experimental finding?

AC: As one can see from the updated version of the plot with stratocumulus clouds (Fig. 5cd in the new version of the manuscript), there are some clouds, which are semi-transparent (optical depth is less than 3). The statistical analysis of cloud columns tells that for a given dataset the number of semi-transparent clouds is about 38% (check the probability density function in Fig. 2 below).

[Figure]

Fig. 2. Probability density function for the optical depths of stratocumulus clouds used in this study, accumulated in vertical columns.

The aforementioned article is dedicated to optically thin marine clouds, so we are not sure whether one should validate our results versus the numbers given in article. Technically, one can adjust the parameters of the model to fit the observations, but this would require additional iterations without any specific outcome for the purposes of this work. The figures and the statistics related to the stratocumulus clouds would have remained the same. We added the reference and a short discussion to the paragraph related to Fig. 5. As for the minimum optical depth, we found that the accumulated optical depth of the faintest cloud show in Fig. 5 is 3e-4, but we afraid that this value itself is not very meaningful, considering the number of thick clouds generated for this scene.

RC, Line 289, Figure 1: how is the noise generated? is it generated for each single shot profile prior to averaging? and what surface albedos are used for daytime data?

AC: The Poisson noise (or NSF-modulated Poisson noise in the case of CALIOP) is generated for each single shot and saved. If the profile corresponds to an average of several shots, the noise is scaled as an inverse square root of number of shots. Please, see the answer to the 6[th] major comment and the new section 3.2.2 in the manuscript.

RC, Line 289, Figure 1: I thought detection was done at 480-m?

AC: This is true, we have updated the figure, thanks.

RC, Line 296: please expand the description of the processes involved in this "third part of the simulation chain". in particular, please explain (a) the differences (if any) between ATLID

and CALIOP detectors; CALIOP uses analog detection. does ATLID use photon counting? (b) the differences in solar background rejection capabilities (I suspect these differences explain a huge fraction of the SNR improvement of ATLID relative to CALIOP). the goal of additional description would be to provide readers with a basic understanding of *why* the ATLID daytime SNR is expected to be better than CALIOP's;

AC: In the updated version of the manuscript, we have a new section dedicated to all these issues. We also provide a back-of-envelope estimate of the ratio of solar photons coming to particulate detector of ATLID to number of solar photons reaching the surface of CALIOP's detector per same sampling interval. This ratio is equal to 0.38, and since we do not use the molecular channel (which measures at least the same number of photons), we do not increase the noise. If one adds the molecular channel with its noise, the ratio will be close to 1, but we do not suggest using the molecular channel for cloud detection.

RC, Line 299: accurate characterization of this crosstalk is critically important for retrieving reliable estimates of βpart(355nm, z)

AC: This is true, but, as we already stated, calibration issues are out of the scope of this work.

RC, Line 306: (a) this statement should be made earlier in the manuscript (i.e., in section 2.2, if not earlier) ;(b) don't ask the readers to *assume* the perpendicular detection and calibration is done the same way in both systems; instead, tell them (briefly!) how it is done and point to references from which they can obtain additional detail

AC: We added an explanation to the introduction: "For the sake of simplicity, we do not discuss the depolarized component of the radiation backscattered by particles, assuming that it is backscattered the same way at these wavelengths and that one can always consider a sum of parallel and perpendicular backscatter for cloud detection. "

RC, Line 313: false negative

AC: added, thanks

RC, Line 314: false positive

AC: added, thanks

RC, Line 320: do you mean "arbitrarily"?

AC: Fixed, thanks.

RC, Line 326, Figure 2: consider adding additional annotations to label the night data (a, c, and e) from the day data (b, d, and f)

AC: We have labeled the left hand side and right hand side panels as "Nighttime scene" and "Daytime scene", respectively

RC, Line 331, Figure 3: should be "liquid water content", yes?

AC: Fixed, thanks.

RC, Line 331, Figure 3: check spelling of "accumulated"

AC: Fixed, thanks.

RC, Line 331, Figure 3: consider using a log scale for the optical depths; these should span a range from ~1 to ~50

AC: We have switched to log scale, but since it's an accumulated optical depth, it increases rapidly, and the log scale does not help that much. Still, one can tell demi-transparent clouds from totally opaque ones.

RC, Line 335: suggested revision: "at present there is no space-based measurement that can retrieve all of the optical properties of..."

AC: We have updated the sentence, thanks for the suggestion.

RC, Line 343: which panels show which signals? please state this in the text. also please correct the figure caption.

AC: Fixed, thanks.

RC, Line 348: what surface type/albedo was used?

AC: In the new section dedicated to the calculation of signals and noise, we write "the surface with albedo equal to 0.08 for ocean and 0.15 for land (arbitrary values)".

RC, Line 350: only from CALIOP? are the authors claiming that the ATLID signals are not at all affected by solar background noise?

AC: We have added the word "especially" before "for the CALIOP" to stress that both will be affected, but for CALIOP the effect will be stronger.

RC, Line 350: while FOV differences may play a part, this is not a sufficient explanation. if FOV was the sole deciding criteria, then CATS (FOV = 110 microradians) would have had somewhat better daytime SNR than CALIOP (FOV = 130 microradians). but, as seen in Figure 1 of Pauley et al., 2019; (https://doi.org/10.5194/amt-12-6241-2019), this is demonstrably false: CATS' daytime SNR is much worse than CALIOP's. given the importance of solar background rejection (or lack thereof!) for daytime detection of faint features, I believe the authors should provide much more detail to explain the differences between CALIOP and ATLID background levels. (note too that CATS also flew at a substantially lower altitude than CALIOP, which, all things being equal, should also lead to better SNR than CALIOP. but since all things are not equal, more detailed explanations of the instrument differences are required if readers are to understand the differences in the simulated signals and the cloud detection results being presented here.)

AC: Please, see the new section and the back-of-envelope estimates of ratio of daytime noises, where we list the components related to the number of photons hitting the surface of the detector. This is not the end of the story because the NSF ~=5 further aggravates the situation for CALIOP, but this number alone (0.38) tells us that one has to expect lower solar noise for ATLID.

RC, Line 359: yikes! which set of panels belongs to CALIOP and which set belongs to ATLID? according to the caption, panels a) through d) show data from both instrument while panels e) through h) are not attributed to either instrument.

AC: Fixed, thanks.

RC, Line 360, Figure 5: I think readers could extract much more useful detail from these figures if they used a log color scale

AC: Please, see our answer on using log scale in major comments section.

RC, Line 366: is this true even for estimates of apparent base altitude (i.e., Zopaque) and cloud fraction?

AC: We did not check this specifically, but the signal from stratocumulus clouds is strong, so we do not expect any adverse effects here.

RC, Line 383: I'd think the authors' choice of threshold would be driven by the magnitude of the noise in the signal, not by some arbitrary partitioning of the relative contributions of the

molecular and particulate components of the total backscatter. what was the rationale for the choice of 5 as the threshold for GOCCP? I'd think that applying those same considerations to the ATLID simulations would yield a more useful threshold for use in this exercise; presumably it's ATLID's lower daytime noise that permits the use of the lower threshold. So an in-depth discussion of noise magnitudes would be extremely useful here in helping to understand the rationale for choosing a specific value for the threshold used in the simulation study.

AC: We changed the wording of this paragraph to make it closer to the suggested explanation. Indeed, we based our choice on the SNR values, and added the explanation with partitioning only later.

RC, Line 390: the effect of this second condition is simply to raise the detection threshold substantially above its nominal value of 3

AC: Not exactly. Please, see our explanation to line 190. The detection threshold will be raised only in the upper layers.

RC, Line 403: thank you for this. (this sentence answers my earlier question about whether the authors had constructed a multi-layer scene in addition to the two single layer scenes.)

AC: yes, and we've added a sentence on two-layer cloud scene.

RC, Line 406, Figure 6: this choice of colors makes it difficult to distinguish between "no cloud" and "missed". using white or a very pale gray for "no cloud" regions would help enormously (e.g., see figures 5.5 and 5.7 in the CALIPSO layer detection ATBD at https://dev-calipso.larc.nasa.gov/resources/pdfs/PC-SCI-202_Part2_rev1x01.pdf)

AC: We updated the color scheme, and on our monitors the bright blue "missed" cases are now clearly distinguishable from black "no cloud" points.

RC, Line 406, Figure 6: a log color scale would better highlight regions where a purely molecular atmosphere transitions to a faint cloud. these regions will be the most interesting when discussing possible detection improvements of ATLID relative to CALIOP.

AC: Please, see our comment to log scale plots in the main comments section. We shifter the color scale to make the features visible.

RC, Line 406, Figure 6: consider rearranging the plots so that scattering ratios are on the left (i.e., beneath the scattering ratio color bar) and cloud detection results are on the right (i.e., beneath the cloud detection color bar). put the nighttime data on top and the daytime data underneath.

AC: thanks for the suggestion. Indeed, it's more logical like this, so we have rearranged the plots.

RC, Line 409, Figure 7: are these estimated attenuated scattering ratios derived using equation 5? or are they particulate scattering ratios derived by dividing equation 2 by equation 3? I'd expect somewhat more noise for the attenuated scattering ratio estimates.

AC: No, these are the SR values as we define them in Eq, 5, but the AMB component is calculated without noise. This is, as we said before, one of the advantages of our method.

RC, Line 412, Table 2: 90?

AC: There are rounding errors in this table, because we wanted to stay with zero digits after the decimal point. The new table is also prone to the same issue, but in our opinion it's already too busy.

RC, Line 420: please clarify: this is a 1 km along-track distance, correct? (I note that at cirrus altitudes the CALIOP data is averaged on-board to a 1-km (3 shot) along track resolution. if ATLID will be downlinking data without horizontal averaging, (a) ATLID will be able to probe cirrus small scale variability that CALIOP cannot, but (b) to faithfully compare CALIOP to ATLID for CLIMP-LT requires some along-track averaging of the ATLID data prior to detecting layers. also provide more detail. for CALIOP, averaging to 1 km along-track equates to averaging the backscatter from 3 consecutive laser pulses. but the nominal distance between consecutive ATLID footprints is 285 m, and 3 x 285 = 855 and 4 x 285 = 1140. (1000/285 = 3.5088) so how many ATLID laser pulses were averaged?

AC: To compare apples to apples in terms of signal statistics, we took 4 laser shots of CALIOP and 4 laser shots of ATLID that are equivalent to two "effective" doubled laser shots (see the answer to the comment to line 149). We added this information to the text.

RC, Line 404: but, per the previous comment, since CALIOP data above ~8.2 km is averaged to 1km along-track resolution prior to being downlinked, these 1-km averages are a much more accurate representation of CALIOP's actual performance.

AC: this is true, and with the updated noise estimates we do not show the profiles without averaging, even though we discuss them.

RC, Line 433: it might be very interesting/useful to estimate ATLID's "minimum detectable backscatter", as is done initially in CALIOP's layer detection ATBD (https://dev-calipso.larc.nasa.gov/resources/pdfs/PC-SCI-202_Part2_rev1x01.pdf) and subsequently verified in McGill et al. 2007 (https://doi.org/10.1029/2007JD008768)

AC: We thank the reviewer for this idea. We have estimated the MDB value using the following approach: in the simulated dataset, we have scanned the noisy ATB profiles altitude per altitude and for each subset we took only the points which contained the clouds. Then we analyzed the remaining subset by moving a 5km window and analyzing the signal and noise in this window. The signal values for the points with SNR = $1\pm0.1$ were stored in an array, which was analyzed after all altitudes and subsets have been scanned this way. Namely, we have built a histogram of these minimum signal values and analyzed its peak near zero. Its location and halfwidth were used as the MDB value and its uncertainty. For the ATLID, the approach was different in a sense that we used noisy APB at 355nm summed with noise-free AMB at 532nm to imitate the ATB signal at 532 nm to compare with MDB for CALIOP. According to our estimates made over our cirrus data set, the minimal detectable backscatter for CALIOP should be $4.0\pm2.0 \times 10^{-7}$ m$^{-1}$ sr$^{-1}$ for the nighttime and $1.3\pm0.2 \times 10^{-6}$ m$^{-1}$ sr$^{-1}$ for the daytime if we use the same averaging distance as in (McGill et al., 2007). The daytime value obtained in our approach for CALIOP is in good agreement with (McGill et al., 2007). The nighttime value is somewhat lower than that retrieved from the measurements. For ATLID, we obtained $3.0\pm1.0 \times 10^{-7}$ m$^{-1}$ sr$^{-1}$ for the nighttime and $4.0\pm1.0 \times 10^{-7}$ m$^{-1}$ sr$^{-1}$ for the daytime in equivalent ATB at 532nm. Qualitatively, this agrees with the nighttime and daytime behavior of CALIOP and ATLID signals in Fig. 8-11. If these results will be confirmed in a real mission, the MDB of ATLID will be comparable to that of nighttime MDB of CALIOP for 5km averaging. We added a short paragraph with these numbers to the end of Section 3.4.

RC, Line 460: since CALIPSO data is averaged to 1-km resolution aboard the satellite, I'd think that it was the comparisons at 1-km resolution that were really important.

AC: we agree with this statement, and we changed the discussion accordingly.

RC, Line 468, Table 3: is this a typo? should these be 95 instead?

AC: Please, see our reply to Line 412 above.

RC, Line 493: please explain how opaque clouds are identified. do the authors expect ATLID and CALIOP to have similar performance in identifying opaque clouds? please explain. given that ATLID is supposed to have less noise than CALIOP, especially during daytime, one might expect some differences. how are these differences reconciled in assembling CLIMP-LT?

AC: The COSP simulator described in Section 4.2 takes as input monthly-averaged global 3D grids of atmospheric properties generated by GCM, and from them creates consistent grids of simulated lidar-derived cloud properties. Note that the COSP simulator chain used in Sect. 4.2 involves a first step for application on GCM outputs, which is absent from the COSP simulator chain that is directly applied on high-resolution synthetic profiles (as in Sect. 3.2). The process through which the COSP simulator uses GCM globally gridded outputs, and from them generates simulated lidar-derived cloud properties, is described in Chepfer et al. 2008. To sum up, this process involves generating for each GCM gridbox an ensemble of subgrid-scale profiles by generating 50 subcolumns stochastically (using the SCOPS subgrid-scale scheme, Klein and Jakob 1999), with the constraint that statistics over the subcolumns should respect the initial GCM profile. The lidar simulator is then applied to each of the subcolumns, and the 50 simulated profiles are eventually averaged to obtain a single gridbox value. This is now clarified in the text.

About the expected difference in opacity detection between CALIPSO and EarthCARE, we understand the reviewer's concerns, which show we did not make our purpose in Section 4 clear. In that section, we do not attempt to find ways to make ATLID cloud properties consistent with those derived from CALIOP, nor do we try to evaluate how the performance of each instrument compare in similar cloud situations. Instead, we start with the hypothesis that each instrumental and orbital difference between CALIOP and ATLID are fully compensable, that it will be possible to make ATLID and CALIOP cloud properties consistent and merge them, and that it will be possible to construct a merged record that combines cloud variables from both instruments with perfect intercalibration. Given the possible existence of such a perfectly merged record, we investigate how long the ATLID record should be to make reliable anthropogenic trends emerge from the merged record. Since it is unlikely that all differences between CALIOP and ATLID should be perfectly compensable, our study provides a best-scenario prevision of the record length required.

Following this comment, we have tried to make the purpose of the analysis clearer in the text

RC, Line 501: I'm surprised that maximum cloud top altitude isn't also one of the diagnostics investigated here

AC: The top altitude is indeed a property that is closely linked to cloud feedbacks on climate. Here we have first focused on the altitude of full attenuation as it has already been studied extensively in the past. We intend to include the cloud top altitude among the properties we will consider in future studies.

RC, Line 516: this statement throws up all sorts of red flags. I thought the whole point of the first part of this paper was to *demonstrate* that ATLID and CALIOP could be fused into a relatively seamless time history of cloud heights.

Indeed, the point of the first part of the paper is to show that ATLID and CALIOP could be fused into a relatively seamless time history of cloud heights. However:

1) The results of this first part are based on simulations, and ATLID signals might eventually behave in ways that are today unforeseen.
2) The first part investigates how ATLID and CALIOP cloud detections could be merged. To do this, it investigates how the match in GOCCP cloud detections from

both instruments can be improved through changes in signal thresholds. Beyond this strategy, there remains several differences whose impact on detection should be evaluated, for instance the different local time of overpass of both missions. Depending on the magnitude of the associated effects, ways to compensate for them might be required, whose effects on fusion performance are still unclear.

3) The first part focuses on the evolution of Scattering Ratios within clouds. The last part focuses on statistics of lidar full attenuation due to clouds, since this is a lidar-derived cloud property with a clear relationship to cloud radiative effect and climate feedbacks. In CALIPSO-GOCCP the altitude of full attenuation is derived from both the vertical profile of Scattering Ratios and the surface echo at the finest CALIOP horizontal resolution (333m). The behavior of ATLID surface echo compared to CALIOP's in various cloud situations is still not well understood, and making them consistent will require further investigation.

Even though the first part of the paper is a necessary first step towards showing that CALIPSO and ATLID cloud detections can be merged, the reasons above suggest these results are not sufficient to guarantee that merging will eventually be possible. It thus seems to us prudent to remind the reader that further work is still required to make CALIPSO and ATLID cloud properties consistent.

Because we cannot today evaluate completely how each of the points mentioned above will affect the continuity of the lidar-derived cloud record, we decide in the rest of the section to assume perfect intercalibration will eventually be reached, in order to provide a best-case scenario of the record length required to detect an anthropogenic trend in cloud properties.

Following this comment, we have tried to express our intent in a clearer way in the text. We now explain straight away (Section 4.2) that in this section we start from the hypothesis that it will eventually be possible to reconcile both CALIPSO and EarthCARE records.

RC, Line 519: how is Copaque determined?

AC: Please, see the answer to the previous comment

RC, Line 522: normal distribution assumed?

AC: Indeed. This assumption is now made explicit in the text

RC, Line 548: I can understand excluding CALIOP data prior to November 2007, when the off nadir angle was changed from 0.3° to 3.0°. but what is the rationale for excluding the data from 2008 through 2010? please explain.

AC: We thank the reviewer for his comment, that highlights poor writing on our part. The two first years of CALIOP data (2008 and 2009) are not excluded from the analysis. However, during the first two years of the record, trends can only be retrieved from a limited number of points and fluctuate wildly. The associated uncertainties are accordingly quite large. To represent the large uncertainties of the first two years, the axes in figure 10 and 11 need to be scaled up, which makes it a lot harder to decipher the behavior of uncertainties once the record gets longer than 2 years (2010 and later). Since the focus here is on what happens when the uncertainties get relatively small, i.e. once the record gets longer than ~10 years, we decided to mask the uncertainties before 2010 so the later uncertainties would be easier to read. The legend of figure 10 now makes explicit 1) that years 2008 and 2009 were part of the analysis and 2) the reasons for masking the first two years of trend uncertainties.

RC, Line 616: two points here. first, HSRL capability will be especially advantageous for aerosol studies, but those are not addressed here. second, the cloud detection approach described in section 2 (i.e., the use of attenuated scattering ratio rather than particulate

scattering ratio) effectively ignores many of the advantages of HSRL relative to elastic backscatter lidar.

We partially agree with the reviewer's assessment. Our work does not show explicitly how the HSRL technique will benefit cloud studies. However, we believe that the proposed technique of excluding the measured (and noisy) molecular component helps to increase the daytime SNRs. We have modified the text to rather conclude that our work shows how the EarthCARE HSRL can help reconcile its cloud detections with CALIOP's.

**References:**

Chepfer H., G. Cesana, D. Winker, B. Getzewich, and M. Vaughan, 2013: Comparison of two different cloud climatologies derived from CALIOP Level 1 observations: the CALIPSO-ST and the CALIPSO-GOCCP, J. Atmos. Ocean. Tech., doi.10.1175/JTECH-D-12-00057.1

Feofilov. A.G., Chepfer, H., Noel, V., and Szczap, F., Towards establishing a long-term cloud record from space-borne lidar observations, EECLAT 2023 Workshop, Banyuls, France, available online at https://eeclat.ipsl.fr/2023/02/13/eeclat-2023-workshop-23-26-jan-2023-banyuls, 2023

---

## Referee Report (RR1)

**A Second Review of "Incorporating EarthCARE observations into a multi-lidar cloud climate record: the ATLID Cloud Climate Product" by A. G. Feofilov, H. Chepfer, V. Noël, and F. Szczap**

Reviewed by Mark Vaughan (mark.a.vaughan@nasa.gov)

The authors' revision does an excellent job of addressing my many and highly detailed comments on their original manuscript. The newly added paragraph beginning on line 113 nicely summarizes the goals of the study while at the same time clearly explaining the rationale some self-imposed limits (e.g., their choice to deemphasize contributions from the perpendicular channel signal). Throughout the manuscript, assumptions underlying various choices in the analysis are now clearly identified. And the depth of discussion and detail added in section 3 is genuinely impressive. The added insights on noise generation in the simulations and the minimum detectable backscatter comparisons were both hugely helpful, and I fully expect future readers will agree with this assessment. Overall, I very much appreciate the extra clarity their thorough and thoughtful revisions have added to the manuscript.

I have only one serious quibble with this latest effort. On lines 425–427 the authors say, "However, using this value in synthetic noise calculations leads to an overestimation of the daytime noise, for the calculations below we took a more conservative value $NSF$=3.16, which better represents real CALIOP nighttime and daytime noises in the aerosol-free stratosphere". In their responses to my comments, the authors raise some interesting points about the formulation of the noise scale factor and its application to CALIOP data. However, a brief email discussion of their ideas with Zhaoyan Liu (i.e., the lead author of Liu et al., 2006) highlighted some very real differences of opinion about the correct approach. So, rather than obliquely raising their issues with the original NSF formulation in this manuscript, the authors should instead surface their concerns in a published comment in Applied Optics (i.e., the journal that published Liu et al., 2006). The ensuing response from Liu et al. would ensure that the authors' NSF criticisms are properly resolved in a totally public and transparent way that would be readily available for review and comment by the entire lidar community. With regards to the current manuscript, the authors need to delete the adjective clause (i.e., the text in red). While NSF = 3.16 value may be the best choice for the synthetic data they have generated, suggesting that it is also the correct choice for "real CALIOP" is a bold and controversial statement that is not supported by any development given in the manuscript. As demonstrated in the document appended below (which is publicly available via the CALIPSO web site), the NSF values reported in the CALIPSO level 1 data files correctly characterize noise throughout CALIOP's full vertical profile. The correlations between adjacent bins due to the electronic bandwidth of the CALIOP receiver are explained further in Appendix A of Vaillant de Guélis et al., 2021.

Updated on November 7, 2012

Uncertainties for the attenuated backscatter, $\beta'$, are not explicitly reported in the CALIOP Level 1 (L1) data products to save data volume, which would otherwise approximately double the L1 data volume. If needed, users can compute random errors for the attenuated backscatter products using

$$\Delta\beta'(k, r_i) = \left[\frac{r_i^2 \cdot NSF^2(k) \cdot \beta'(k, r_i)}{E \cdot C(k)} + \left(\frac{r_i^2 \cdot RMS(k)}{E \cdot G_A \cdot C(k)}\right)^2\right]^{0.5} \frac{f_{correct}[N_{bin}(r_i), N_{shift}]}{\sqrt{N_{bin}(r_i), N_{shot}}}. \qquad (1)$$

In this equation, $r_i$ is the range from the CALIPSO satellite to the ith range bin, *NSF* the noise scale factor, *E* the laser energy, *C* the calibration coefficient, $G_A$ the gain of the amplifier, *RMS* the random noise of the background signal including detector dark current, background radiation, etc. $N_{bin}$ and $N_{shot}$ are, respectively, the number of range bins and laser shots averaged for the different altitude ranges as shown in Table 1. $f_{correct}$ is a correction factor used to account for the partial correlation among neighboring samples in a raw Level 0 (L0) profile [Liu et al., 2006], and additional correlation due to data redistribution in the altitude registration of L0 data samples during the L1 processing. The integral time of the amplifier of the lidar receiver is slightly longer than 0.02 ms (30 meter in distance) and is larger than the onboard sampling interval (15 m), causing the down linked data (averaged over different numbers of 15-m samples for different altitude ranges as listed in Table 1) to be partially correlated. In addition, there may be an offset in the altitude registration of a profile due to the variation of the nadir viewing angle of the lidar system. In the L1 processing, each 30-m bin in the -0.5 km – 8.2 km altitude range (altitude indices of 288 – 577) is registered to the nearest bin of the altitude array. For the other altitude ranges, because the bin size is larger than 30 meters (60 to 300 meters), the shift is accomplished by regridding then reaveraging the L0 data, thus redistributing the magnitudes of neighboring data samples, and thereby introducing additional correlation in the L1 data. $N_{shift}$ is the number of 15-m bins shifted. $f_{correct}$ can then be computed using

$$f_{correct}(N_{bin}, N_{shift}) = \left\{\left[\left(\frac{N_{bin} - N_{shift}}{N_{bin}}\right)^2 + \left(\frac{N_{shift}}{N_{bin}}\right)^2\right]f^2(N_{bin})\right.$$

$$\left. + 2\left(\frac{N_{bin} - N_{shift}}{N_{bin}}\frac{N_{shift}}{N_{bin}}\right)\left(\sum_{m=1}^{N_{bin}}\frac{m}{N_{bin}}R(m) + \sum_{m=1}^{N_{bin}-1}\frac{N_{bin}-m}{N_{bin}}R(N_{bin}+m)\right)\right\}^{0.5} \qquad (2)$$

where $f(N_{bin}) = \left[1 + 2\sum_{m=1}^{N_{bin}-1}\left(\frac{N_{bin}-m}{N_{bin}}\right)R(m)\right]^{1/2}$ and *R* represents the autocorrelation

coefficients [Liu et al., 2006]. The computed $f_{correct}$ values are given Table 2, using the R values determined based on the prelaunch lab experiment data.

Liu, Z., et al., 2006: Estimating Random Errors Due to Shot Noise in Backscatter Lidar Observations, *Appl. Opt.*, **45**, 4437-4447.

Table 1 numbers of 15-m range bins and laser shots averaged for different altitude ranges in L1B data products

| Altitude range (km) | Altitude index range | 532 nm | | 1064 nm | |
|---|---|---|---|---|---|
| | | $N_{bin}$ | $N_{shot}$ | $N_{bin}$ | $N_{shot}$ |
| 39.9 – 30.3 | 0-32 | 20 | 15 | N/A | N/A |
| 30.0 – 20.3 | 33-87 | 12 | 5 | 12 | 5 |
| 20.2 – 8.3 | 88-287 | 4 | 3 | 4 | 3 |
| 8.2 – -0.5 | 288-577 | 2 | 1 | 4 | 1 |
| -0.6 – -1.8 | 578-582 | 20 | 1 | 20 | 1 |

Table 2 $f_{correct}$ for different altitude range and number of 30 meter bins shifted

| Bin index | $N_{shift}$ | | | | | | | | | | | Remark |
|---|---|---|---|---|---|---|---|---|---|---|---|---|
| | 0 | 1 | 2 | 3 | 4 | 5 | 6 | 7 | 8 | 9 | 10 | |
| 0-32 | 1.598 | 1.450 | 1.324 | 1.226 | 1.163 | 1.141 | 1.163 | 1.226 | 1.324 | 1.450 | 1.598 | Cycle of 10 |
| 33-87 | 1.578 | 1.350 | 1.192 | 1.134 | 1.192 | 1.350 | 1.578 | 1.578 | 1.350 | 1.192 | 1.134 | Cycle of 6 |
| 88-287 | 1.489 | 1.105 | 1.489 | 1.105 | 1.489 | 1.105 | 1.489 | 1.489 | 1.105 | 1.489 | 1.105 | Cycle of 2 |
| 288-577 | 1.386 1.489 | 1.386 1.489 | 1.386 1.489 | 1.386 1.489 | 1.386 1.489 | 1.386 1.489 | 1.386 1.489 | 1.386 1.489 | 1.386 1.489 | 1.386 1.489 | 1.386 1.489 | 532 nm 1064 nm |
| 578-582 | 1.598 | 1.450 | 1.324 | 1.226 | 1.163 | 1.141 | 1.163 | 1.226 | 1.324 | 1.450 | 1.598 | Cycle of 10 |

[Figure]

Figure 1: Random uncertainties computed using equation (1) for a nighttime CALIOP data segment acquired while passing over the southern Atlantic Ocean, as indicated by the white box in the upper browse image.  The lower row of images shows uncertainty estimates for the 532 nm perpendicular (left panel in the lower row) and parallel (middle panel) channels and 1064 nm channels.  The red lines represent the mean of uncertainties calculated using equation (1), and the blue lines show the standard deviation of the single-shot profiles. Good agreement is seen in the NSF-estimated uncertainties and standard deviations, except in the 532-nm perpendicular signal and the upper part of the 532-nm parallel signal where the return signal is very weak.

[Figure]

Figure 2 Same as Figure 1, but for a data segment acquired during daytime.

---

## Author Response (AR2)

We appreciate the feedback provided by the Reviewers on the revised version of the manuscript.

In response to Dr. Mark Vaughan's request, we have removed the portion of the sentence highlighted in red: "which better represents real CALIOP nighttime and daytime noises in the aerosol-free stratosphere."

Furthermore, we have made updates to the Acknowledgements section to properly acknowledge the individuals and organizations who have contributed to the preparation and improvement of this article.